# DELTA-XAI: A UNIFIED FRAMEWORK FOR EXPLAINING PREDICTION CHANGES IN ONLINE TIME SERIES MONITORING

**Changhun Kim**[1*]    **Yechan Mun**[1*]    **Hyeongwon Jang**[2]
**Eunseo Lee**[3]    **Sangchul Hahn**[1†]    **Eunho Yang**[1,2†]
[1]AITRICS    [2]KAIST    [3]Sungkyunkwan University
{changhun.kim, ycmun, s.hahn, eunhoy}@aitrics.com
janghw0911@kaist.ac.kr, dldmstj311@g.skku.edu

## ABSTRACT

Explaining online time series monitoring models is crucial across sensitive domains such as healthcare and finance, where temporal and contextual prediction dynamics underpin critical decisions. While recent XAI methods have improved the explainability of time series models, they mostly analyze each time step independently, overlooking temporal dependencies. This results in further challenges: explaining prediction changes is non-trivial, methods fail to leverage online dynamics, and evaluation remains difficult. To address these challenges, we propose Delta-XAI, which adapts 14 existing XAI methods through a wrapper function and introduces a principled evaluation suite for the online setting, assessing diverse aspects, such as faithfulness, sufficiency, and coherence. Experiments reveal that classical gradient-based methods, such as Integrated Gradients (IG), can outperform recent approaches when adapted for temporal analysis. Building on this, we propose Shifted Window Integrated Gradients (SWING), which incorporates past observations in the integration path to systematically capture temporal dependencies and mitigate out-of-distribution effects. Extensive experiments consistently demonstrate the effectiveness of SWING across diverse settings with respect to diverse metrics. Our code is publicly available at https://github.com/AITRICS/Delta-XAI.

## 1 INTRODUCTION

Time series data, inherently gleaned online, span safety- and mission-critical domains such as healthcare (Johnson et al., 2016; Reyna et al., 2020), transportation (Polson & Sokolov, 2017; Zheng & Huang, 2020), finance (Sezer et al., 2020; Xu et al., 2024), and climate monitoring (Camps-Valls et al., 2021; Reichstein et al., 2019). Despite their success in time series prediction (Mahmoud & Mohammed, 2021; Wang et al., 2024), the black-box nature of deep neural networks (Zhao et al., 2023) and their use in sensitive domains (Wang & Chung, 2022) make explainability essential. When it comes to online time series, practitioners often place greater emphasis on *prediction differences* between adjacent time steps rather than isolated predictions, as identical predictions can *signify different contextual meanings*. For instance, in clinical settings, a decrease in the probability of sepsis from 90% to 50% indicates patient improvement, while an increase from 10% to 50% signals potentially severe deterioration (Boussina et al., 2024; Shashikumar et al., 2017). This epitomizes the necessity to contextualize prediction changes in online time series and attribute them to the features driving such transitions.

Albeit diverse explainable artificial intelligence (XAI) methods for time series (Ribeiro et al., 2016; Lundberg & Lee, 2017; Sundararajan et al., 2017; Shrikumar et al., 2017; Suresh et al., 2017; Tonekaboni et al., 2020; Leung et al., 2023; Crabbé & Van Der Schaar, 2021; Enguehard, 2023; Liu et al., 2024b; Queen et al., 2024; Liu et al., 2024a; Jang et al., 2025) have been suggested, existing approaches mostly explain predictions at isolated time steps, neglecting how predictions have evolved over time. In this regard, directly applying single-time attribution methods to explain prediction

---

*Equal contribution. †Equal advising.

changes poses significant challenges. First, these methods inherently overlook the contextual dynamics of attributions. For instance, this frequently leads to practically irrelevant explanations (see Section 3 for more details). Second, it is generally impossible to explain prediction differences using isolated attributions from individual time steps, which are irrelevant to prediction changes. For example, Table 5 reveals that the existing XAI approach (Crabbé & Van Der Schaar, 2021) produces implausible attributions when subtracting across two time steps. Last, appropriate evaluation metrics for attribution in explaining prediction changes remain largely unexplored. Common evaluation metrics, such as performance degradation after removing salient features, often exhibit limited correlation with attribution quality in evolving prediction settings.

To bridge this gap, we introduce Delta-XAI, a unified framework for explaining prediction changes in online time series monitoring. We begin with a problem setup of *explaining prediction changes in online time series*, attributing features to differences between adjacent steps rather than single-step predictions. Within this framework, we successfully adapt 14 mainstream XAI methods—including general approaches (Ribeiro et al., 2016; Lundberg & Lee, 2017; Sundararajan et al., 2017; Shrikumar et al., 2017; Suresh et al., 2017) and time series-specific ones (Tonekaboni et al., 2020; Leung et al., 2023; Crabbé & Van Der Schaar, 2021; Enguehard, 2023; Liu et al., 2024b; Queen et al., 2024; Liu et al., 2024a; Jang et al., 2025)—via a wrapper function that transforms single-time explanations to directly quantify prediction changes. We also propose a novel evaluation suite for online time series XAI, assessing attributions for faithfulness, sufficiency, completeness, coherence, and efficiency, providing a holistic standard over fragmented practices. Within this framework, we uncover that conventional gradient-based methods like Integrated Gradients (IG) (Sundararajan et al., 2017) typically outperform recent alternatives, consistent with recent findings in single-time settings (Jang et al., 2025).

Motivated by these observations, we introduce Shifted Window Integrated Gradients (SWING), a novel XAI method for explaining prediction changes in online time series. IG typically exploits integration along a straight path from a zero baseline to the input, failing to capture temporal dynamics and inducing an out-of-distribution (OOD) problem. SWING addresses these issues by generalizing the integration path with the retrospective prediction window as baseline and performing piecewise line integrals through intermediate windows between two time steps. This yields faithful and interpretable attributions that capture temporal dynamics and mitigate OOD issues, while satisfying desirable theoretical properties for online time series monitoring—completeness, implementation invariance, and skew-symmetry. Extensive experiments under a wide range of settings, spanning diverse datasets and model architectures, verify that SWING outperforms existing baselines.

Our contribution can be summarized as follows:

- We formulate a problem of explaining prediction changes in online time series monitoring and propose a unified framework that adapts 14 existing XAI methods via a prediction wrapper, alongside comprehensive evaluation metrics tailored to this setup.
- We propose SWING, an XAI method for online time series that extends IG by incorporating past observations into the integration path, enabling temporal dynamics to be captured while mitigating OOD effects, while satisfying key theoretical properties.
- Through extensive experiments across diverse benchmarks and backbone architectures, we systematically analyze existing time series attribution methods and demonstrate that SWING surpasses state-of-the-art alternatives under diverse evaluation metrics.

## 2 PROBLEM SETUP

Let $f : \mathbb{R}^{W \times D} \to [0, 1]^C$ be an online time series classifier where $W$ is the fixed lookback window size, $D$ is the number of features, and $C$ is the number of output classes, which can be either binary or multiclass. Given an online time series $\mathbf{X} \in \mathbb{R}^{L \times D}$ of length $L$, the model receives at each time step $T$ the input window $\mathbf{X}_{T-W+1:T} \in \mathbb{R}^{W \times D}$ and outputs predicted class probabilities $f(\mathbf{X}_{T-W+1:T}) \in \Delta^{C-1} := \{p \in [0, 1]^C \mid \sum_{c=1}^{C} p_c = 1\}$. In a conventional time series XAI setup, the goal is to explain the model's prediction at time $T$ by estimating the contribution of each input feature within the window: for each $t$ and $d$, existing approaches calculate the attribution $\varphi(f, \mathbf{X}_{t,d} \mid T)$, which quantifies the contribution of feature $d$ at time $t$ to the model output $f(\mathbf{X}_{T-W+1:T})_{\hat{c}}$, where $\hat{c} = \arg\max_c f(\mathbf{X}_{T-W+1:T})_c$.

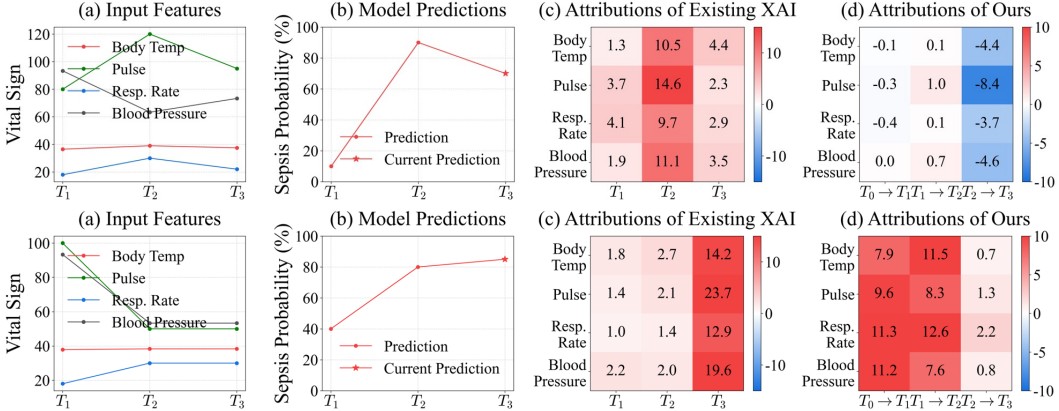

**Figure 1:** Motivation for explaining prediction changes through illustrative scenarios that are not generated by actual XAI outputs. (top) Vital signs across $T_1, T_2, T_3$: risk rises from 10% to 90% then partially recovers (70%). Conventional attribution at $T_3$ misleads, while our method highlights features driving recovery. (bottom) Risk evolves from 10% at $T_1$ to 80% at $T_2$ and slightly increases (85%) at $T_3$ due to delayed effects. Our method attributes prediction changes between $T_1 \rightarrow T_2$ and $T_2 \rightarrow T_3$, resolving these issues. Here, $T_0$ denotes a hypothetical past time step at which the baseline input is placed to compute the initial score.

In contrast, our goal is to explain *prediction changes* between two time steps $T_1 < T_2$. We define the target class as the one with the largest probability increase: $\hat{c} = \arg\max_c f(\mathbf{X}_{T_2-W+1:T_2})_c - f(\mathbf{X}_{T_1-W+1:T_1})_c$, with the corresponding change $\Delta = f(\mathbf{X}_{T_2-W+1:T_2})_{\hat{c}} - f(\mathbf{X}_{T_1-W+1:T_1})_{\hat{c}}$. To explain this change $\Delta$, we compute attributions over $t \in \{T_1 - W + 1, \ldots, T_2\}, d \in \{1, \ldots, D\}$, where each $\varphi(f, \mathbf{X}_{t,d} \mid T_1 \rightarrow T_2)$ quantifies the contribution of feature $d$ at time $t$ to $\Delta$. By default, we take $\hat{c}$ to be the class with the largest probability increase, motivated by the fact that increasing-probability classes are often the most *practically relevant*. For instance, in disease risk prediction, even if *normal* remains the top class, a sharp rise in *respiratory failure* probability from $T_1$ to $T_2$ is clinically significant and merits explanation. However, Delta-XAI also supports flexible setups such as attributing the difference between the predicted classes, *i.e.*, $\Delta = f(\mathbf{X}_{T_2-W+1:T_2})_{\hat{c}_2} - f(\mathbf{X}_{T_1-W+1:T_1})_{\hat{c}_1}$, where $\hat{c}_1 = \arg\max_c f(\mathbf{X}_{T_1-W+1:T_1})_c$ and $\hat{c}_2 = \arg\max_c f(\mathbf{X}_{T_2-W+1:T_2})_c$, though this is not our primary concern. We further restrict to $T_2 - T_1 < W$, as larger gaps yield non-overlapping windows and no common temporal context for attribution.

## 3  MOTIVATION: CHALLENGES IN ONLINE TIME SERIES EXPLANATION

In time series applications, stakeholders often care less about static predictions and more about *why they change*. Standard XAI methods attribute importance to a single prediction, but clinicians, financiers, or engineers instead want to know which features *caused* risk to rise, fall, or remain stable. For instance, a doctor may ask which signals explain a patient's sudden recovery, or an analyst may seek factors driving a credit score drop. Explaining such changes requires attribution methods tailored to highlight features responsible for prediction differences over time.

With regard to this, existing XAI methods explain single-time predictions without historical context, making it hard to tell whether a feature caused risk to rise, fall, or stay stable. The top of Figure 1 shows a sepsis model over three steps $T_1 < T_2 < T_3$: risk rises from 10% at $T_1$ to 90% at $T_2$, then drops to 70% at $T_3$ (b). Clinicians want explanations for the *recovery* between $T_2$ and $T_3$, but standard methods (c) still highlight $T_2$'s features, linking them to high risk. Explaining this 20% reduction instead requires attribution methods explicitly designed for prediction changes (d).

Real-world time series further suffer from irregular sampling and imputation, *e.g.*, by forward filling. As shown in the bottom part of Figure 1, when all values at $T_3$ are imputed (a), standard methods (c) wrongly assign high attribution to them because models emphasize recent inputs, whereas clinicians require attribution for actual observations. Our setup (d) instead *supports time-wise attribution*,

computing $T_2$'s attributions from the change $T_1 \to T_2$ and $T_3$'s from $T_2 \to T_3$, thereby highlighting genuine observations while de-emphasizing imputed ones.

# 4 DELTA-XAI: A FRAMEWORK FOR EXPLAINING PREDICTION CHANGES

In this section, we elaborate on our approach for attributing prediction changes in online time series monitoring. To achieve this, we first introduce a prediction difference wrapper that seamlessly adapts existing single-prediction time series explainers to explain prediction changes; we also highlight the special cases where the wrapper function simplifies to differences of attributions due to linearity properties (Section 4.1). Second, motivated by empirical findings that classical Integrated Gradients (IG) outperform recent alternatives, we propose Shifted Window Integrated Gradients (SWING), specifically designed to explain prediction differences, incorporating historical contexts and thereby addressing the limitations of IG in explaining prediction changes for online time series (Section 4.2).

## 4.1 FROM STATIC TO DYNAMIC: EXTENDING XAI TO PREDICTION CHANGES

Given an online time series classifier $f : \mathbb{R}^{W \times D} \to [0, 1]^C$, our goal is to attribute prediction change $f(\mathbf{X}_{T_2-W+1:T_2})_{\hat{c}} - f(\mathbf{X}_{T_1-W+1:T_1})_{\hat{c}}$ with $\hat{c} = \arg\max_c f(\mathbf{X}_{T_2-W+1:T_2})_c - f(\mathbf{X}_{T_1-W+1:T_1})_c$ between two time steps $T_1 < T_2$. However, in general, it is impossible to explain prediction differences using *isolated attributions from individual time steps*, since they are irrelevant to such change. Indeed, neither computing attributions on differenced inputs (as $f$ is generally nonlinear), nor subtracting attributions across outputs (as attribution algorithms are also nonlinear), provides valid explanations, often yielding implausible results; for example, Table 5 shows that Dynamask (Crabbé & Van Der Schaar, 2021) produces misleading explanations when subtracting attributions across two time steps. To address this, we devise a prediction difference wrapper $g$:

$$g : \mathbb{R}^{(T_2-T_1+W) \times D} \to [0, 1]^C, \; g(\mathbf{X}_{T_1-W+1:T_2}) := f(\mathbf{X}_{T_2-W+1:T_2}) - f(\mathbf{X}_{T_1-W+1:T_1}), \quad (1)$$

which allows any single-prediction XAI method $\varphi$ to be directly applied to explain prediction changes:

$$\varphi(f, \mathbf{X}_{t,d} \mid T_1 \to T_2) = \varphi(g, \mathbf{X}_{t,d} \mid T_2), \; \forall t \in \{T_1 - W + 1, \dots, T_2\}, d \in \{1, \dots, D\}. \quad (2)$$

This wrapper reformulates attribution of prediction differences as a single-prediction explanation, making it broadly applicable across existing methods. It further enables single-time XAI approaches to be adapted with little or no modification, as detailed in Section D.

**Special case: linear and complete XAI methods.** While our wrapper function $g$ is broadly applicable, it requires recomputing attributions for each time pair $(T_1, T_2)$. When $\varphi$ satisfies *linearity* in the attribution space, prediction changes reduce to the difference of single-time attributions, allowing those from one step to be reused across any pair that includes it, without computing them through $g$:

$$\varphi(f, \mathbf{X}_{t,d} \mid T_1 \to T_2) = \varphi(f, \mathbf{X}_{t,d} \mid T_2) - \varphi(f, \mathbf{X}_{t,d} \mid T_1). \quad (3)$$

If $\varphi$ also satisfies *completeness* in the attribution space, *e.g.*, SHAP variants (Lundberg & Lee, 2017), IG (Sundararajan et al., 2017), DeepLIFT (Shrikumar et al., 2017), this formulation guarantees *online completeness*, ensuring that summed attributions exactly match the prediction change:

**Theorem 1** (Attribution Decomposition Theorem for Online Completeness). *Given a linear and complete attribution method $\varphi$ with a* fixed *baseline, the following decomposition holds:*

$$f(\boldsymbol{X}_{T_2-W+1:T_2})_{\hat{c}} - f(\boldsymbol{X}_{T_1-W+1:T_1})_{\hat{c}} = \underbrace{\sum_{t=T_1+1}^{T_2} \sum_{d=1}^{D} \varphi(f, \boldsymbol{X}_{t,d} \mid T_2)}_{\textit{Addition of newest features}}$$

$$+ \underbrace{\sum_{t=T_2-W+1}^{T_1} \sum_{d=1}^{D} [\varphi(f, \boldsymbol{X}_{t,d} \mid T_2) - \varphi(f, \boldsymbol{X}_{t,d} \mid T_1)]}_{\textit{Delayed effect of intermediate features}} - \underbrace{\sum_{t=T_1-W+1}^{T_2-W} \sum_{d=1}^{D} \varphi(f, \boldsymbol{X}_{t,d} \mid T_1)}_{\textit{Removal of oldest features}}. \quad (4)$$

This theorem indicates that summed attributions across all features and time points exactly equal the prediction change, providing clear interpretability. The proof is in Section F.

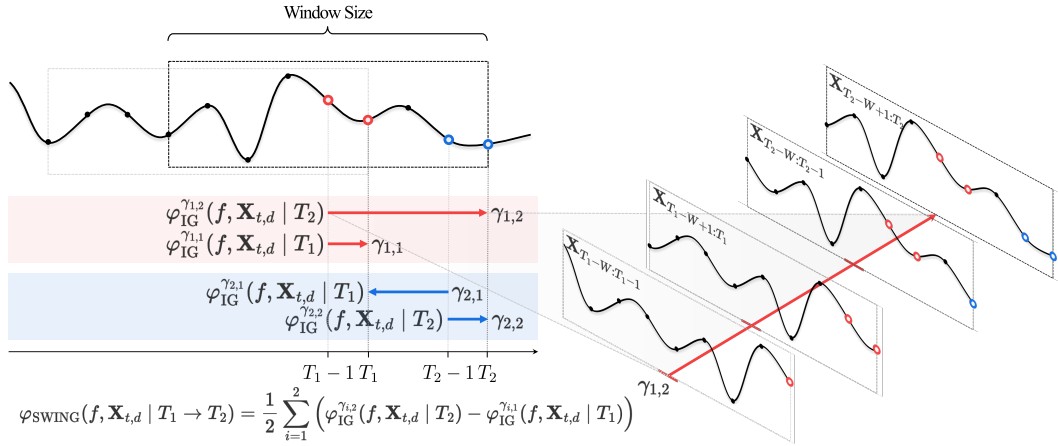

**Figure 2:** Overview of the proposed SWING framework for explaining prediction changes in online patient monitoring. SWING extends conventional Integrated Gradients (IG) by replacing zero-baseline straight paths with line integrals over shifted historical windows and piecewise-linear paths, capturing temporal dynamics and mitigating out-of-distribution effects.

## 4.2 SWING: SHIFTED WINDOW INTEGRATED GRADIENTS

Standard Integrated Gradients (IG) remains competitive for explaining prediction changes but suffers from OOD artifacts and ignores temporal dynamics. We propose SWING, extending IG with i) retrospective baseline selection (RBS), ii) dual-path integration (DPI) ensuring online completeness, and iii) piecewise-linear historical integration (PHI), yielding reliable explanations while preserving key theoretical properties. The overall pipeline is shown in Figure 2, with the detailed procedure outlined in Algorithm 1.

**Extending IG to line integrals over parameterized curves.** We generalize IG as a line integral over a parameterized curve $\gamma : [0,1] \to \mathbb{R}^{W \times D}$ connecting a baseline $\mathbf{X}'$ to the input $\mathbf{X}_{T-W+1:T}$:

$$\varphi_{\text{IG}}^{\gamma}(f, \mathbf{X}_{t,d} \mid T) = \int_0^1 \frac{\partial f(\gamma(\alpha))_{\hat{c}}}{\partial \mathbf{X}_{t,d}} \frac{\partial \gamma_{t,d}(\alpha)}{\partial \alpha} d\alpha. \tag{5}$$

Here, standard IG appears as the special case of a straight-line path $\gamma(\alpha) = (1-\alpha)\mathbf{X}' + \alpha\mathbf{X}_{T-W+1:T}$, where $\partial\gamma_{t,d}(\alpha)/\partial\alpha = \mathbf{X}_{t,d} - \mathbf{X}'_{t,d}$, yielding $\varphi_{\text{IG}}^{\gamma}(f, \mathbf{X}_{t,d} \mid T) = (\mathbf{X}_{t,d} - \mathbf{X}'_{t,d}) \int_0^1 \frac{\partial f(\mathbf{X}' + \alpha(\mathbf{X} - \mathbf{X}'))_{\hat{c}}}{\partial \mathbf{X}_{t,d}} d\alpha$.

**Retrospective baseline selection.** Motivated by the intuition that the most realistic baseline is the *recent past observation*, we generalize the baseline to the window $d$ steps before the input. For an input window $\mathbf{X}_{T-W+1:T}$, this generalized baseline is $\mathbf{X}_{T-W+1-d:T-d}$. In practice, we set $d = 1$ as the default, since the immediate past provides the most stable and realistic reference. Formally, with $d = 1$, we denote by $\gamma_i : [0,1] \to \mathbb{R}^{W \times D}$ the straight-line path from the baseline $\mathbf{X}_{T_i-W:T_i-1}$ to the input $\mathbf{X}_{T_i-W+1:T_i}$, parameterized as: $\gamma_i(\alpha) = (1-\alpha)\mathbf{X}_{T_i-W:T_i-1} + \alpha\mathbf{X}_{T_i-W+1:T_i}$, $\alpha \in [0,1]$. This keeps $\gamma_i(\alpha)$ near the data manifold and mitigates OOD issues, yielding:

$$\varphi_{\text{RBS}}(f, \mathbf{X}_{t,d} \mid T_1 \to T_2) = \varphi_{\text{IG}}^{\gamma_2}(f, \mathbf{X}_{t,d} \mid T_2) - \varphi_{\text{IG}}^{\gamma_1}(f, \mathbf{X}_{t,d} \mid T_1). \tag{6}$$

**Dual-path integration.** Since $\varphi_{\text{RBS}}$ uses distinct baselines at $T_1$ and $T_2$, relying on a single path may lead to incomplete explanations. Thus, we define $\tilde{\gamma}_{i,j} : [0,1] \to \mathbb{R}^{W \times D}$ as the straight-line path: $\tilde{\gamma}_{i,j}(\alpha) = (1-\alpha)\mathbf{X}_{T_i-W:T_i-1} + \alpha\mathbf{X}_{T_j-W+1:T_j}$, $\alpha \in [0,1]$, $i,j \in \{1,2\}$. DPI integrates along all four baseline–input pairs $\tilde{\gamma}_{i,j}$ and averages the results:

$$\varphi_{\text{DPI}}(f, \mathbf{X}_{t,d} \mid T_1 \to T_2) = \frac{1}{2}\sum_{i=1}^{2}\left(\varphi_{\text{IG}}^{\tilde{\gamma}_{i,2}}(f, \mathbf{X}_{t,d} \mid T_2) - \varphi_{\text{IG}}^{\tilde{\gamma}_{i,1}}(f, \mathbf{X}_{t,d} \mid T_1)\right). \tag{7}$$

This symmetric construction balances both baselines and, more importantly, ensures *online completeness* in Theorem 1, so that summed attributions equal the prediction change.

**Piecewise-linear historical integration.**   When the temporal gap between the baseline at $T_i - 1$ and the target at $T_j$ is large, directly interpolating between $X_{T_i-W:T_i-1}$ and $X_{T_j-W+1:T_j}$ may traverse regions far from the data manifold, leading to unstable attributions. To prevent such off-manifold transitions, we introduce a piecewise-linear sliding path $\gamma_{i,j}(\alpha)$ that incrementally shifts the window across intermediate historical time points, ensuring a smooth and temporally consistent progression from baseline to target. We describe the case $i < j$; the case $i > j$ is obtained by reversing the temporal order. Let $M = T_j - T_i + 1$ denote the number of window transitions between $T_i - 1$ and $T_j$, and for $\alpha \in [0, 1]$ set:

$$K = \underbrace{\lfloor \alpha M \rfloor}_{\text{segment index}} , \quad \tilde{\alpha} = \underbrace{\alpha M - K}_{\text{local interpolation ratio}} . \tag{8}$$

Then, the integration path is defined as $\gamma_{i,j}(\alpha) = (1-\tilde{\alpha})\mathbf{X}_{T_i+K-W:T_i+K-1} + \tilde{\alpha}\mathbf{X}_{T_i+K-W+1:T_i+K}$. As $\alpha$ increases from 0 to 1, the interpolation shifts smoothly from the baseline window at $T_i - 1$ to the target window at $T_j$ in $M$ small linear steps, ensuring that the trajectory remains close to the temporal data manifold rather than jumping directly across distant windows. This sliding construction yields more stable and faithful attributions for long-range temporal changes. SWING then aggregates contributions over all baseline–target combinations, producing a temporally consistent explanation while preserving online completeness:

$$\varphi_{\text{SWING}}(f, \mathbf{X}_{t,d} \mid T_1 \rightarrow T_2) = \frac{1}{2} \sum_{i=1}^{2} \left( \varphi_{\text{IG}}^{\gamma_{i,2}}(f, \mathbf{X}_{t,d} \mid T_2) + \varphi_{\text{IG}}^{\gamma_{i,1}}(f, \mathbf{X}_{t,d} \mid T_1) \right). \tag{9}$$

In practice, we uniformly sample $\alpha$ at $n_{\text{samples}}$ points within $[0, 1]$ to obtain a tractable approximation of the path integral.

**Theoretical properties of SWING.**   SWING extends the axiomatic guarantees of IG—Online Completeness, Implementation Invariance, and Skew-Symmetry—providing principled interpretations of online prediction changes. The proofs provided in Section F.

**Theorem 2** (Online Completeness). *The sum of SWING attributions equals the prediction difference between two time steps:* $\sum_{t,d} \varphi_{\text{SWING}}(f, \boldsymbol{X}_{t,d} \mid T_1 \rightarrow T_2) = f(\boldsymbol{X}_{T_2-W+1:T_2}) - f(\boldsymbol{X}_{T_1-W+1:T_1})$.

**Theorem 3** (Implementation Invariance). *SWING attributions depend only on the model function, remaining invariant to equivalent network implementations.*

**Theorem 4** (Skew-Symmetry). *For reversed prediction changes, SWING attributions satisfy skew-symmetry:* $\varphi_{\text{SWING}}(f, \boldsymbol{X}_{t,d} \mid T_1 \rightarrow T_2) = -\varphi_{\text{SWING}}(f, \boldsymbol{X}_{t,d} \mid T_2 \rightarrow T_1)$.

## 5    EVALUATION METRICS FOR EXPLAINING PREDICTION CHANGES

This section introduces new metrics for online time series monitoring. We first examine issues with zero and average imputation (Section 5.1), then propose metrics for attribution faithfulness and sufficiency (Section 5.2), and finally extend beyond these to a unified evaluation standard (Section 5.3).

### 5.1    PROBLEM OF EXISTING EVALUATION METRICS

Modern time series XAI methods (Liu et al., 2024b;a) typically assess *faithfulness*—how well attributions reflect model decisions— and *sufficiency*—how well retained features preserve predictions— by substituting removed features with zero or average baselines. These substitutions ignore temporal locality and autocorrelation, yielding out-of-distribution (OOD) samples. Our analysis on MIMIC-III (Johnson et al., 2016) (LSTM backbone, SWING attributions) in

**Table 1:** Substitution analysis for XAI evaluation.

| Substitution | CPD | OOD Score |
|---|---|---|
| Zero | 28.12 | 0.840 |
| Average | 14.85 | 0.222 |
| Forward-Fill | 12.98 | 0.093 |

Table 1 shows that zero/average substitution produces substantial OOD samples, with high Cumulative Prediction Difference (CPD) and large OOD scores measured by conditional generative model MSE, exaggerating prediction differences and distorting correlations. We therefore adopt forward-filling for faithfulness evaluation, as it reduces OOD effects and improves reliability.

## 5.2 Proposed Evaluation Metrics for Faithfulness and Sufficiency

Recently, TIMING (Jang et al., 2025) has identified a key issue in faithfulness and sufficiency evaluation of time series XAI: removing top/bottom salient points *simultaneously* can inflate scores by rewarding mere sign alignment. To mitigate this, TIMING introduced Cumulative Prediction Difference (CPD) and Cumulative Prediction Preservation (CPP), which remove features *sequentially*. In our wrapper setting $g$ with input $\mathbf{X}$:

$$\text{CPD}(g, \mathbf{X}, K) = \sum_{k=0}^{K-1} \left\| g(\mathbf{X}_k^\uparrow) - g(\mathbf{X}_{k+1}^\uparrow) \right\|_1, \ \text{CPP}(g, \mathbf{X}, K) = \sum_{k=0}^{K-1} \left\| g(\mathbf{X}_k^\downarrow) - g(\mathbf{X}_{k+1}^\downarrow) \right\|_1, \quad (10)$$

where $\mathbf{X}_k^\uparrow$ and $\mathbf{X}_k^\downarrow$ denote the inputs obtained by removing the top-$k$ and bottom-$k$ features, respectively, from the full set of $(T_2 - T_1 + W) \times D$ features across the entire time window. These metrics showed that gradient-based methods like IG (Sundararajan et al., 2017) often outperform recent masking-based ones (Liu et al., 2024b; Queen et al., 2024; Liu et al., 2024a), a trend we also find for online prediction changes.

However, these metrics have key limitations: 1) they ignore the relative ranking among top-$k$ features, 2) they overlook evolving attributions across time, and 3) they assess only ranking, not magnitude. To address 1), we introduce area-based metrics, Area Under Prediction Difference/Preservation (AUPD, AUPP). AUPD is defined as the average of CPD values over all prefixes of the top-$k$ features:

$$\text{AUPD}(g, \mathbf{X}, K) = \frac{1}{2K} \sum_{k=1}^{K} \Big( \text{CPD}(g, \mathbf{X}, k) + \text{CPD}(g, \mathbf{X}, k-1) \Big). \quad (11)$$

AUPP is defined analogously using CPP instead of CPD. For 2), we aggregate attributions with a centered sliding window: $\varphi(\mathbf{X}_{t,d} \mid T) = 1/(2W-1) \sum_{T'=t-W+1}^{t+W-1} \varphi(\mathbf{X}_{t,d} \mid T')$. We denote the resulting macro-level metrics as Macro Prediction Difference/Preservation (MPD, MPP) and their area-based variants as Area Under Macro Prediction Difference/Preservation (AUMPD, AUMPP). For 3), we propose Corr., the correlation between ordered attributions and prediction differences:

$$\begin{aligned}
\text{Corr.}(\varphi, \mathbf{X}, K) = \text{Corr.}\bigg( & \left[ \varphi^{(1)}, \ldots, \varphi^{(K)}, \varphi^{(W \times D - K + 1)}, \ldots, \varphi^{(W \times D)} \right], \\
& \left[ \left| g_1^\uparrow - g_0^\uparrow \right|, \ldots, \left| g_K^\uparrow - g_{K-1}^\uparrow \right|, \left| g_1^\downarrow - g_0^\downarrow \right|, \ldots, \left| g_K^\downarrow - g_{K-1}^\downarrow \right| \right] \bigg),
\end{aligned} \quad (12)$$

Together, these metrics capture ranking consistency, temporal dynamics, and attribution magnitudes, providing more faithful and interpretable evaluations.

## 5.3 Beyond Faithfulness and Sufficiency: Broader Evaluation Metrics

Existing time series XAI studies (Tonekaboni et al., 2020; Leung et al., 2023; Crabbé & Van Der Schaar, 2021; Enguehard, 2023; Liu et al., 2024b; Queen et al., 2024; Liu et al., 2024a) mainly assess faithfulness and sufficiency. While crucial—and expanded here with nine detailed metrics—these alone are insufficient for practical utility. We therefore incorporate: 1) Coherence, checking alignment with domain knowledge (case study); and 2) Time/Memory Complexity, measuring real-time feasibility (empirically). Together, these provide a more comprehensive evaluation of XAI methods.

## 6 Experiments

In this section, we comprehensively evaluate SWING against 14 time series XAI baselines within our Delta-XAI framework. We first describe the experimental setup in Section 6.1, and subsequently address the following key research questions through our empirical analysis:

- Does SWING provide more faithful explanations of prediction changes compared to existing time series XAI baselines? (Section 6.2)
- What is the contribution of each component of SWING to overall performance, as revealed through ablation studies? (Section 6.3)

**Table 2:** Performance comparison of XAI methods on clinical prediction tasks: MIMIC-III decompensation benchmark using LSTM as backbone architecture. Evaluation is performed by removing the most or least salient 50 feature points per time step, using forward-fill substitution.

| Algorithm | Removal of Most Salient 50 Points | | | | Removal of Least Salient 50 Points | | | | Corr. ↑ |
|---|---|---|---|---|---|---|---|---|---|
| | CPD ↑ | AUPD ↑ | MPD ↑ | AUMPD ↑ | CPP ↓ | AUPP ↓ | MPP ↓ | AUMPP ↓ | |
| LIME (Ribeiro et al., 2016) | 2.26±0.04 | 1.72±0.03 | 13.78±0.08 | 7.70±0.02 | 32.46±0.16 | 14.26±0.10 | 33.45±0.14 | 15.30±0.10 | 0.02±0.00 |
| GradSHAP (Lundberg & Lee, 2017) | 13.73±0.06 | 9.05±0.04 | 16.68±0.08 | 11.19±0.05 | 32.97±0.13 | 13.96±0.06 | 30.13±0.17 | 11.95±0.09 | 0.14±0.00 |
| IG (Sundararajan et al., 2017) | 13.42±0.06 | 9.10±0.05 | 16.14±0.07 | 11.31±0.04 | 33.55±0.12 | 13.97±0.04 | 29.46±0.17 | 10.85±0.05 | 0.17±0.00 |
| DeepLIFT (Shrikumar et al., 2017) | 13.58±0.06 | 9.42±0.04 | 16.03±0.08 | 11.25±0.06 | 35.96±0.16 | 14.61±0.07 | 31.53±0.15 | 11.41±0.06 | 0.19±0.00 |
| FO (Suresh et al., 2017) | 13.14±0.10 | 9.92±0.07 | 17.79±0.10 | 12.92±0.06 | 44.02±0.19 | 22.32±0.09 | 16.92±0.09 | 6.24±0.03 | 0.26±0.00 |
| AFO (Tonekaboni et al., 2020) | 13.24±0.07 | 9.30±0.05 | 17.16±0.07 | 11.95±0.03 | 36.13±0.22 | 16.64±0.09 | 24.13±0.16 | 9.32±0.07 | 0.28±0.00 |
| FIT (Tonekaboni et al., 2020) | 3.40±0.04 | 2.70±0.03 | 7.11±0.04 | 6.14±0.04 | 35.52±0.11 | 17.55±0.08 | **12.07±0.05** | 10.19±0.05 | 0.06±0.00 |
| WinIT (Leung et al., 2023) | 19.64±0.07 | 12.25±0.04 | **24.87±0.13** | 15.45±0.08 | 29.22±0.07 | 13.05±0.05 | 26.11±0.12 | 11.92±0.06 | 0.21±0.00 |
| Dynamask (Crabbé & Van Der Schaar, 2021) | 11.72±0.08 | 7.56±0.04 | 13.15±0.08 | 8.25±0.04 | 53.07±0.24 | 26.22±0.08 | 49.80±0.16 | 24.26±0.06 | 0.04±0.00 |
| Extrmask (Enguehard, 2023) | 16.66±0.11 | 10.47±0.06 | 17.51±0.12 | 10.63±0.05 | 29.91±0.17 | 15.13±0.09 | 29.64±0.17 | 14.84±0.12 | 0.08±0.00 |
| ContraLSP (Liu et al., 2024b) | 12.88±0.36 | 8.69±0.26 | 18.00±0.16 | 11.11±0.18 | 41.62±0.30 | 21.17±0.10 | 42.67±0.29 | 21.94±0.10 | 0.03±0.00 |
| TimeX (Queen et al., 2024) | 16.99±0.09 | 11.45±0.06 | 19.45±0.10 | 12.45±0.06 | 50.34±0.10 | 24.11±0.04 | 51.06±0.09 | 24.50±0.05 | 0.03±0.00 |
| TimeX++ (Liu et al., 2024a) | 11.12±0.05 | 7.00±0.04 | 13.14±0.02 | 7.76±0.02 | 34.21±0.17 | 13.72±0.07 | 32.34±0.11 | 13.08±0.05 | 0.03±0.00 |
| TIMING (Jang et al., 2025) | 14.99±0.07 | 9.71±0.05 | 16.50±0.08 | 11.53±0.04 | 31.22±0.16 | 13.36±0.05 | 27.19±0.19 | 10.24±0.07 | 0.19±0.00 |
| SWING | **23.87±0.16** | **16.23±0.10** | 22.27±0.19 | **15.52±0.12** | **17.76±0.04** | **5.85±0.04** | 18.20±0.06 | **6.06±0.05** | **0.40±0.00** |

- How does SWING perform qualitatively, and does it produce temporally coherent explanations? (Section 6.4)
- How does SWING behave under diverse evaluation settings, and what are its computational characteristics? (Section 6.5)

## 6.1 EXPERIMENTAL SETUP

**Datasets.** Following prior work (Liu et al., 2024a;b), we use two large-scale clinical datasets commonly adopted for online time series monitoring: MIMIC-III (Johnson et al., 2016) for decompensation prediction and PhysioNet 2019 (Reyna et al., 2020) for early sepsis detection, where predictions are updated as new data arrive. To assess generalizability, we also test on Activity, a human activity recognition dataset (Vidulin et al., 2010), and synthetic benchmarks with controlled temporal dynamics (Delayed Spike (Leung et al., 2023), Switch-Feature (Tonekaboni et al., 2020)). Further details are provided in Section G.

**Model architectures.** We mainly evaluate XAI methods with LSTM architectures, a fundamental choice for time series classification (Tonekaboni et al., 2020; Leung et al., 2023). To show our framework's versatility, we also implement a CNN with stacked convolutions and a Transformer encoder for long-range dependencies.

**XAI baselines.** We comprehensively implement and evaluate all XAI methods in Delta-XAI, including our proposed SWING. Existing methods are categorized as: 1) modality-agnostic perturbation-based (LIME (Ribeiro et al., 2016), FO (Suresh et al., 2017), AFO (Tonekaboni et al., 2020)); 2) gradient-based (IG (Sundararajan et al., 2017), DeepLIFT (Shrikumar et al., 2017), GradSHAP (Lundberg & Lee, 2017)); and 3) time series-specific methods, including online explainers (FIT (Tonekaboni et al., 2020), WinIT (Leung et al., 2023)), masking frameworks (Dynamask (Crabbé & Van Der Schaar, 2021), ExtrMask (Enguehard, 2023), ContraLSP (Liu et al., 2024b), TimeX (Queen et al., 2024), TimeX++ (Liu et al., 2024a)), and TIMING (Jang et al., 2025), augmenting IG with random masking.

**Implementation details.** Our method uses a single hyperparameter ($n_{\text{samples}} = 50$). We set $T_2 - T_1 = 1$ for adjacent-step explanations, remove $K = 50$ points, and absolutize directional attributions for fairness. Explanations are obtained through the wrapper $g$, which highlights features driving the $T_1 \rightarrow T_2$ change. Metrics are scaled by $10^3$ except for correlations, memory, and time. Results are reported as mean±standard error over five runs, with best and second-best marked in **bold** and underline. More details are at `https://github.com/AITRICS/Delta-XAI`.

## 6.2 RESULTS ON ATTRIBUTION FAITHFULNESS AND SUFFICIENCY

**Main results.** Tables 2 and 6 shows that, under the Delta-XAI protocol, SWING achieves the best performance on most metrics across both clinical datasets (MIMIC-III and PhysioNet 2019). Other gradient-based explainers such as IG, DeepLIFT, and TIMING also perform consistently well. In

**Table 3:** Ablation study of SWING examining retrospective baseline selection (RBS), dual-path integration (DPI), and piecewise-linear historical integration (PHI) on the MIMIC-III decompensation benchmark (Johnson et al., 2016), with LSTM (Hochreiter & Schmidhuber, 1997) backbone and interval $T_2 - T_1 = 24$. We vary the baseline distance $d$ (default: 1) and remove the most or least salient 50 feature points per time step, using forward-fill substitution.

| Algorithm | Removal of Most Salient 50 Points | | | | Removal of Least Salient 50 Points | | | | Corr. ↑ |
|---|---|---|---|---|---|---|---|---|---|
| | CPD ↑ | AUPD ↑ | MPD ↑ | AUMPD ↑ | CPP ↓ | AUPP ↓ | MPP ↓ | AUMPP ↓ | |
| w/o RBS, PHI | 45.82±0.20 | 26.85±0.13 | 40.85±0.22 | 26.31±0.14 | 104.05±0.34 | 48.43±0.19 | 77.94±0.30 | 29.56±0.11 | 0.20±0.00 |
| w/o RBS | 40.25±0.22 | 24.39±0.12 | 46.71±0.35 | 29.10±0.21 | 79.70±0.30 | 31.29±0.07 | 82.59±0.35 | 33.33±0.11 | 0.16±0.00 |
| w/o DPI ($\gamma_{1,1}, \gamma_{2,2}$) | **55.01**±**0.35** | **32.39**±**0.21** | **53.09**±**0.38** | **33.08**±**0.21** | 71.01±0.30 | 26.12±0.10 | 76.26±0.34 | 29.29±0.09 | 0.19±0.00 |
| $d = 0$ | 33.80±0.23 | 20.11±0.13 | 48.98±0.44 | 29.78±0.24 | 82.28±0.29 | 33.99±0.10 | 83.69±0.35 | 33.31±0.12 | 0.11±0.00 |
| $d = 3$ | 42.00±0.33 | 26.00±0.28 | 47.49±0.44 | 29.74±0.34 | 68.53±0.49 | 25.60±0.18 | 71.88±0.62 | 27.54±0.27 | 0.19±0.00 |
| $d = 5$ | 41.98±0.31 | 25.65±0.25 | 47.18±0.45 | 29.44±0.33 | 72.31±0.66 | 27.88±0.22 | 74.62±0.71 | 29.11±0.26 | 0.18±0.00 |
| $d = 10$ | 41.02±0.28 | 24.94±0.18 | 46.89±0.39 | 29.23±0.25 | 76.55±0.48 | 29.96±0.16 | 78.11±0.44 | 31.04±0.16 | 0.17±0.00 |
| SWING | 41.07±0.22 | 26.46±0.18 | 50.58±0.28 | 32.29±0.20 | **60.29**±**0.14** | **21.60**±**0.08** | **64.43**±**0.19** | **23.87**±**0.10** | **0.21**±**0.00** |

contrast, surrogate-driven methods like LIME, TimeX, and TimeX++ exhibit lower performance on preservation metrics, likely due to the data- and hyperparameter-sensitivity of their surrogate models. These findings underscore SWING's dominance and refine the performance hierarchy of XAI techniques within Delta-XAI.

**Diverse synthetic and real-world benchmarks.**    On the real-world Activity dataset (Table 7), SWING achieves the highest scores across all metrics. On the synthetic State and Switch-Feature benchmarks, it performs best or second-best on most metrics, with competitive results elsewhere, underscoring its robustness across both practical and controlled scenarios.

**Different backbone architectures.**    To assess generalizability across architectures, we re-evaluate the five strongest clinical baselines and SWING on the MIMIC-III dataset using CNN and Transformer backbones in Table 8. SWING outperforms competing methods across most of the metrics; these findings confirm that SWING maintains robust faithfulness across diverse backbones within Delta-XAI.

**Larger time differences.**    To assess robustness over longer time gaps, we compare SWING with five strong baselines on MIMIC-III at $T_2 - T_1 = 6$ and 24 (Table 9), with WinIT reported only for 6 due to generator limits. SWING achieves best or near-best scores on most metrics, with particularly dominant gains on preservation metrics, though performance gaps narrow for longer intervals as CPD and AUPD converge across methods.

## 6.3    ABLATION STUDY

This subsection examines the contributions of SWING components—retrospective baseline selection (RBS), piecewise-linear historical integration (PHI), and dual-path integration (DPI). The meaning of each ablation configuration and its corresponding mathematical formulation are minutely detailed in Section H. As shown in Table 3, removing both RBS and PHI causes substantial degradation, while individually removing either module also reduces performance, highlighting their complementary roles. Without DPI, the model attains the best faithfulness scores (CPD, AUPD, MPD, AUMPD) but falls behind SWING in preservation metrics, suggesting that DPI primarily stabilizes preservation. Overall, SWING sacrifices a small margin in prediction-difference metrics to achieve strong preservation and correlation performance. Varying the baseline offset $d$ further shows that using the immediate past window ($d = 1$) yields the best trade-off, as both too short ($d = 0$) and longer offsets ($d \geq 3$) degrade results. Finally, SWING maintains stable performance across varying $n_{\text{samples}}$ (10 to 100), demonstrating robustness to hyperparameter changes (Figure 4).

## 6.4    QUALITATIVE ANALYSIS

**Case study on feature attributions.**    Beyond quantitative evaluation, we qualitatively assess XAI methods within the Delta-XAI framework. Figures 5 to 9 show raw input trajectories with attribution heatmaps for fifteen baselines at $T_2 - T_1 = 1$. FO, TimeX, and TimeX++ tend to spread attributions broadly across the time axis, while Dynamask and TimeX++ align closely with input fluctuations,

reflecting their strong benchmark scores. In contrast, SWING yields sharper, localized attributions that emphasize recent time steps most responsible for prediction changes.

**Coherence analysis.** We further assess whether SWING aligns with clinical knowledge by inspecting attributions on a representative MIMIC-III case in Figure 10. A sharp SBP drop at the last time step increases risk (a, b), consistent with evidence linking hypotensive episodes to decompensation (Toki et al., 2025). In contrast, an $SpO_2$ rise lowers risk (c), reflecting the stabilizing effect of improved oxygenation (Semler et al., 2022), while a steep fall in blood pH raises risk (d), in line with studies associating acidemia with poor outcomes (Henrique et al., 2023). To maximize visibility, we further closely examine how the features at the last time step influence the model's predictions in Figure 11. A sharp blood pressure drop at the last time step increases risk (a), consistent with evidence linking hypotensive episodes to decompensation (Toki et al., 2025). In contrast, an $SpO_2$ rise lowers risk (b), while an $SpO_2$ drop increases risk (c), reflecting the destabilizing effect of oxygen desaturation (Semler et al., 2022). A rise in GCS lowers risk (d), consistent with evidence linking improved consciousness to better outcomes (Marincowitz et al., 2018). These examples demonstrate that SWING produces attribution patterns consistent with established clinical findings.

## 6.5 FURTHER ANALYSIS

**In-depth analysis under diverse settings.** To further examine SWING's behavior, we perform subgroup analyses by splitting cases according to whether predictions or class labels change (Tables 10 and 11), where it consistently preserves explanatory advantages. We also test different temporal resolutions with 24- and 72-length windows (Table 12), confirming that SWING yields stable attributions and clear gains over baselines across both horizons. These results demonstrate that SWING provides reliable explanations under diverse prediction dynamics and temporal contexts, reinforcing its applicability to real-world clinical monitoring.

**Efficiency analysis: runtime and memory.** We assess SWING's efficiency along two axes—runtime and memory footprint. As shown in Figure 3, SWING attains the highest AUPD while requiring only 0.35s per sample, comparable to other gradient-based explainers (DeepLIFT 0.11s, GradSHAP 0.18s). For memory, it consumes 448 MB per sample, identical to IG (448 MB) and close to GradSHAP and DeepLIFT (438 MB). These results demonstrate that SWING achieves state-of-the-art explanatory quality without incurring additional computational or memory costs.

## 7 CONCLUSION

In this paper, we have introduced the task of explaining prediction changes in online time series monitoring and proposed Delta-XAI, a unified framework that integrates 14 XAI methods with a dedicated evaluation suite for temporal dynamics. Through extensive experiments, we demonstrated that, when adapted, classical gradient-based methods, such as Integrated Gradients (IG), remain strong baselines. Motivated by this, we developed SWING, an extension of IG that robustly captures temporal feature evolution. We believe our contributions significantly advance XAI for online time series by shifting the paradigm from static interpretations toward dynamic, context-sensitive explanations—an essential step toward trustworthy AI in time-critical domains.

## ETHICS STATEMENT

This work develops explainable AI (XAI) methods for online time series monitoring in domains such as healthcare and finance. We use only publicly available open-source benchmark datasets (*e.g.*, MIMIC-III, PhysioNet 2019, Activity), adhering to their usage protocols and ethical standards, and do not collect new human subject data. Our contributions are methodological, aiming to enhance the transparency and interpretability of time series models. Potential misuse may arise if explanations are taken as direct clinical or financial advice; therefore, we emphasize that outputs should be interpreted by domain experts.

## REPRODUCIBILITY STATEMENT

We provide an anonymized implementation of Delta-XAI and SWING at the anonymous repository link `https://github.com/AITRICS/Delta-XAI`. All experimental details—including dataset preprocessing, model architectures, training protocols, hyperparameter settings, and evaluation metrics—are provided in Sections 6 and G, while theoretical proofs and pre-processing steps for MIMIC-III, PhysioNet 2019, Activity, and synthetic benchmarks are elaborated in Section G. In addition, several ablation studies and robustness analyses over multiple iterations further validate the stability of results. All procedures are publicly available and fully reproducible, enabling independent researchers to reproduce and verify our findings.

## ACKNOWLEDGEMENTS

This work was supported by AITRICS and by the Institute for Information & Communications Technology Planning & Evaluation (IITP) grant funded by the Korean government (MSIT) (No. 2019-0-00075, Artificial Intelligence Graduate School Program, KAIST).

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

## A    LIMITATIONS AND BROADER IMPACTS

**Limitations.**    Our proposed approach has few limitations. First, while SWING extends IG by incorporating historical points with shifted window paths, it introduces additional computational overhead compared to simpler methods like standard IG. Second, while our prediction wrapper function can seamlessly incorporate most existing single-time XAI algorithms, a few require minor adjustments—particularly those that rely on probability outputs or internal model representations. The specifics of such adaptations are tangential to our main contribution and are left to future algorithm designers.

**Broader impacts.**    Our work significantly advances explainability in time-critical domains by enabling a nuanced explanation of temporal prediction changes. By providing insights into why and how model predictions evolve, our framework supports decision-making in critical areas such as healthcare, finance, and transportation, potentially improving outcomes through enhanced transparency and accountability. While responsible interpretation and privacy considerations remain important, the benefits of improved trustworthiness and actionable insights in high-stakes environments significantly outweigh these concerns.

## B    LLM USAGE DISCLOSURE

In drafting this manuscript, we made limited use of large language models (LLMs) for minor writing improvements, such as grammar polishing and readability enhancement. The LLMs were not used for research conception, experimental design, analysis, or generation of substantive content. Its role was strictly limited to language editing, and all scientific contributions are attributable solely to the authors.

## C    RELATED WORK

**Modality-agnostic explainable artificial intelligence.**    Although deep neural networks have achieved impressive results across domains like vision (He et al., 2016; Dosovitskiy et al., 2021), language (Vaswani et al., 2017; Brown et al., 2020), and time series (Gamboa, 2017), they often act as black boxes, limiting transparency and accountability—especially in high-stakes areas such as healthcare (Christoph, 2020). To address this, various modality-agnostic XAI methods have been developed. Popular approaches such as LIME (Ribeiro et al., 2016) and SHAP (Shapley, 1953; Lundberg & Lee, 2017) attribute predictions to input features by estimating their contribution strength and direction. Variants like KernelSHAP, GradientSHAP, and DeepSHAP (Kokhlikyan et al., 2020) expand their applicability. Gradient-based methods, including Integrated Gradients (IG) (Sundararajan et al., 2017) and DeepLIFT (Shrikumar et al., 2017), compute attributions using model gradients. Perturbation-based methods like Feature Occlusion (FO) (Suresh et al., 2017) and Augmented Feature Occlusion (AFO) (Tonekaboni et al., 2020) measure feature importance by replacing inputs and observing prediction changes. While these methods have enhanced model explainability, most have been evaluated in vision tasks (Das & Rad, 2020). Their application to time series—particularly for explaining prediction changes in online settings—remains limited, despite the importance of capturing temporal dependencies for meaningful explanations.

**Explainable artificial intelligence for time series.**    XAI for time series presents unique challenges due to temporal dependencies, where the order and historical context of observations significantly affect model behavior. Standard modality-agnostic XAI methods, which often assume independently distributed inputs, fail to capture such dynamics. To address this, a number of time series-specific attribution methods have been proposed (Bento et al., 2021; Tonekaboni et al., 2020; Leung et al., 2023; Crabbé & Van Der Schaar, 2021; Enguehard, 2023; Liu et al., 2024b; Queen et al., 2024; Liu et al., 2024a; Kim et al., 2025). More recent methods have improved temporal modeling through dynamic masking (Crabbé & Van Der Schaar, 2021; Enguehard, 2023), contrastive learning (Liu et al., 2024b), and interpretable surrogate modeling (Queen et al., 2024; Liu et al., 2024a), with TimeX++ incorporating an information bottleneck to mitigate trivial explanations. TIMING (Jang et al., 2025) introduces novel evaluation metrics and enhances IG with random masking to improve sensitivity to temporal variation. Despite recent progress, existing methods fall short in explaining

prediction changes in online time series, lacking contextual insight and temporal evaluation. Our Delta-XAI addresses these gaps by attributing prediction changes directly and introducing metrics aligned with sequential dynamics.

**Explainable artificial intelligence for online time series monitoring.** Among time series XAI methods, FIT (Tonekaboni et al., 2020) and WinIT (Leung et al., 2023) are particularly relevant to online prediction tasks. FIT estimates feature importance by comparing predictive distributions under observed and counterfactual inputs using KL divergence, whereas WinIT models delayed effects by assessing how past observations influence future predictions. However, our proposed framework significantly extends beyond these methods by offering a comprehensive and unified approach. Unlike FIT, which quantifies feature importance solely based on predictive distribution changes at consecutive time points, and WinIT, which evaluates feature relevance within fixed observation windows—both producing static attributions across the entire series—our Delta-XAI explicitly explains prediction changes between distinct time points. Specifically, we attribute changes in model predictions directly rather than attributing individual predictions independently, thus generating dynamic and prediction-time-specific attributions that accurately capture temporal evolution in feature importance. Additionally, we introduce SWING, an advanced attribution method demonstrating superior empirical performance and fulfilling essential theoretical properties, including linearity, completeness, and directional attribution. Finally, our framework systematically integrates existing attribution methods and proposes specialized evaluation metrics tailored explicitly to assessing prediction change explanations in online time series monitoring.

## D  ADAPTING EXISTING XAI ALGORITHMS

In this section, we provide a detailed description of how we adapt existing XAI baselines to our setting. Specifically, Section D.1 presents the algorithms that can be applied without modification, *i.e.*, with seamless integration into our prediction-difference framework, while Section D.2 describes those that require minimal adjustments. In both cases, the adaptation is realized through a wrapper function $g$, which standardizes the attribution process for time series inputs. Notably, this wrapper neither necessitates additional post-training procedures nor alters the underlying models, ensuring fair and consistent comparison across baselines.

### D.1  NO MODIFICATION

The following attribution algorithms operate without modification: LIME (Ribeiro et al., 2016), IG (Sundararajan et al., 2017), DeepLIFT (Shrikumar et al., 2017), FO (Suresh et al., 2017), AFO (Tonekaboni et al., 2020), WinIT (Leung et al., 2023), Dynamask (Crabbé & Van Der Schaar, 2021), Extrmask (Enguehard, 2023), ContraLSP (Liu et al., 2024b), and TIMING (Jang et al., 2025). In all of these cases, the only adjustment involves the use of the prediction-difference wrapper $g$ instead of the single-time prediction model $f$. This wrapper applies uniformly across baselines without altering their internal mechanisms, so the algorithms remain unmodified in our framework.

### D.2  MINIMAL MODIFICATION

The following attribution algorithms require minimal modification: GradSHAP (Lundberg & Lee, 2017), FIT (Tonekaboni et al., 2020), TimeX (Queen et al., 2024), and TimeX++ (Liu et al., 2024a).

**GradSHAP.** Applying GradSHAP (Lundberg & Lee, 2017) directly to the wrapper $g$ violates completeness, since the baseline input for $g$ depends on whether the evaluation is at $T_1$ or $T_2$. For example, if $T_1$ corresponds to a window $\mathbf{X}_{T_1-W+1:T_1}$ and $T_2$ to $\mathbf{X}_{T_2-W+1:T_2}$, then the baseline for $g$ differs depending on which window is active, leading to inconsistency. To address this, we compute GradSHAP directly on $f$, perform two runs with a shared baseline, and subtract the resulting attributions—mirroring the construction of SWING. This procedure preserves completeness while remaining consistent with the prediction-difference formulation.

**FIT.** For FIT (Tonekaboni et al., 2020), we bypass the wrapper $g$ and operate directly on the base model $f$. We construct both the current and previous input windows, $\mathbf{X}_{T_2-W+1:T_2}$ and $\mathbf{X}_{T_1-W+1:T_1}$,

and compute the corresponding class-specific outputs $f(\mathbf{X}_{T_2-W+1:T_2})_{\hat{c}}$ and $f(\mathbf{X}_{T_1-W+1:T_1})_{\hat{c}}$, where $\hat{c}$ is defined as in the main text. At each time step and feature dimension, we sample candidate imputations from the generator, apply them to $\mathbf{X}_{T_2-W+1:T_2}$, and measure their effect on $f(\mathbf{X}_{T_2-W+1:T_2})_{\hat{c}}$. We then quantify the divergence between the perturbed prediction and the original prediction difference $f(\mathbf{X}_{T_2-W+1:T_2})_{\hat{c}} - f(\mathbf{X}_{T_1-W+1:T_1})_{\hat{c}}$. For the divergence metric, we follow the original implementation and support both the KL divergence and the mean absolute deviation. This adaptation preserves FIT's perturbation mechanism while aligning it with attribution of prediction differences.

**TimeX.** For TimeX (Queen et al., 2024), the prediction-difference wrapper $g$ is implemented as a Python class, where the `forward()` method returns the prediction difference $f(\mathbf{X}_{T_2-W+1:T_2}) - f(\mathbf{X}_{T_1-W+1:T_1})$. To support the Model Behavior Consistency (MBC) loss, the class defines an auxiliary method that returns the difference between latent embeddings at the two time points. Specifically, while TimeX formulates the MBC loss $\mathcal{L}_{\text{MBC}}$ as:

$$\mathcal{L}_{\text{MBC}}(Z, Z^E) = \frac{1}{N^2} \sum_{i,j} \left[ D_Z(z_i, z_j) - D_{Z^E}(z_i^E, z_j^E) \right]^2, \tag{13}$$

we obtain latent embeddings using the encoder of the model $f$ with $f = \text{dec} \circ \text{enc}$. For an input sequence $\mathbf{X}_{T_1-W+1:T_2}$, we split it into two windows $\mathbf{X}_{T_1-W+1:T_1}$ and $\mathbf{X}_{T_2-W+1:T_2}$, then compute embeddings as:

$$z = \text{enc}(\mathbf{X}_{T_2-W+1:T_2}) - \text{enc}(\mathbf{X}_{T_1-W+1:T_1}). \tag{14}$$

We also experiment with concatenated embeddings $z = [\text{enc}(\mathbf{X}_{T_2-W+1:T_2}), \text{enc}(\mathbf{X}_{T_1-W+1:T_1})]$ and observe negligible differences in performance. In either case, this embedding construction provides a meaningful space over which the surrogate model operates, while remaining consistent with the prediction-difference wrapper $g$.

**TimeX++.** For TimeX++ (Liu et al., 2024a), we adopt the same strategy: embeddings are obtained in the same manner as in TimeX, and the surrogate model operates on the difference $z_{T_2} - z_{T_1}$. We further modify the label consistency loss. Instead of the cross-entropy formulation in the original implementation, we employ mean squared error (MSE) loss for stability. Here, $\mathbf{X}$ denotes the original input sequence and $\tilde{\mathbf{X}}$ denotes the perturbed version of the input produced by the explanation method. Concretely, the label consistency objective $\mathcal{L}_{\text{LC}}$ is defined as:

$$\mathcal{L}_{\text{LC}}(f(\mathbf{X}), f(\tilde{\mathbf{X}})) = \mathbb{E}\left[ D_{\text{JS}}(f(\mathbf{X}) \| f(\tilde{\mathbf{X}})) \right], \tag{15}$$

following the Jensen–Shannon divergence form in (Queen et al., 2024), but realized with an MSE surrogate. This yields a loss function better aligned with the prediction-difference framework.

# E  ALGORITHM

We provide the detailed procedure of SWING in Algorithm 1, which computes attribution scores for prediction changes in online time series monitoring. The algorithm explicitly defines integration paths using historically observed shifted windows, computes gradients along piecewise-linear paths via interpolation, and averages attributions from dual integration paths. This approach ensures compliance with the completeness property, provides realistic integration trajectories, and mitigates out-of-distribution (OOD) issues.

# F  PROOFS

## F.1  PROOF OF THEOREM 1

By completeness of $\varphi$, for any baseline $\mathbf{X}' \in \mathbb{R}^{W \times D}$, we have:

$$f(\mathbf{X}_{T_2-W+1:T_2})_{\hat{c}} - f(\mathbf{X}')_{\hat{c}} = \sum_{t=T_2-W+1}^{T_2} \sum_{d=1}^{D} \varphi(f, \mathbf{X}_{t,d} \mid T_2), \tag{16}$$

and similarly,

$$f(\mathbf{X}_{T_1-W+1:T_1})_{\hat{c}} - f(\mathbf{X}')_{\hat{c}} = \sum_{t=T_1-W+1}^{T_1} \sum_{d=1}^{D} \varphi(f, \mathbf{X}_{t,d} \mid T_1). \tag{17}$$

---

**Algorithm 1** SWING: Shifted Window Integrated Gradients

---

1: **Input:** Model $f$, inputs $\mathbf{X}_{T_1-W+1:T_1}$, $\mathbf{X}_{T_2-W+1:T_2}$, discretization steps $n_{\text{samples}}$.
2: **Output:** Attributions $\varphi_{\text{SWING}}(f, \mathbf{X}_{t,d} \mid T_1 \to T_2)$ for all $t \in \{T_1 - W + 1, \ldots, T_2\}$, $d \in \{1, \ldots, D\}$.
3: **for** each $\gamma_{i,j}$ with $i, j \in \{1, 2\}$ **do**
4:      $M \leftarrow |T_j - (T_i - 1)|, \quad \sigma \leftarrow \text{sign}(T_j - (T_i - 1))$
5:      $\mathbf{X}^{(0)} \leftarrow \mathbf{X}_{T_i-W:T_i-1}$
6:      $G^{(0)} \leftarrow \partial f(\mathbf{X}^{(0)})/\partial \mathbf{X}^{(0)}$
7:      $\varphi_{\text{IG}}^{\gamma_{i,j}} \leftarrow \mathbf{0}_{(T_2-T_1+W) \times D}$
8:      **for** $m = 1$ **to** $n_{\text{samples}}$ **do**                         ▷ Parallelized in our implementation
9:          $\alpha_m \leftarrow m/n_{\text{samples}}$
10:          $K \leftarrow \underbrace{\min\left(\lfloor \alpha_m M \rfloor, M-1\right)}_{\text{segment index}} \qquad \tilde{\alpha} \leftarrow \underbrace{\alpha_m M - K}_{\text{local interpolation ratio}}$
11:          $s \leftarrow (T_i - 1) + \sigma K$
12:          $\mathbf{X}^{(m)} \leftarrow (1 - \tilde{\alpha}) \mathbf{X}_{s-W+1:s} + \tilde{\alpha} \mathbf{X}_{(s+\sigma)-W+1:(s+\sigma)}$
13:          $G^{(m)} \leftarrow \partial f(\mathbf{X}^{(m)})/\partial \mathbf{X}^{(m)}$
14:          $\varphi_{\text{IG}}^{\gamma_{i,j}}(t,d) \leftarrow \varphi_{\text{IG}}^{\gamma_{i,j}}(t,d) + \left(\mathbf{X}_{t,d}^{(m)} - \mathbf{X}_{t,d}^{(m-1)}\right) \frac{G_{t,d}^{(m)} + G_{t,d}^{(m-1)}}{2}, \; \forall t, d$
15:      **end for**
16:      $M \leftarrow |T_j - (T_i - 1)|, \quad \sigma \leftarrow \text{sign}(T_j - (T_i - 1))$
17: **end for**
18:

$$\varphi_{\text{SWING}}(f, \mathbf{X}_{t,d} \mid T_1 \to T_2) \leftarrow \frac{1}{2} \sum_{i=1}^{2} \left( \varphi_{\text{IG}}^{\gamma_{i,2}}(f, \mathbf{X}_{t,d} \mid T_2) - \varphi_{\text{IG}}^{\gamma_{i,1}}(f, \mathbf{X}_{t,d} \mid T_1) \right)$$

---

By subtracting two equations, we obtain:

$$\begin{aligned}
f(\mathbf{X}_{T_2-W+1:T_2})_{\hat{c}} - f(\mathbf{X}_{T_1-W+1:T_1})_{\hat{c}} &= \sum_{t=T_1+1}^{T_2} \sum_{d=1}^{D} \varphi(f, \mathbf{X}_{t,d} \mid T_2) \\
&+ \sum_{t=T_2-W+1}^{T_1} \sum_{d=1}^{D} [\varphi(f, \mathbf{X}_{t,d} \mid T_2) - \varphi(f, \mathbf{X}_{t,d} \mid T_1)] - \sum_{t=T_1-W+1}^{T_2-W} \sum_{d=1}^{D} \varphi(f, \mathbf{X}_{t,d} \mid T_1).
\end{aligned} \tag{18}$$

### F.2 PROOF OF THEOREM 2

**Lemma 5** (Completeness of Integrated Gradients along General Paths). *Let $f : \mathbb{R}^{W \times D} \to \mathbb{R}^C$ be continuously differentiable and let $\gamma : [0,1] \to \mathbb{R}^{W \times D}$ be any continuously differentiable curve with $\gamma(0) = \mathbf{X}'$ (baseline) and $\gamma(1) = \mathbf{X}_{T-W+1:T}$ (input). Define the generalized Integrated Gradients attribution for each coordinate $(t, d)$ by:*

$$\varphi_{\text{IG}}^{\gamma}(f, \mathbf{X}_{d} \mid T) := \int_0^1 \frac{\partial f(\gamma(\alpha))_{\hat{c}}}{\partial \mathbf{X}_{t,d}} \frac{\partial \gamma_{t,d}(\alpha)}{\partial \alpha} \, d\alpha. \tag{19}$$

*Then completeness holds along* any *such curve:*

$$\sum_{t,d} \varphi_{\text{IG}}^{\gamma}(f, \mathbf{X}_{d} \mid T) = f(\mathbf{X}_{T-W+1:T})_{\hat{c}} - f(\mathbf{X}')_{\hat{c}}. \tag{20}$$

*Proof.* Stacking coordinates into a vector gives:

$$\sum_{t,d} \varphi_{\text{IG}}^{\gamma}(f, \mathbf{X}_{t,d} \mid T) = \int_0^1 \nabla f(\gamma(\alpha))_{\hat{c}}^{\top} \gamma'(\alpha) \, d\alpha = \int_{\gamma} \nabla f_{\hat{c}} \cdot d\mathbf{s} = f(\gamma(1))_{\hat{c}} - f(\gamma(0))_{\hat{c}} \tag{21}$$

by the Fundamental Theorem of Line Integrals, since the integrand is the gradient field of the scalar potential $f_{\hat{c}}$. Substituting $\gamma(1) = \mathbf{X}_{T-W+1:T}$ and $\gamma(0) = \mathbf{X}'$ completes the proof. □

**Proof of Theorem 2.** By Lemma 5 we have:

$$\sum_{i=T_1-W+1}^{T_2} \sum_{d=1}^{D} \varphi_{\text{IG}}^{\gamma_{1,2}}(f, \mathbf{X}_{i,d} \mid T_2) = f(\mathbf{X}_{T_2-W+1:T_2})_{\hat{c}} - f(\mathbf{X}_{T_1-W:T_1-1})_{\hat{c}},$$

$$\sum_{i=T_1-W+1}^{T_2} \sum_{d=1}^{D} \varphi_{\text{IG}}^{\gamma_{2,2}}(f, \mathbf{X}_{i,d} \mid T_2) = f(\mathbf{X}_{T_2-W+1:T_2})_{\hat{c}} - f(\mathbf{X}_{T_2-W:T_2-1})_{\hat{c}},$$

$$\sum_{i=T_1-W+1}^{T_2} \sum_{d=1}^{D} \varphi_{\text{IG}}^{\gamma_{1,1}}(f, \mathbf{X}_{i,d} \mid T_1) = f(\mathbf{X}_{T_1-W+1:T_1})_{\hat{c}} - f(\mathbf{X}_{T_1-W:T_1-1})_{\hat{c}},$$

$$\sum_{i=T_1-W+1}^{T_2} \sum_{d=1}^{D} \varphi_{\text{IG}}^{\gamma_{2,1}}(f, \mathbf{X}_{i,d} \mid T_1) = f(\mathbf{X}_{T_1-W+1:T_1})_{\hat{c}} - f(\mathbf{X}_{T_2-W:T_2-1})_{\hat{c}}. \tag{22}$$

SWING averages the attributions of the two paths to $\mathbf{X}_{T_2-W+1:T_2}$ and subtracts the average of the two paths to $\mathbf{X}_{T_1-W+1:T_1}$, yielding:

$$\sum_{t,d} \varphi_{\text{SWING}}(f, \mathbf{X}_{t,d} \mid T_1 \to T_2) = \frac{1}{2} \sum_{i=1}^{2} \left( \varphi_{\text{IG}}^{\gamma_{i,2}}(f, \mathbf{X}_{t,d} \mid T_2) - \varphi_{\text{IG}}^{\gamma_{i,1}}(f, \mathbf{X}_{t,d} \mid T_1) \right)$$

$$= \frac{1}{2} \left[ 2f(\mathbf{X}_{T_2-W+1:T_2})_{\hat{c}} - 2f(\mathbf{X}_{T_1-W+1:T_1})_{\hat{c}} \right] = f(\mathbf{X}_{T_2-W+1:T_2})_{\hat{c}} - f(\mathbf{X}_{T_1-W+1:T_1})_{\hat{c}}, \tag{23}$$

which establishes online completeness.

### F.3 PROOF OF THEOREM 3

SWING is defined as an average of Integrated Gradients (IG) attributions computed along multiple paths $\gamma$. For a continuously differentiable curve $\gamma : [0,1] \to \mathbb{R}^{W \times D}$ with $\gamma(0) = \mathbf{X}'$ (baseline) and $\gamma(1) = \mathbf{X}$ (input), the IG attribution is:

$$\varphi_{\text{IG}}^{\gamma}(f, \mathbf{X}_{t,d}) = \int_0^1 \frac{\partial f_{\hat{c}}(\gamma(\alpha))}{\partial \mathbf{X}_{t,d}} \frac{\partial \gamma_{t,d}(\alpha)}{\partial \alpha} \, d\alpha. \tag{24}$$

Therefore, the resulting SWING attribution $\varphi_{\text{SWING}}(f, \mathbf{X}_{t,d})$, obtained by averaging over its designated paths, depends solely on the function $f(\mathbf{X})$ and not on the particular architecture or parameterization used to realize $f$. Consequently, any two models implementing the same function yield identical SWING attributions, establishing implementation invariance.

### F.4 PROOF OF THEOREM 4

By definition,

$$\sum_{t,d} \varphi_{\text{SWING}}(f, \mathbf{X}_{t,d} \mid T_1 \to T_2) = \frac{1}{2} \sum_{i=1}^{2} \left( \sum_{t,d} \varphi_{\text{IG}}^{\gamma_{i,2}}(f, \mathbf{X}_{t,d} \mid T_2) - \sum_{t,d} \varphi_{\text{IG}}^{\gamma_{i,1}}(f, \mathbf{X}_{t,d} \mid T_1) \right). \tag{25}$$

Consider the reversed prediction change $T_2 \to T_1$. SWING uses the same four designated curves but traversed in reverse, so each IG term flips sign (antisymmetry of line integrals under reversed limits) and the two to $T_2$ terms and the two to $T_1$ terms swap roles. Hence,

$$\sum_{t,d} \varphi_{\text{SWING}}(f, \mathbf{X}_{t,d} \mid T_2 \to T_1) = \frac{1}{2} \sum_{i=1}^{2} \left( - \sum_{t,d} \varphi_{\text{IG}}^{\gamma_{i,1}}(f, \mathbf{X}_{t,d} \mid T_1) + \sum_{t,d} \varphi_{\text{IG}}^{\gamma_{i,2}}(f, \mathbf{X}_{t,d} \mid T_2) \right)$$

$$= - \sum_{t,d} \varphi_{\text{SWING}}(f, \mathbf{X}_{t,d} \mid T_1 \to T_2). \tag{26}$$

**Table 4:** We evaluate our method on five datasets: three real-world benchmarks—MIMIC-III (Johnson et al., 2016), PhysioNet 2019 (Reyna et al., 2020), and Activity (Reiss & Stricker, 2012)—and two widely used synthetic datasets—Delayed Spike (Leung et al., 2023) and Switch-Feature (Tonekaboni et al., 2020; Liu et al., 2024b).

| Type | Name | Task | # ID | # Sample / ID | Window Size | Feature | Class |
|------|------|------|------|---------------|-------------|---------|-------|
| Real-world | MIMIC-III | Decompensation prediction | 6,221 | 5 | 48 | 32 | 2 |
| | PhysioNet 2019 | Sepsis prediction | 8,066 | 5 | 48 | 40 | 2 |
| | Activity | Human action recognition | 5 | 200 | 50 | 12 | 7 |
| Synthetic | Delayed Spike | Binary classification | 1,000 | 5 | 40 | 3 | 2 |
| | Switch-Feature | Binary classification | 600 | 5 | 50 | 3 | 2 |

## G    DETAILS OF DATASETS

**MIMIC-III.**    For the MIMIC-III Clinical Database (Johnson et al., 2016), we adopt the decompensation prediction benchmark defined in Johnson et al. (2016). The dataset consists of over 41,000 ICU stays from 2001–2012, with rich multivariate time series covering vital signs, laboratory values, interventions, and demographics. Following the benchmark setup, we use sliding windows of length 48 hours with a prediction horizon of 24 hours, labeling an instance positive if the patient dies within the horizon. This yields roughly 2.5 million prediction windows, of which about 63,000 (2.5%) are positive. Each window contains irregularly sampled trajectories across up to 32 variables, making it a challenging setting for both temporal modeling and interpretability.

**PhysioNet 2019.**    For the PhysioNet 2019 dataset (Reyna et al., 2020), we adopt the sepsis prediction task defined under the Sepsis-3 criteria (Singer et al., 2016). The dataset comprises nearly 40,000 ICU stays collected across multiple hospital systems, with each stay providing multivariate physiological time series such as vitals, labs, and demographics. We define prediction windows using a 48-hour observation length and a 12-hour prediction horizon, labeling a sample positive if sepsis onset occurs within the horizon. This yields around 1.1 million prediction instances, of which approximately 27,000 (2.5%) are positive. Compared to MIMIC-III, this dataset presents higher variability in measurement density across hospitals, offering a complementary benchmark for evaluating both predictive performance and explanation reliability.

**Activity.**    We adopt the Activity dataset (Frank, 2010) and follow the preprocessing protocol from Latent ODEs (Rubanova et al., 2019). The dataset comprises 25 sequences from five individuals, each having around 6,600 time points. We segment each sequence into overlapping windows of 50 time points using a stride of 1 (unlike the stride 25 used in the original Latent ODEs paper). Labels are provided at each time point across 11 fine-grained actions; to reduce ambiguity, we merge them into seven coarse classes as in  (Rubanova et al., 2019): *walking*, *falling*, *lying*, *sitting*, *standing up*, *on all fours*, and *sitting on the ground*. For evaluation, we split by individual: the first three for training, the fourth for validation, and the fifth for testing.

**Delayed Spike.**    We adopt the Delayed Spike dataset (Leung et al., 2023), a variant of the Spike benchmark originally introduced by Tonekaboni et al. (2020) and later extended by Leung et al. (2023). The standard Spike dataset consists of three multivariate NARMA sequences with added linear trends and random spikes, where the label flips from 0 to 1 immediately after a spike appears in feature 0 and remains positive thereafter. In the Delayed Spike version, however, the label transition is shifted by two time steps: it becomes 1 exactly two steps after the spike in feature 0. This modification forces explanation methods to correctly identify the causal spike event rather than simply aligning with the delayed label change.

**Switch-Feature.**    We generate the Switch-Feature dataset following the design in FIT (Tonekaboni et al., 2020). Similar to the State dataset, it is constructed based on a three-state hidden Markov model with an initial distribution $\pi = (1/3, 1/3, 1/3)$ and the following transition matrix:

$$\begin{pmatrix} 0.95 & 0.02 & 0.03 \\ 0.02 & 0.95 & 0.03 \\ 0.03 & 0.02 & 0.95 \end{pmatrix}.$$

Each hidden state emits a time series from a Gaussian process with an RBF kernel ($\gamma = 0.2$) and a fixed marginal variance of 0.1 for all features. The mean vectors of the three states are given by $\mu_1 = [0.8, 0.5, 0.2]$, $\mu_2 = [0, 1.0, 0]$, and $\mu_3 = [0.2, 0.2, 0.8]$. Labels $y_i[t]$ are sampled from a Bernoulli distribution Bernoulli($p_i[t]$), where:

$$p_i[t] = \begin{cases} (1 + \exp(-\mathbf{X}_{t,1}))^{-1} & \text{if } s_t = 0, \\ (1 + \exp(-\mathbf{X}_{t,2}))^{-1} & \text{if } s_t = 1, \\ (1 + \exp(-\mathbf{X}_{t,3}))^{-1} & \text{if } s_t = 2. \end{cases} \tag{27}$$

and $s_t$ denotes the latent state at time $t$. In our experiments, we generate sequences of length 100 and extract multiple online prediction samples from each sequence using a fixed-length sliding window of size 50.

## H    DETAILS OF ABLATION STUDY

In this section, we clarify the meaning of each ablation configuration and present its corresponding mathematical formulation. We independently verify the contribution of each SWING component by removing 1) both RBS and PHI, 2) RBS alone, and 3) DPI ($\gamma_{1,1}$, $\gamma_{2,2}$). The following variants correspond directly to the results in Table 3 and ensure that each ablation is rigorously defined and remains faithful to the original algorithm.

**SWING w/o RBS, PHI.**    This variant removes both the retrospective baseline selection (RBS) and the piecewise-linear historical integration (PHI), leaving only DPI active. Both integration paths start from the zero baseline, and the resulting attribution reduces to subtracting Integrated Gradients computed at $T_2$ and $T_1$:

$$\varphi_{\text{SWING (w/o RBS, PHI)}} = \varphi_{\text{IG}}^{\gamma_2}(f, X_{t,d} \mid T_2) - \varphi_{\text{IG}}^{\gamma_1}(f, X_{t,d} \mid T_1), \tag{28}$$

where each path is a straight line from zero to the corresponding input window $\gamma_i(\alpha) = \alpha X_{T_i - W + 1:T_i}$. This configuration is mathematically equivalent to applying IG independently at two time points and comparing their attributions.

**SWING w/o RBS.**    This configuration removes only the baseline-selection component. The retrospective baselines $X_{T_i - W:T_i - 1}$ are replaced with the zero baseline, while all DPI mechanisms and the PHI interpolation remain unchanged. As a result, only the initial segment of each integration path is modified: the path originates at zero rather than at $X_{T_i - W:T_i - 1}$, but all subsequent interpolation segments and the dual-path structure are preserved. This ablation isolates the effect of removing temporal baseline adaptation while retaining both historical integration and dual-path comparison.

**SWING w/o DPI ($\gamma_{1,1}$, $\gamma_{2,2}$).**    This variant removes the self-window DPI components $\gamma_{1,1}$ and $\gamma_{2,2}$, while keeping the cross-window paths $\gamma_{1,2}$ and $\gamma_{2,1}$ active. Since the cross-window paths are essential for attributing prediction changes between $T_1$ and $T_2$, PHI remains applied to them, yielding the following attribution:

$$\varphi_{\text{SWING (w/o DPI)}} = \varphi_{\text{IG}}^{\gamma_{2,1}}(f, X_{t,d} \mid T_2) - \varphi_{\text{IG}}^{\gamma_{1,2}}(f, X_{t,d} \mid T_1). \tag{29}$$

By eliminating only the self-window paths while retaining the cross-window integration, this ablation isolates the contribution of DPI to SWING's temporal attribution mechanism.

## I    ADDITIONAL EXPERIMENTS

This section presents additional experiments that complement the main findings. The content is organized into extended quantitative evaluations, backbone generalization, longer prediction intervals, ablation and robustness analyses, efficiency, qualitative case studies, and clinical coherence.

**Extended benchmarks.**    Tables 6 and 7 provide further results on clinical datasets, synthetic benchmarks, and the Activity dataset. Across all settings, SWING achieves the best or second-best scores on most metrics, reinforcing its robustness across both controlled and real-world scenarios.

**Table 5:** Evaluation metrics for prediction-change explanations on the MIMIC-III decompensation benchmark with an LSTM backbone. We follow the same setting as the main experiments, but evaluate by removing the most salient 50 feature points per time step with forward-fill substitution.

| Algorithm | CPD ↑ | AUPD ↑ | Corr. ↑ |
|---|---|---|---|
| Dynamask w/ Naive Subtraction | $0.14_{\pm 0.00}$ | $0.12_{\pm 0.00}$ | $-0.19_{\pm 0.00}$ |
| Dynamask in Delta-XAI | $\mathbf{11.72}_{\pm 0.08}$ | $\mathbf{7.56}_{\pm 0.04}$ | $\mathbf{0.04}_{\pm 0.00}$ |

**Table 6:** Performance comparison of XAI methods on clinical prediction tasks: PhysioNet 2019 sepsis benchmark using LSTM as backbone architecture. Evaluation is performed by removing the most or least salient 50 feature points per time step, using forward-fill substitution.

| Algorithm | Removal of Most Salient 50 Points | | | | Removal of Least Salient 50 Points | | | | Corr. ↑ |
|---|---|---|---|---|---|---|---|---|---|
| | CPD ↑ | AUPD ↑ | MPD ↑ | AUMPD ↑ | CPP ↓ | AUPP ↓ | MPP ↓ | AUMPP ↓ | |
| LIME (Ribeiro et al., 2016) | $0.29_{\pm 0.00}$ | $0.21_{\pm 0.00}$ | $1.83_{\pm 0.01}$ | $1.02_{\pm 0.00}$ | $3.66_{\pm 0.01}$ | $1.82_{\pm 0.00}$ | $3.96_{\pm 0.00}$ | $2.09_{\pm 0.01}$ | $-0.08_{\pm 0.00}$ |
| GradSHAP (Lundberg & Lee, 2017) | $1.60_{\pm 0.00}$ | $0.93_{\pm 0.00}$ | $2.52_{\pm 0.01}$ | $1.50_{\pm 0.00}$ | $3.48_{\pm 0.00}$ | $1.64_{\pm 0.00}$ | $3.75_{\pm 0.01}$ | $1.86_{\pm 0.00}$ | $0.02_{\pm 0.00}$ |
| IG (Sundararajan et al., 2017) | $2.68_{\pm 0.00}$ | $1.54_{\pm 0.00}$ | $\underline{3.00}_{\pm 0.01}$ | $\underline{1.93}_{\pm 0.00}$ | $3.19_{\pm 0.01}$ | $1.43_{\pm 0.00}$ | $3.43_{\pm 0.01}$ | $1.61_{\pm 0.00}$ | $0.11_{\pm 0.00}$ |
| DeepLIFT (Shrikumar et al., 2017) | $2.72_{\pm 0.00}$ | $\underline{1.60}_{\pm 0.00}$ | $2.87_{\pm 0.00}$ | $1.86_{\pm 0.00}$ | $\underline{3.06}_{\pm 0.01}$ | $\underline{1.35}_{\pm 0.00}$ | $3.38_{\pm 0.01}$ | $1.56_{\pm 0.00}$ | $\underline{0.16}_{\pm 0.00}$ |
| FO (Suresh et al., 2017) | $1.17_{\pm 0.00}$ | $0.78_{\pm 0.00}$ | $2.75_{\pm 0.00}$ | $1.80_{\pm 0.00}$ | $4.63_{\pm 0.01}$ | $3.13_{\pm 0.01}$ | $2.44_{\pm 0.01}$ | $1.13_{\pm 0.00}$ | $0.02_{\pm 0.00}$ |
| AFO (Tonekaboni et al., 2020) | $1.87_{\pm 0.00}$ | $1.16_{\pm 0.00}$ | $2.88_{\pm 0.01}$ | $1.86_{\pm 0.00}$ | $3.12_{\pm 0.00}$ | $1.48_{\pm 0.00}$ | $3.28_{\pm 0.01}$ | $1.55_{\pm 0.00}$ | $0.15_{\pm 0.00}$ |
| FIT (Tonekaboni et al., 2020) | $0.71_{\pm 0.00}$ | $0.48_{\pm 0.00}$ | $1.10_{\pm 0.00}$ | $0.91_{\pm 0.00}$ | $3.81_{\pm 0.00}$ | $1.96_{\pm 0.00}$ | $\mathbf{1.30}_{\pm 0.00}$ | $\underline{1.02}_{\pm 0.00}$ | $0.00_{\pm 0.00}$ |
| WinIT (Leung et al., 2023) | $1.86_{\pm 0.00}$ | $1.10_{\pm 0.00}$ | $2.85_{\pm 0.01}$ | $1.66_{\pm 0.00}$ | $3.88_{\pm 0.00}$ | $2.02_{\pm 0.00}$ | $3.12_{\pm 0.01}$ | $1.64_{\pm 0.00}$ | $0.06_{\pm 0.00}$ |
| Dynamask (Crabbé & Van Der Schaar, 2021) | $1.69_{\pm 0.00}$ | $1.11_{\pm 0.00}$ | $2.05_{\pm 0.00}$ | $1.36_{\pm 0.00}$ | $4.98_{\pm 0.01}$ | $2.77_{\pm 0.00}$ | $4.74_{\pm 0.01}$ | $2.59_{\pm 0.00}$ | $0.06_{\pm 0.00}$ |
| Extrmask (Enguehard, 2023) | $1.16_{\pm 0.00}$ | $0.72_{\pm 0.00}$ | $1.91_{\pm 0.00}$ | $1.11_{\pm 0.00}$ | $4.04_{\pm 0.01}$ | $2.41_{\pm 0.00}$ | $3.98_{\pm 0.01}$ | $2.36_{\pm 0.01}$ | $0.05_{\pm 0.00}$ |
| ContraLSP (Liu et al., 2024b) | $0.73_{\pm 0.02}$ | $0.34_{\pm 0.01}$ | $2.47_{\pm 0.02}$ | $1.32_{\pm 0.01}$ | $5.32_{\pm 0.04}$ | $2.93_{\pm 0.04}$ | $5.35_{\pm 0.04}$ | $2.97_{\pm 0.04}$ | $0.04_{\pm 0.00}$ |
| TimeX (Queen et al., 2024) | $0.74_{\pm 0.00}$ | $0.36_{\pm 0.00}$ | $1.96_{\pm 0.01}$ | $1.00_{\pm 0.00}$ | $5.33_{\pm 0.01}$ | $2.77_{\pm 0.00}$ | $5.49_{\pm 0.01}$ | $2.89_{\pm 0.01}$ | $-0.03_{\pm 0.00}$ |
| TimeX++ (Liu et al., 2024a) | $1.76_{\pm 0.01}$ | $1.06_{\pm 0.00}$ | $2.07_{\pm 0.01}$ | $1.19_{\pm 0.00}$ | $3.99_{\pm 0.01}$ | $1.81_{\pm 0.00}$ | $4.02_{\pm 0.01}$ | $1.82_{\pm 0.00}$ | $0.05_{\pm 0.00}$ |
| TIMING (Jang et al., 2025) | $\underline{2.73}_{\pm 0.00}$ | $1.56_{\pm 0.00}$ | $\mathbf{3.02}_{\pm 0.00}$ | $\mathbf{1.94}_{\pm 0.00}$ | $3.12_{\pm 0.00}$ | $1.41_{\pm 0.00}$ | $3.38_{\pm 0.00}$ | $1.60_{\pm 0.00}$ | $0.13_{\pm 0.00}$ |
| SWING | $\mathbf{3.10}_{\pm 0.01}$ | $\mathbf{1.96}_{\pm 0.00}$ | $2.81_{\pm 0.01}$ | $1.78_{\pm 0.01}$ | $\mathbf{2.27}_{\pm 0.01}$ | $\mathbf{0.93}_{\pm 0.00}$ | $2.38_{\pm 0.00}$ | $\mathbf{1.01}_{\pm 0.00}$ | $\mathbf{0.32}_{\pm 0.02}$ |

**Backbone architectures.** Table 8 compares CNN and Transformer backbones on MIMIC-III. The results show that SWING retains strong performance across both architectures, demonstrating versatility beyond LSTMs.

**Longer time intervals.** Table 9 evaluates explanations at $T_2 - T_1 = 6$ and 24. Although CPD and AUPD converge across methods with longer intervals, SWING consistently achieves best or near-best results, and shows particularly dominant gains on preservation metrics.

**Ablation and robustness.** Table 3 analyzes the contributions of RBS, PHI, and DPI. Removing RBS or PHI degrades performance, confirming their complementary roles, while DPI mainly stabilizes preservation. Baseline offset analysis further shows that $d = 1$ yields the best trade-off, with $d = 0$ and larger offsets leading to degradation. Figure 4 confirms robustness across a broad range of $n_{\text{samples}}$. Subgroup and temporal resolution analyses (Tables 10 to 12) additionally show that SWING preserves its advantages regardless of prediction or label changes and remains stable across both 24- and 72-length windows.

**Efficiency.** Figure 3 demonstrates that SWING achieves state-of-the-art explanatory quality without significant computational overhead. Runtime (0.35s/sample) and memory (448 MB) remain comparable to gradient-based baselines such as IG and GradSHAP.

**Qualitative and clinical coherence.** Figures 5 to 9 visualize attribution maps, where SWING produces sharper, localized explanations than surrogate or masking methods. Finally, Figure 10 confirms coherence with medical knowledge, correctly highlighting risk increases from SBP drops and pH declines, and protective effects from $SpO_2$ rises.

Overall, these supplementary experiments confirm that SWING consistently delivers reliable, efficient, and clinically meaningful explanations across diverse datasets, models, and experimental conditions.

**Table 7:** Performance of XAI methods on (top) Activity (Vidulin et al., 2010), (middle) Delayed Spike (Leung et al., 2023), and (bottom) Switch-Feature (Tonekaboni et al., 2020) benchmarks with an LSTM backbone. Evaluation is performed by removing the most or least salient 50 feature points per time step, using forward-fill substitution.

| Algorithm | Removal of Most Salient 50 Points | | | | Removal of Least Salient 50 Points | | | | Corr. ↑ |
| --- | --- | --- | --- | --- | --- | --- | --- | --- | --- |
| | CPD ↑ | AUPD ↑ | MPD ↑ | AUMPD ↑ | CPP ↓ | AUPP ↓ | MPP ↓ | AUMPP ↓ | |
| IG (Sundararajan et al., 2017) | $19.80_{\pm0.53}$ | $12.53_{\pm0.31}$ | $7.81_{\pm0.53}$ | $4.76_{\pm0.28}$ | $17.51_{\pm0.70}$ | $7.33_{\pm0.28}$ | $20.23_{\pm1.06}$ | $8.77_{\pm0.45}$ | $0.09_{\pm0.01}$ |
| DeepLIFT (Shrikumar et al., 2017) | $21.01_{\pm0.59}$ | $13.25_{\pm0.36}$ | $8.15_{\pm0.52}$ | $5.10_{\pm0.30}$ | $15.26_{\pm0.49}$ | $6.22_{\pm0.17}$ | $19.07_{\pm0.61}$ | $8.20_{\pm0.23}$ | $0.17_{\pm0.02}$ |
| AFO (Tonekaboni et al., 2020) | $15.20_{\pm0.52}$ | $10.11_{\pm0.29}$ | $7.51_{\pm0.38}$ | $4.79_{\pm0.25}$ | $13.69_{\pm0.92}$ | $5.66_{\pm0.35}$ | $18.23_{\pm0.77}$ | $7.99_{\pm0.28}$ | $0.17_{\pm0.01}$ |
| WinIT (Leung et al., 2023) | $6.62_{\pm0.38}$ | $3.34_{\pm0.15}$ | $6.61_{\pm0.42}$ | $3.87_{\pm0.25}$ | $24.10_{\pm1.26}$ | $12.61_{\pm0.63}$ | $20.16_{\pm1.09}$ | $9.45_{\pm0.54}$ | $0.09_{\pm0.01}$ |
| TIMING (Jang et al., 2025) | $13.75_{\pm0.71}$ | $8.08_{\pm0.37}$ | $7.18_{\pm0.54}$ | $4.35_{\pm0.35}$ | $19.39_{\pm1.06}$ | $9.28_{\pm0.53}$ | $19.44_{\pm0.74}$ | $8.69_{\pm0.33}$ | $0.08_{\pm0.01}$ |
| SWING | $21.77_{\pm0.80}$ | $14.96_{\pm0.63}$ | $9.20_{\pm0.83}$ | $6.19_{\pm0.60}$ | $7.64_{\pm0.33}$ | $2.70_{\pm0.13}$ | $15.15_{\pm0.37}$ | $6.05_{\pm0.12}$ | $0.46_{\pm0.03}$ |
| IG (Sundararajan et al., 2017) | $290.74_{\pm1.01}$ | $246.74_{\pm1.03}$ | $281.63_{\pm1.67}$ | $225.76_{\pm1.20}$ | $13.14_{\pm0.22}$ | $4.30_{\pm0.07}$ | $15.78_{\pm0.32}$ | $5.59_{\pm0.10}$ | $0.46_{\pm0.01}$ |
| DeepLIFT (Shrikumar et al., 2017) | $289.95_{\pm1.37}$ | $251.30_{\pm1.05}$ | $277.38_{\pm1.08}$ | $228.97_{\pm0.71}$ | $11.47_{\pm0.23}$ | $3.69_{\pm0.08}$ | $14.85_{\pm0.22}$ | $5.11_{\pm0.08}$ | $0.58_{\pm0.00}$ |
| AFO (Tonekaboni et al., 2020) | $256.70_{\pm1.90}$ | $189.27_{\pm1.42}$ | $240.04_{\pm1.70}$ | $165.29_{\pm1.22}$ | $7.83_{\pm0.19}$ | $2.66_{\pm0.06}$ | $12.29_{\pm0.33}$ | $4.46_{\pm0.19}$ | $0.19_{\pm0.00}$ |
| WinIT (Leung et al., 2023) | $426.69_{\pm5.33}$ | $114.50_{\pm1.37}$ | $191.86_{\pm1.38}$ | $56.91_{\pm0.31}$ | $398.05_{\pm4.40}$ | $344.09_{\pm3.76}$ | $180.94_{\pm1.96}$ | $46.32_{\pm0.48}$ | $-0.02_{\pm0.00}$ |
| TIMING (Jang et al., 2025) | $327.29_{\pm3.46}$ | $199.72_{\pm2.20}$ | $205.82_{\pm2.33}$ | $96.03_{\pm0.94}$ | $261.30_{\pm2.42}$ | $230.69_{\pm2.10}$ | $205.20_{\pm2.37}$ | $92.31_{\pm0.97}$ | $-0.01_{\pm0.00}$ |
| SWING | $317.95_{\pm2.50}$ | $252.16_{\pm1.86}$ | $305.09_{\pm2.06}$ | $232.50_{\pm1.42}$ | $3.35_{\pm0.04}$ | $1.13_{\pm0.02}$ | $5.70_{\pm0.10}$ | $1.85_{\pm0.04}$ | $0.44_{\pm0.00}$ |
| IG (Sundararajan et al., 2017) | $464.24_{\pm6.43}$ | $422.12_{\pm5.83}$ | $474.36_{\pm5.58}$ | $395.55_{\pm4.59}$ | $10.97_{\pm0.13}$ | $4.29_{\pm0.05}$ | $17.94_{\pm0.61}$ | $6.56_{\pm0.16}$ | $0.65_{\pm0.00}$ |
| DeepLIFT (Shrikumar et al., 2017) | $525.16_{\pm7.67}$ | $470.90_{\pm6.67}$ | $523.95_{\pm5.31}$ | $430.36_{\pm4.01}$ | $8.66_{\pm0.10}$ | $3.45_{\pm0.04}$ | $16.35_{\pm0.56}$ | $5.97_{\pm0.21}$ | $0.63_{\pm0.01}$ |
| AFO (Tonekaboni et al., 2020) | $533.94_{\pm7.78}$ | $475.55_{\pm6.78}$ | $529.43_{\pm6.72}$ | $431.48_{\pm4.92}$ | $9.88_{\pm0.21}$ | $3.98_{\pm0.10}$ | $13.30_{\pm0.30}$ | $5.37_{\pm0.14}$ | $0.58_{\pm0.00}$ |
| WinIT (Leung et al., 2023) | $507.88_{\pm5.15}$ | $447.50_{\pm5.05}$ | $496.19_{\pm7.13}$ | $391.83_{\pm6.16}$ | $12.46_{\pm0.19}$ | $7.46_{\pm0.10}$ | $311.71_{\pm5.23}$ | $113.50_{\pm1.90}$ | $0.53_{\pm0.01}$ |
| TIMING (Jang et al., 2025) | $418.65_{\pm3.35}$ | $391.51_{\pm3.17}$ | $446.03_{\pm4.22}$ | $373.96_{\pm3.50}$ | $13.97_{\pm0.22}$ | $7.54_{\pm0.12}$ | $507.71_{\pm6.18}$ | $255.11_{\pm3.26}$ | $0.64_{\pm0.01}$ |
| SWING | $536.43_{\pm6.46}$ | $481.90_{\pm5.86}$ | $519.55_{\pm5.26}$ | $436.18_{\pm4.42}$ | $7.32_{\pm0.10}$ | $2.99_{\pm0.06}$ | $15.84_{\pm0.53}$ | $5.81_{\pm0.20}$ | $0.74_{\pm0.00}$ |

**Table 8:** Performance comparison of XAI methods on MIMIC-III decompensation prediction task with different backbone architectures: CNN (top) and Transformer (bottom) architectures. Evaluation is performed by removing the most or least salient 50 feature points per time step, using forward-fill substitution.

| Algorithm | Removal of Most Salient 50 Points | | | | Removal of Least Salient 50 Points | | | | Corr. ↑ |
| --- | --- | --- | --- | --- | --- | --- | --- | --- | --- |
| | CPD ↑ | AUPD ↑ | MPD ↑ | AUMPD ↑ | CPP ↓ | AUPP ↓ | MPP ↓ | AUMPP ↓ | |
| IG (Sundararajan et al., 2017) | $28.68_{\pm0.04}$ | $16.83_{\pm0.03}$ | $32.16_{\pm0.09}$ | $20.25_{\pm0.06}$ | $41.37_{\pm0.06}$ | $19.08_{\pm0.04}$ | $37.25_{\pm0.05}$ | $15.63_{\pm0.04}$ | $0.24_{\pm0.00}$ |
| DeepLIFT (Shrikumar et al., 2017) | $29.59_{\pm0.06}$ | $17.60_{\pm0.04}$ | $32.14_{\pm0.13}$ | $20.29_{\pm0.08}$ | $41.69_{\pm0.12}$ | $19.13_{\pm0.05}$ | $37.68_{\pm0.10}$ | $15.65_{\pm0.03}$ | $0.28_{\pm0.00}$ |
| AFO (Tonekaboni et al., 2020) | $20.64_{\pm0.14}$ | $13.42_{\pm0.09}$ | $32.35_{\pm0.15}$ | $20.16_{\pm0.09}$ | $50.97_{\pm0.12}$ | $25.25_{\pm0.07}$ | $34.30_{\pm0.08}$ | $14.49_{\pm0.04}$ | $0.30_{\pm0.00}$ |
| WinIT (Leung et al., 2023) | $23.15_{\pm0.07}$ | $13.35_{\pm0.04}$ | $32.69_{\pm0.09}$ | $18.44_{\pm0.06}$ | $55.61_{\pm0.16}$ | $25.09_{\pm0.08}$ | $42.21_{\pm0.07}$ | $19.95_{\pm0.04}$ | $0.23_{\pm0.00}$ |
| TIMING (Jang et al., 2025) | $29.80_{\pm0.06}$ | $17.26_{\pm0.05}$ | $32.33_{\pm0.12}$ | $20.45_{\pm0.08}$ | $40.54_{\pm0.10}$ | $18.86_{\pm0.05}$ | $36.10_{\pm0.09}$ | $15.26_{\pm0.06}$ | $0.26_{\pm0.00}$ |
| SWING | $44.02_{\pm0.17}$ | $27.85_{\pm0.12}$ | $40.36_{\pm0.12}$ | $25.86_{\pm0.10}$ | $24.47_{\pm0.07}$ | $8.23_{\pm0.03}$ | $24.97_{\pm0.03}$ | $8.44_{\pm0.03}$ | $0.53_{\pm0.00}$ |

| Algorithm | Removal of Most Salient 50 Points | | | | Removal of Least Salient 50 Points | | | | Corr. ↑ |
| --- | --- | --- | --- | --- | --- | --- | --- | --- | --- |
| | CPD ↑ | AUPD ↑ | MPD ↑ | AUMPD ↑ | CPP ↓ | AUPP ↓ | MPP ↓ | AUMPP ↓ | |
| IG (Sundararajan et al., 2017) | $16.33_{\pm0.06}$ | $9.77_{\pm0.05}$ | $15.26_{\pm0.07}$ | $10.42_{\pm0.05}$ | $33.52_{\pm0.21}$ | $14.58_{\pm0.12}$ | $27.42_{\pm0.10}$ | $10.42_{\pm0.07}$ | $0.15_{\pm0.00}$ |
| DeepLIFT (Shrikumar et al., 2017) | $15.11_{\pm0.07}$ | $9.87_{\pm0.06}$ | $15.40_{\pm0.04}$ | $10.53_{\pm0.05}$ | $33.24_{\pm0.22}$ | $14.13_{\pm0.11}$ | $28.24_{\pm0.17}$ | $10.40_{\pm0.10}$ | $0.14_{\pm0.00}$ |
| AFO (Tonekaboni et al., 2020) | $14.61_{\pm0.06}$ | $10.22_{\pm0.03}$ | $17.51_{\pm0.03}$ | $12.06_{\pm0.03}$ | $37.26_{\pm0.19}$ | $18.10_{\pm0.09}$ | $23.78_{\pm0.16}$ | $9.23_{\pm0.10}$ | $0.23_{\pm0.00}$ |
| WinIT (Leung et al., 2023) | $17.20_{\pm0.05}$ | $10.04_{\pm0.04}$ | $23.99_{\pm0.14}$ | $14.12_{\pm0.08}$ | $34.44_{\pm0.12}$ | $16.04_{\pm0.09}$ | $25.92_{\pm0.10}$ | $12.36_{\pm0.06}$ | $0.12_{\pm0.00}$ |
| TIMING (Jang et al., 2025) | $22.53_{\pm0.05}$ | $12.59_{\pm0.03}$ | $15.95_{\pm0.07}$ | $10.94_{\pm0.05}$ | $31.53_{\pm0.13}$ | $14.04_{\pm0.08}$ | $25.78_{\pm0.16}$ | $10.01_{\pm0.08}$ | $0.18_{\pm0.00}$ |
| SWING | $25.50_{\pm0.13}$ | $16.38_{\pm0.09}$ | $23.71_{\pm0.15}$ | $15.84_{\pm0.09}$ | $17.57_{\pm0.13}$ | $5.98_{\pm0.06}$ | $18.05_{\pm0.13}$ | $6.30_{\pm0.07}$ | $0.26_{\pm0.00}$ |

**Table 9:** Performance comparison of XAI methods on MIMIC-III decompensation prediction task with various time interval settings between two time steps $T_1 < T_2$: 6 timestamps interval (top) and 24 timestamps interval (bottom) using LSTM as backbone architecture. Evaluation is performed by removing the most or least salient 50 feature points per time step, using forward-fill substitution.

| Algorithm | Removal of Most Salient 50 Points | | | | Removal of Least Salient 50 Points | | | | Corr. ↑ |
| --- | --- | --- | --- | --- | --- | --- | --- | --- | --- |
| | CPD ↑ | AUPD ↑ | MPD ↑ | AUMPD ↑ | CPP ↓ | AUPP ↓ | MPP ↓ | AUMPP ↓ | |
| IG (Sundararajan et al., 2017) | $44.17_{\pm0.15}$ | $26.92_{\pm0.10}$ | $45.53_{\pm0.23}$ | $29.23_{\pm0.13}$ | $91.67_{\pm0.46}$ | $39.68_{\pm0.17}$ | $76.67_{\pm0.28}$ | $28.60_{\pm0.14}$ | $0.22_{\pm0.00}$ |
| DeepLIFT (Shrikumar et al., 2017) | $42.00_{\pm0.18}$ | $25.91_{\pm0.10}$ | $44.50_{\pm0.22}$ | $28.54_{\pm0.12}$ | $100.64_{\pm0.54}$ | $41.62_{\pm0.21}$ | $84.92_{\pm0.43}$ | $30.38_{\pm0.18}$ | $0.19_{\pm0.00}$ |
| AFO (Tonekaboni et al., 2020) | $35.21_{\pm0.10}$ | $22.98_{\pm0.07}$ | $47.47_{\pm0.16}$ | $30.01_{\pm0.09}$ | $106.36_{\pm0.57}$ | $49.81_{\pm0.23}$ | $65.85_{\pm0.47}$ | $25.13_{\pm0.22}$ | $0.22_{\pm0.00}$ |
| WinIT (Leung et al., 2023) | $52.98_{\pm0.02}$ | $28.48_{\pm0.01}$ | $19.19_{\pm0.01}$ | $10.47_{\pm0.00}$ | $91.22_{\pm0.09}$ | $43.19_{\pm0.04}$ | $83.18_{\pm0.51}$ | $41.23_{\pm0.25}$ | $0.14_{\pm0.00}$ |
| TIMING (Jang et al., 2025) | $52.76_{\pm0.23}$ | $30.71_{\pm0.13}$ | $47.30_{\pm0.17}$ | $30.32_{\pm0.12}$ | $87.90_{\pm0.29}$ | $38.52_{\pm0.14}$ | $72.48_{\pm0.35}$ | $27.26_{\pm0.15}$ | $0.23_{\pm0.00}$ |
| SWING | $51.46_{\pm0.24}$ | $33.55_{\pm0.14}$ | $55.64_{\pm0.36}$ | $35.81_{\pm0.24}$ | $61.90_{\pm0.29}$ | $20.75_{\pm0.11}$ | $63.95_{\pm0.19}$ | $21.90_{\pm0.11}$ | $0.29_{\pm0.00}$ |

| Algorithm | Removal of Most Salient 50 Points | | | | Removal of Least Salient 50 Points | | | | Corr. ↑ |
| --- | --- | --- | --- | --- | --- | --- | --- | --- | --- |
| | CPD ↑ | AUPD ↑ | MPD ↑ | AUMPD ↑ | CPP ↓ | AUPP ↓ | MPP ↓ | AUMPP ↓ | |
| IG (Sundararajan et al., 2017) | $45.75_{\pm0.19}$ | $26.81_{\pm0.14}$ | $40.79_{\pm0.24}$ | $26.37_{\pm0.15}$ | $103.48_{\pm0.45}$ | $48.12_{\pm0.20}$ | $77.63_{\pm0.44}$ | $29.38_{\pm0.15}$ | $0.20_{\pm0.00}$ |
| DeepLIFT (Shrikumar et al., 2017) | $40.53_{\pm0.18}$ | $24.80_{\pm0.12}$ | $40.16_{\pm0.26}$ | $25.96_{\pm0.15}$ | $109.50_{\pm0.52}$ | $49.11_{\pm0.22}$ | $83.81_{\pm0.39}$ | $30.47_{\pm0.15}$ | $0.18_{\pm0.00}$ |
| AFO (Tonekaboni et al., 2020) | $31.04_{\pm0.21}$ | $20.46_{\pm0.16}$ | $42.44_{\pm0.23}$ | $26.92_{\pm0.15}$ | $120.00_{\pm0.30}$ | $61.17_{\pm0.12}$ | $64.36_{\pm0.35}$ | $25.25_{\pm0.08}$ | $0.18_{\pm0.00}$ |
| TIMING (Jang et al., 2025) | $58.83_{\pm0.30}$ | $32.38_{\pm0.22}$ | $42.52_{\pm0.26}$ | $27.42_{\pm0.17}$ | $97.93_{\pm0.41}$ | $46.75_{\pm0.18}$ | $72.36_{\pm0.35}$ | $27.82_{\pm0.15}$ | $0.22_{\pm0.00}$ |
| SWING | $41.07_{\pm0.22}$ | $26.46_{\pm0.18}$ | $50.58_{\pm0.28}$ | $32.29_{\pm0.20}$ | $60.29_{\pm0.14}$ | $21.60_{\pm0.08}$ | $64.43_{\pm0.19}$ | $23.87_{\pm0.10}$ | $0.21_{\pm0.00}$ |

**Table 10:** Subgroup analysis of attribution methods under different score-change conditions (high: $|\Delta| \geq 0.1$ (top), low: $|\Delta| < 0.1$ (bottom), where $\Delta = f(\mathbf{X}_{T_2-W+1:T_2}) - f(\mathbf{X}_{T_1-W+1:T_1})$). Results are reported on the MIMIC-III decompensation benchmark with an LSTM backbone with $T_2-T_1 = 1$. Evaluation is performed by removing the most or least salient 50 feature points per time step, using forward-fill substitution.

| Algorithm | Removal of Most Salient 50 Points | | | | Removal of Least Salient 50 Points | | | | Corr. ↑ |
|---|---|---|---|---|---|---|---|---|---|
| | CPD ↑ | AUPD ↑ | MPD ↑ | AUMPD ↑ | CPP ↓ | AUPP ↓ | MPP ↓ | AUMPP ↓ | |
| IG (Sundararajan et al., 2017) | 281.32±0.42 | 222.60±0.37 | 319.69±0.75 | 251.67±0.58 | 381.59±0.71 | 186.83±0.36 | 349.24±0.73 | 147.10±0.22 | 0.37±0.00 |
| DeepLIFT (Shrikumar et al., 2017) | 296.81±0.72 | 236.89±0.50 | 326.36±0.43 | 257.72±0.42 | 371.51±0.66 | 181.49±0.35 | 334.79±0.64 | 140.56±0.45 | 0.29±0.00 |
| AFO (Tonekaboni et al., 2020) | 279.71±0.31 | 224.13±0.19 | 329.23±0.83 | 257.53±0.36 | 371.05±0.86 | 190.70±0.78 | 316.06±0.52 | 133.02±0.38 | 0.42±0.00 |
| WinIT (Leung et al., 2023) | 341.14±0.65 | 244.84±0.35 | 408.03±0.65 | 283.17±0.26 | 350.94±1.06 | 182.03±0.51 | 366.12±1.34 | 194.42±0.77 | 0.23±0.00 |
| TIMING (Jang et al., 2025) | 285.87±0.53 | 224.72±0.42 | 322.54±0.81 | 253.72±0.76 | 378.87±1.29 | 184.84±0.49 | 346.25±0.39 | 145.34±0.21 | 0.37±0.00 |
| SWING | **397.28±0.73** | **311.94±0.52** | 388.43±0.93 | **308.56±0.79** | **261.28±0.55** | **96.11±0.24** | **264.34±0.77** | **98.10±0.29** | **0.57±0.00** |

| Algorithm | Removal of Most Salient 50 Points | | | | Removal of Least Salient 50 Points | | | | Corr. ↑ |
|---|---|---|---|---|---|---|---|---|---|
| | CPD ↑ | AUPD ↑ | MPD ↑ | AUMPD ↑ | CPP ↓ | AUPP ↓ | MPP ↓ | AUMPP ↓ | |
| IG (Sundararajan et al., 2017) | 11.44±0.03 | 7.44±0.01 | 14.04±0.06 | 9.53±0.03 | 30.33±0.06 | 12.58±0.03 | 26.29±0.12 | 9.56±0.04 | 0.09±0.00 |
| DeepLIFT (Shrikumar et al., 2017) | 11.44±0.03 | 7.63±0.02 | 13.79±0.06 | 9.39±0.04 | 32.95±0.07 | 13.29±0.04 | 28.57±0.07 | 10.26±0.04 | 0.13±0.00 |
| AFO (Tonekaboni et al., 2020) | 11.35±0.08 | 7.71±0.05 | 15.03±0.09 | 10.20±0.06 | 33.20±0.07 | 15.29±0.04 | 21.31±0.05 | 8.24±0.03 | 0.19±0.00 |
| WinIT (Leung et al., 2023) | 17.58±0.10 | 10.82±0.07 | 22.35±0.13 | 13.65±0.08 | 25.97±0.19 | 11.39±0.06 | 23.13±0.16 | 10.38±0.06 | 0.20±0.00 |
| TIMING (Jang et al., 2025) | 13.04±0.07 | 8.08±0.03 | 14.35±0.07 | 9.72±0.04 | 28.14±0.08 | 12.01±0.03 | 24.18±0.07 | 9.05±0.02 | 0.11±0.00 |
| SWING | **21.25±0.08** | **14.09±0.07** | 19.63±0.13 | 13.34±0.10 | **15.34±0.09** | **4.97±0.02** | **15.09±0.12** | **5.21±0.02** | **0.32±0.00** |

**Table 11:** Subgroup analysis of attribution methods with respect to prediction stability. Cases are divided into *changed* (top) vs. *unchanged* (bottom), depending on whether the predicted class label (*i.e.*, $\arg\max f(\mathbf{X}_{T_1-W+1:T_1})$ vs. $\arg\max f(\mathbf{X}_{T_2-W+1:T_2})$) is different or remains the same. Results are reported on the MIMIC-III decompensation benchmark with an LSTM backbone ($T_2 - T_1 = 1$). Evaluation is performed by removing the most or least salient 50 feature points per time step, using forward-fill substitution.

| Algorithm | Removal of Most Salient 50 Points | | | | Removal of Least Salient 50 Points | | | | Corr. ↑ |
|---|---|---|---|---|---|---|---|---|---|
| | CPD ↑ | AUPD ↑ | MPD ↑ | AUMPD ↑ | CPP ↓ | AUPP ↓ | MPP ↓ | AUMPP ↓ | |
| IG (Sundararajan et al., 2017) | 227.09±0.27 | 177.97±0.23 | 256.60±0.81 | 202.19±0.62 | 277.70±0.62 | 138.81±0.36 | 252.27±0.90 | 107.53±0.38 | 0.45±0.00 |
| DeepLIFT (Shrikumar et al., 2017) | 230.31±0.22 | 181.47±0.20 | 256.18±0.38 | 202.44±0.24 | 272.82±0.68 | 136.23±0.42 | 246.01±0.77 | 104.30±0.42 | 0.32±0.00 |
| AFO (Tonekaboni et al., 2020) | 216.81±0.49 | 171.28±0.48 | 260.44±0.56 | 203.22±0.48 | 276.04±1.46 | 145.05±1.03 | 228.50±1.23 | 98.48±0.29 | 0.45±0.00 |
| WinIT (Leung et al., 2023) | 290.84±0.41 | 207.59±0.35 | 340.70±0.77 | 238.05±0.40 | 252.09±0.64 | 130.44±0.50 | 269.08±1.12 | 141.56±0.70 | 0.27±0.00 |
| TIMING (Jang et al., 2025) | 230.00±0.57 | 179.32±0.32 | 258.60±0.55 | 203.50±0.43 | 276.34±0.92 | 137.86±0.61 | 249.23±0.72 | 106.18±0.56 | 0.44±0.00 |
| SWING | **323.55±0.25** | **253.80±0.20** | 311.35±0.92 | **246.80±0.74** | **185.96±0.45** | **69.09±0.22** | **189.32±0.58** | **70.43±0.34** | **0.62±0.00** |

| Algorithm | Removal of Most Salient 50 Points | | | | Removal of Least Salient 50 Points | | | | Corr. ↑ |
|---|---|---|---|---|---|---|---|---|---|
| | CPD ↑ | AUPD ↑ | MPD ↑ | AUMPD ↑ | CPP ↓ | AUPP ↓ | MPP ↓ | AUMPP ↓ | |
| IG (Sundararajan et al., 2017) | 12.39±0.08 | 8.26±0.05 | 15.01±0.05 | 10.36±0.03 | 31.93±0.31 | 13.25±0.12 | 27.87±0.24 | 10.14±0.07 | 0.13±0.00 |
| DeepLIFT (Shrikumar et al., 2017) | 12.61±0.07 | 8.60±0.05 | 14.88±0.05 | 10.32±0.03 | 34.14±0.34 | 13.79±0.12 | 29.75±0.29 | 10.70±0.08 | 0.16±0.00 |
| AFO (Tonekaboni et al., 2020) | 12.36±0.08 | 8.54±0.05 | 16.04±0.09 | 11.03±0.06 | 34.50±0.30 | 15.86±0.11 | 22.49±0.26 | 8.68±0.07 | 0.24±0.00 |
| WinIT (Leung et al., 2023) | 18.59±0.11 | 11.47±0.08 | 23.41±0.12 | 14.40±0.07 | 27.50±0.15 | 12.20±0.08 | 24.66±0.12 | 11.15±0.07 | 0.20±0.00 |
| TIMING (Jang et al., 2025) | 14.07±0.08 | 8.92±0.06 | 15.37±0.08 | 10.58±0.04 | 29.36±0.32 | 12.58±0.12 | 25.55±0.29 | 9.57±0.09 | 0.15±0.00 |
| SWING | **22.54±0.25** | **15.05±0.17** | 20.94±0.15 | 14.38±0.13 | **16.44±0.25** | **5.39±0.07** | **17.02±0.24** | **5.65±0.08** | **0.36±0.00** |

**Table 12:** Performance of XAI methods on the MIMIC-III dataset with an LSTM backbone, shown for window size 24 (top) and window size 72 (bottom). We evaluate by removing the most or least salient 50 points with forward-fill substitution.

| Algorithm | Removal of Most Salient 50 Points | | | | Removal of Least Salient 50 Points | | | | Corr. ↑ |
|---|---|---|---|---|---|---|---|---|---|
| | CPD ↑ | AUPD ↑ | MPD ↑ | AUMPD ↑ | CPP ↓ | AUPP ↓ | MPP ↓ | AUMPP ↓ | |
| IG (Sundararajan et al., 2017) | 11.98±2.47 | 7.84±1.59 | 13.91±3.00 | 9.40±2.04 | 24.08±6.42 | 10.69±2.64 | 20.67±6.05 | 8.35±2.36 | 0.21±0.00 |
| DeepLIFT (Shrikumar et al., 2017) | 12.40±2.67 | 8.31±1.79 | 13.93±3.07 | 9.45±2.11 | 25.33±7.01 | 11.05±2.79 | 22.36±6.85 | 8.90±2.62 | 0.24±0.00 |
| AFO (Tonekaboni et al., 2020) | 13.35±2.51 | 9.05±1.71 | 17.26±3.32 | 11.60±2.24 | 32.11±7.22 | 15.48±3.30 | 22.38±5.35 | 9.29±2.17 | 0.26±0.00 |
| WinIT (Leung et al., 2023) | 16.75±3.06 | 10.36±1.91 | 22.21±4.27 | 13.44±2.64 | 27.08±6.77 | 12.70±3.08 | 25.44±6.00 | 12.24±2.85 | 0.22±0.00 |
| TIMING (Jang et al., 2025) | 13.35±3.07 | 8.43±1.87 | 14.22±3.16 | 9.59±2.15 | 22.42±5.90 | 10.21±2.49 | 19.27±5.53 | 7.91±2.19 | 0.23±0.00 |
| SWING | **19.33±4.59** | **12.89±2.96** | 17.96±4.25 | 12.22±2.88 | **13.20±4.06** | **4.37±1.35** | **14.21±4.42** | **4.91±1.52** | **0.49±0.03** |

| Algorithm | Removal of Most Salient 50 Points | | | | Removal of Least Salient 50 Points | | | | Corr. ↑ |
|---|---|---|---|---|---|---|---|---|---|
| | CPD ↑ | AUPD ↑ | MPD ↑ | AUMPD ↑ | CPP ↓ | AUPP ↓ | MPP ↓ | AUMPP ↓ | |
| IG (Sundararajan et al., 2017) | 8.71±0.05 | 5.93±0.04 | 10.23±0.06 | 7.16±0.05 | 15.64±0.08 | 7.27±0.03 | 12.40±0.07 | 4.76±0.02 | 0.23±0.00 |
| DeepLIFT (Shrikumar et al., 2017) | 9.28±0.02 | 6.39±0.02 | 10.35±0.05 | 7.29±0.04 | 16.37±0.07 | 7.46±0.03 | 13.05±0.07 | 4.91±0.01 | 0.26±0.00 |
| AFO (Tonekaboni et al., 2020) | 8.73±0.05 | 6.05±0.04 | 10.86±0.04 | 7.56±0.04 | 16.11±0.10 | 8.09±0.04 | 10.57±0.05 | 4.17±0.02 | 0.29±0.00 |
| WinIT (Leung et al., 2023) | 11.44±0.10 | 7.22±0.06 | 14.88±0.08 | 9.17±0.07 | 12.00±0.06 | 5.77±0.02 | 11.56±0.05 | 5.53±0.03 | 0.25±0.00 |
| TIMING (Jang et al., 2025) | 9.14±0.06 | 6.08±0.05 | 10.36±0.07 | 7.24±0.05 | 14.73±0.06 | 7.00±0.03 | 11.55±0.06 | 4.51±0.02 | 0.24±0.00 |
| SWING | **13.79±0.07** | **9.46±0.05** | 13.05±0.07 | 9.10±0.06 | **7.51±0.04** | **2.45±0.02** | **7.75±0.04** | **2.56±0.02** | **0.48±0.00** |

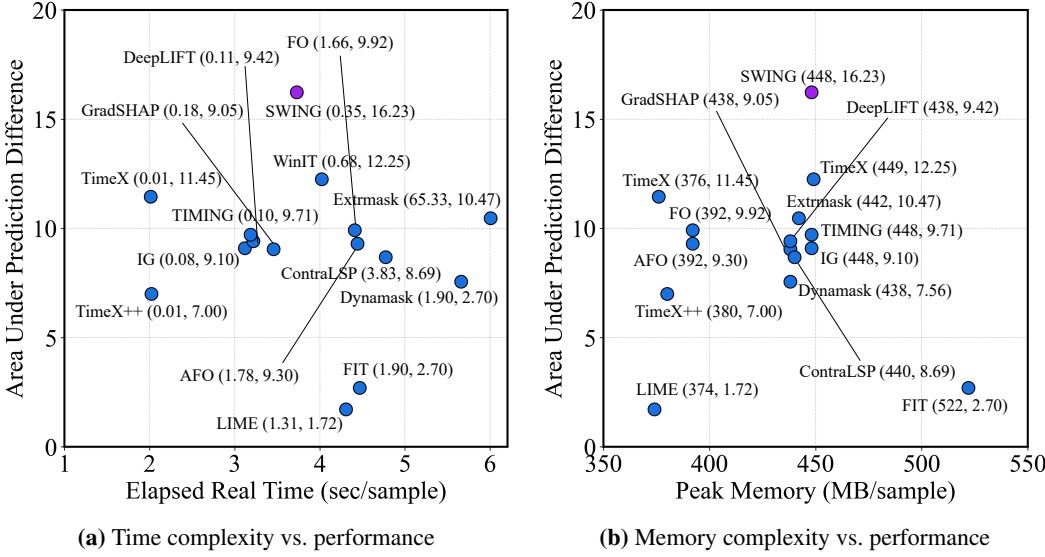

**(a)** Time complexity vs. performance  **(b)** Memory complexity vs. performance

**Figure 3:** Computational efficiency analysis comparing SWING with baselines on the MIMIC-III benchmark. (a) Elapsed real time per sample (sec/sample, log-scale) versus AUPD ($K = 50$). (b) GPU peak memory consumption per sample (MB/sample) versus AUPD ($K = 50$).

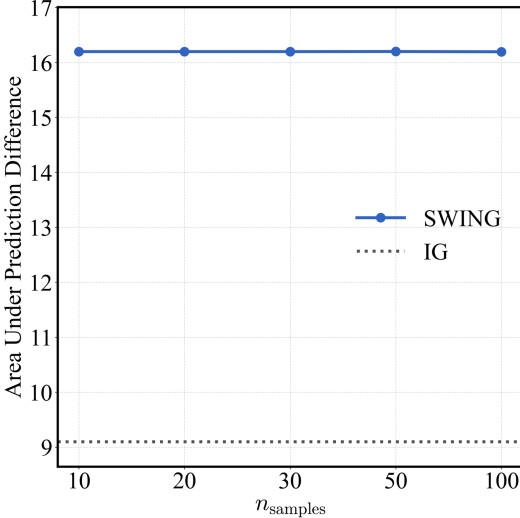

**Figure 4:** Hyperparameter sensitivity of SWING with respect to $n_{\text{samples}}$, compared to IG (dotted gray line).

**Table 13:** List of clinical features used from the MIMIC-III dataset, including their indices and descriptive names for model input.

| Index | Name | Index | Name | Index | Name | Index | Name |
|---|---|---|---|---|---|---|---|
| 0 | Height | 11 | Weight | 22 | GCS-Motor4 | 33 | GCS-Total9 |
| 1 | Hours | 12 | Blood pH | 23 | GCS-Motor5 | 34 | GCS-Total10 |
| 2 | Diastolic BP | 13 | Cap. Refill | 24 | GCS-Total0 | 35 | GCS-Total11 |
| 3 | $FiO_2$ | 14 | GCS-Eye0 | 25 | GCS-Total1 | 36 | GCS-Total12 |
| 4 | Glucose | 15 | GCS-Eye1 | 26 | GCS-Total2 | 37 | GCS-Verbal0 |
| 5 | Heart Rate | 16 | GCS-Eye2 | 27 | GCS-Total3 | 38 | GCS-Verbal1 |
| 6 | Mean BP | 17 | GCS-Eye3 | 28 | GCS-Total4 | 39 | GCS-Verbal2 |
| 7 | $SpO_2$ | 18 | GCS-Motor0 | 29 | GCS-Total5 | 40 | GCS-Verbal3 |
| 8 | Respiratory Rate | 19 | GCS-Motor1 | 30 | GCS-Total6 | 41 | GCS-Verbal4 |
| 9 | Systolic BP | 20 | GCS-Motor2 | 31 | GCS-Total7 | | |
| 10 | Body Temperature | 21 | GCS-Motor3 | 32 | GCS-Total8 | | |

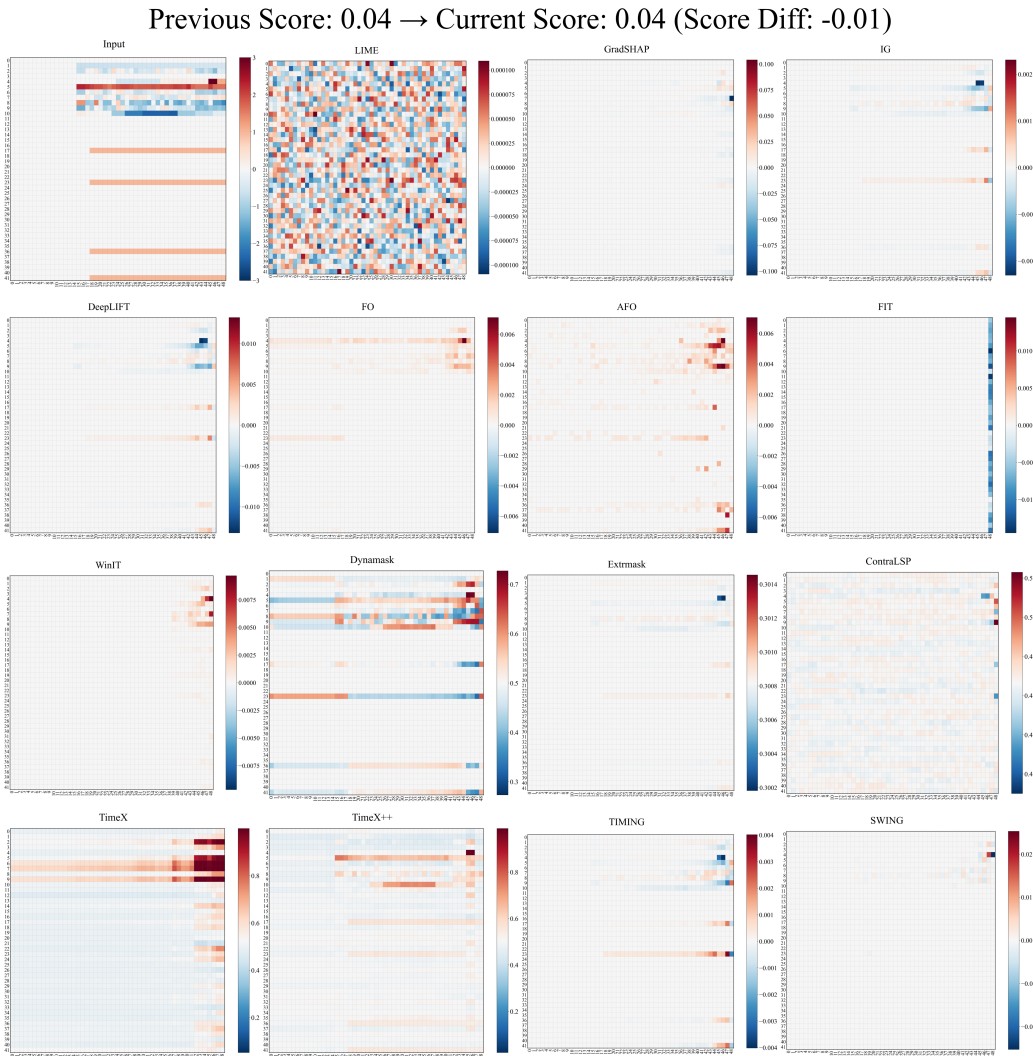

Previous Score: 0.04 → Current Score: 0.04 (Score Diff: -0.01)

**Figure 5:** Qualitative case study showing attributions extracted with XAI methods on MIMIC-III (Johnson et al., 2016) using LSTM (Hochreiter & Schmidhuber, 1997) backbone with $T_1 = 47$ and $T_2 = 48$, *i.e.*, $T_2 - T_1 = 1$. The uppermost-left heatmap displays the normalized input features, while the remaining fifteen panels illustrate the attribution heatmaps generated by each XAI method under the Delta-XAI framework, reflecting their respective explanations of the score changes.

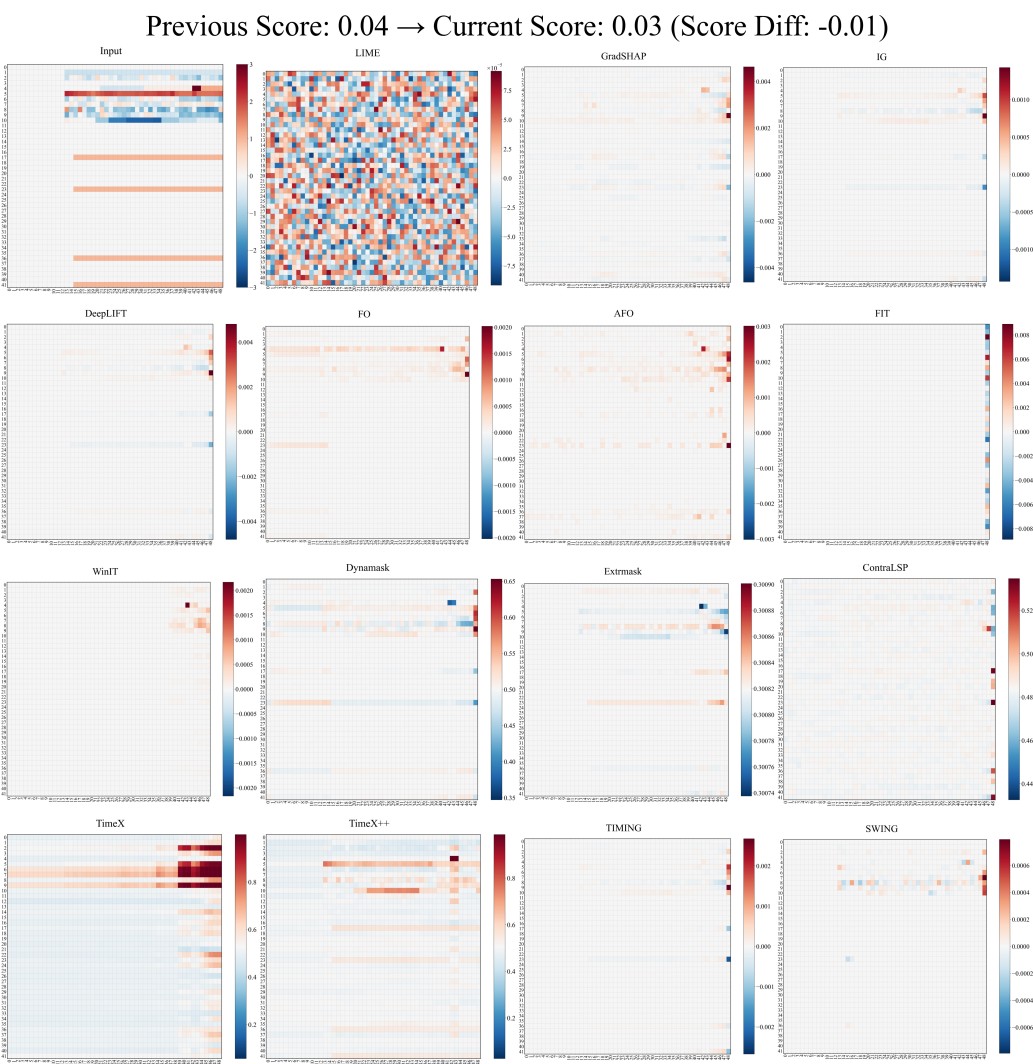

**Figure 6:** Qualitative case study showing attributions extracted with XAI methods on MIMIC-III (Johnson et al., 2016) using LSTM (Hochreiter & Schmidhuber, 1997) backbone with $T_1 = 47$ and $T_2 = 48$, *i.e.*, $T_2 - T_1 = 1$. The uppermost-left heatmap displays the normalized input features, while the remaining fifteen panels illustrate the attribution heatmaps generated by each XAI method under the Delta-XAI framework, reflecting their respective explanations of the score changes.

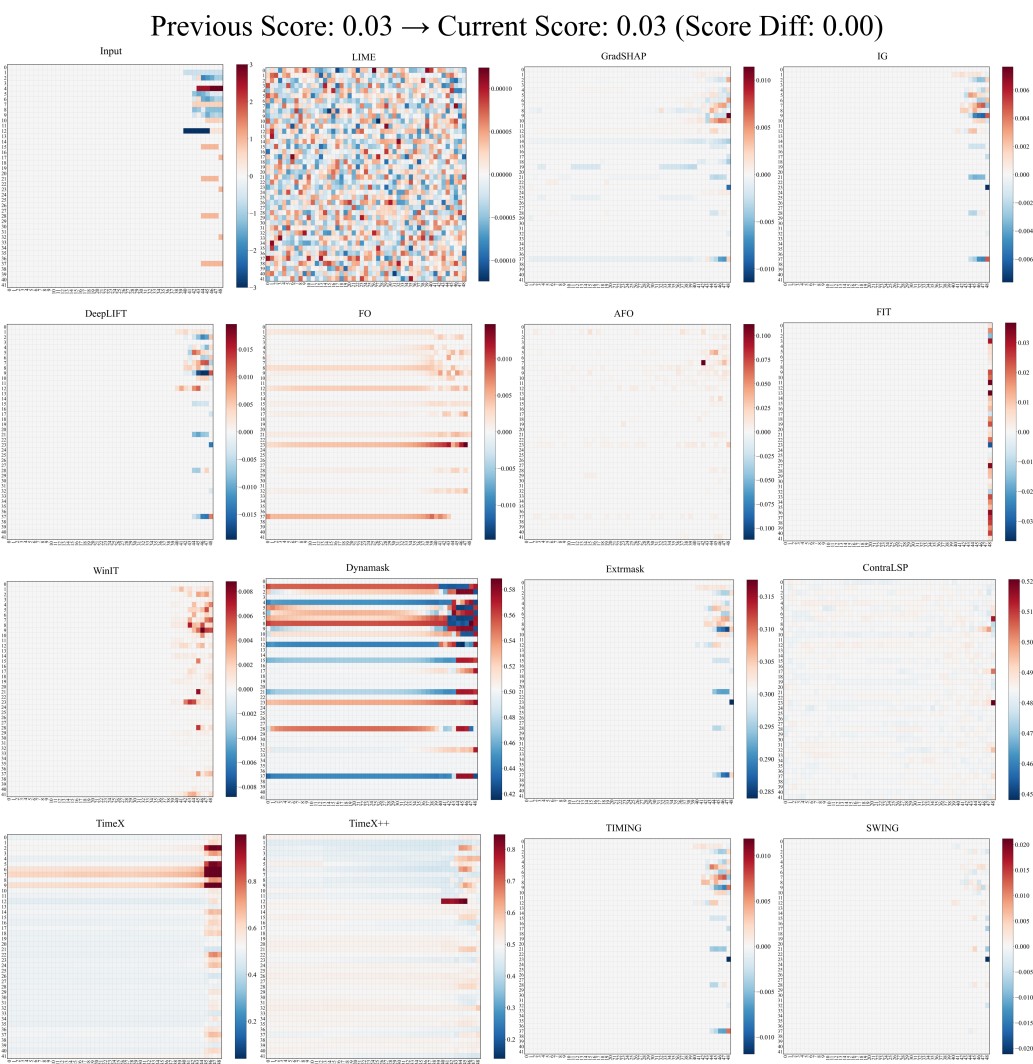

**Figure 7:** Qualitative case study showing attributions extracted with XAI methods on MIMIC-III (Johnson et al., 2016) using LSTM (Hochreiter & Schmidhuber, 1997) backbone with $T_1 = 47$ and $T_2 = 48$, *i.e.*, $T_2 - T_1 = 1$. The uppermost-left heatmap displays the normalized input features, while the remaining fifteen panels illustrate the attribution heatmaps generated by each XAI method under the Delta-XAI framework, reflecting their respective explanations of the score changes.

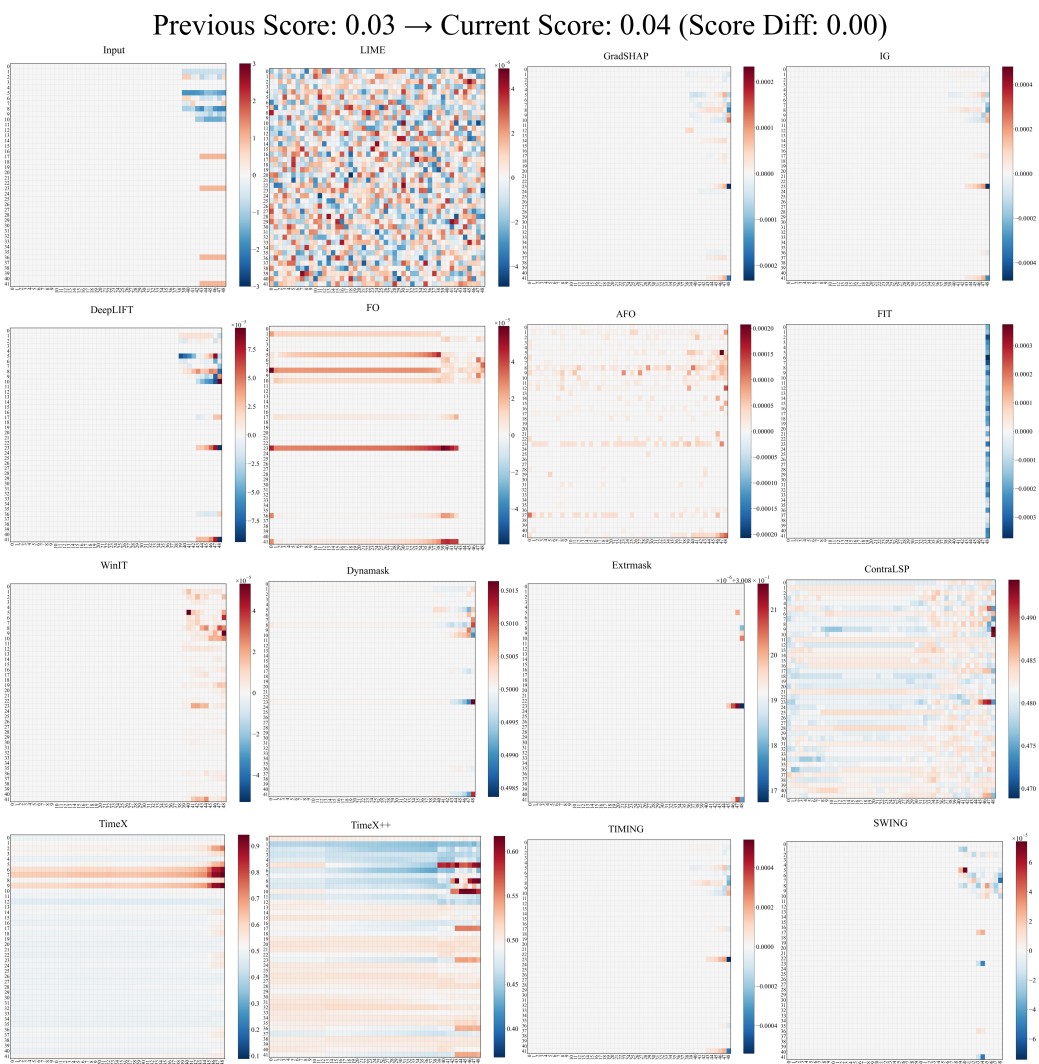

**Figure 8:** Qualitative case study showing attributions extracted with XAI methods on MIMIC-III (Johnson et al., 2016) using LSTM (Hochreiter & Schmidhuber, 1997) backbone with $T_1 = 47$ and $T_2 = 48$, *i.e.*, $T_2 - T_1 = 1$. The uppermost-left heatmap displays the normalized input features, while the remaining fifteen panels illustrate the attribution heatmaps generated by each XAI method under the Delta-XAI framework, reflecting their respective explanations of the score changes.

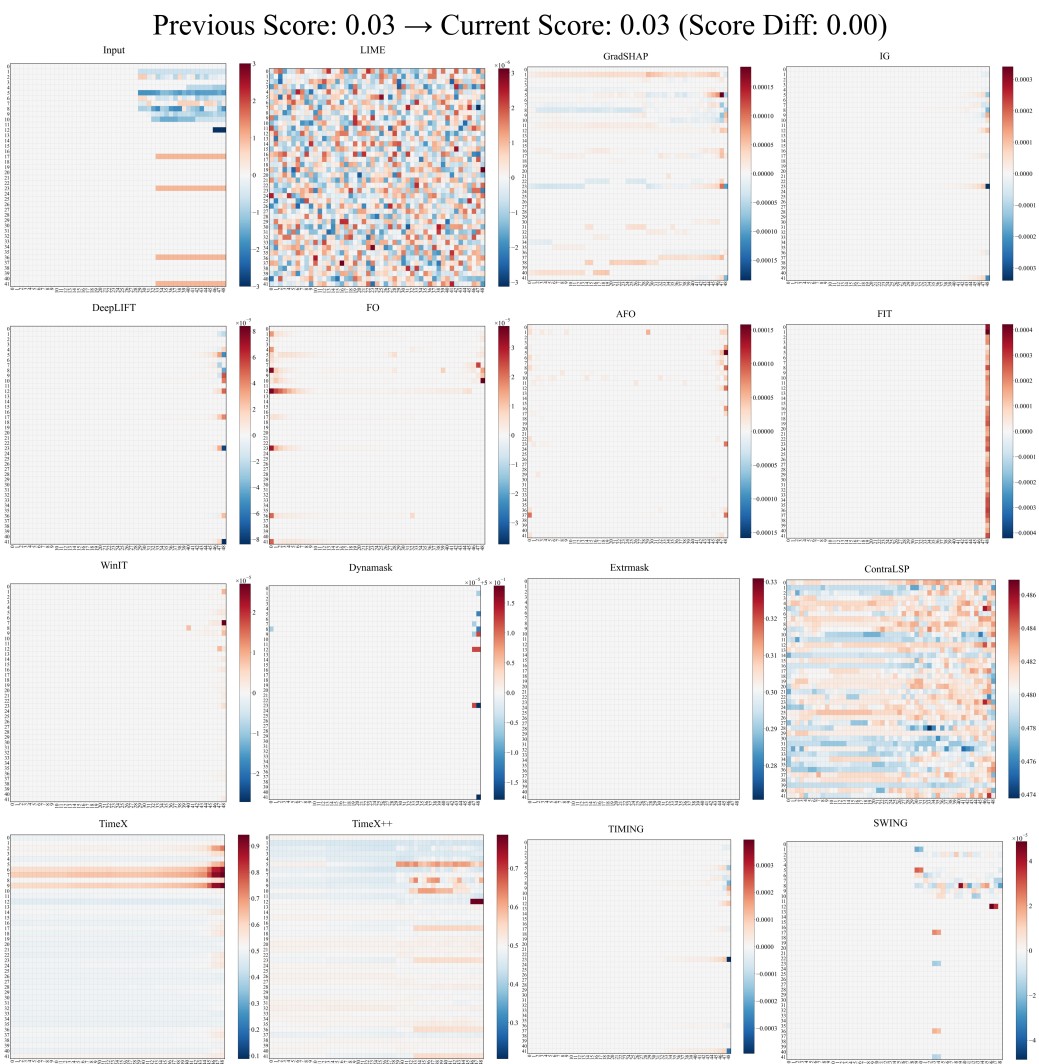

**Figure 9:** Qualitative case study showing attributions extracted with XAI methods on MIMIC-III (Johnson et al., 2016) using LSTM (Hochreiter & Schmidhuber, 1997) backbone with $T_1 = 47$ and $T_2 = 48$, *i.e.*, $T_2 - T_1 = 1$. The uppermost-left heatmap displays the normalized input features, while the remaining fifteen panels illustrate the attribution heatmaps generated by each XAI method under the Delta-XAI framework, reflecting their respective explanations of the score changes.

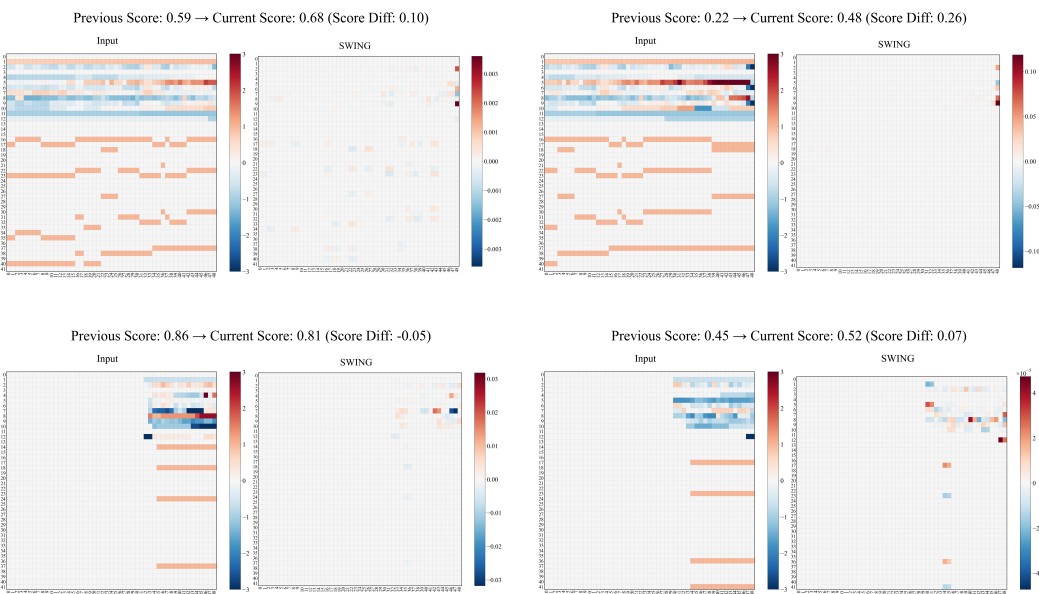

**Figure 10:** Qualitative case study for coherence analysis on decompensation risk. Each subfigure (a-d) shows two heatmaps for a MIMIC-III (Johnson et al., 2016) sample processed with a LSTM (Hochreiter & Schmidhuber, 1997) backbone: the left heatmap visualizes normalized input features, and the right heatmap displays SWING's feature attributions. These panels reveal the clinically relevant features and temporal patterns that SWING identifies as most influential for the observed changes in decompensation risk score.

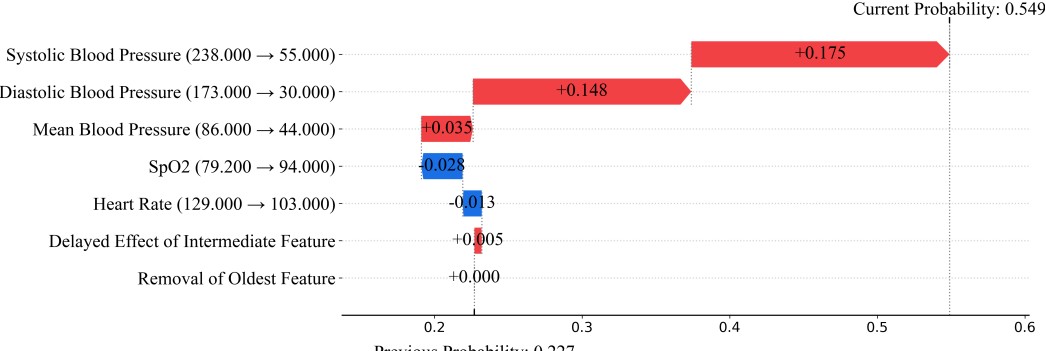

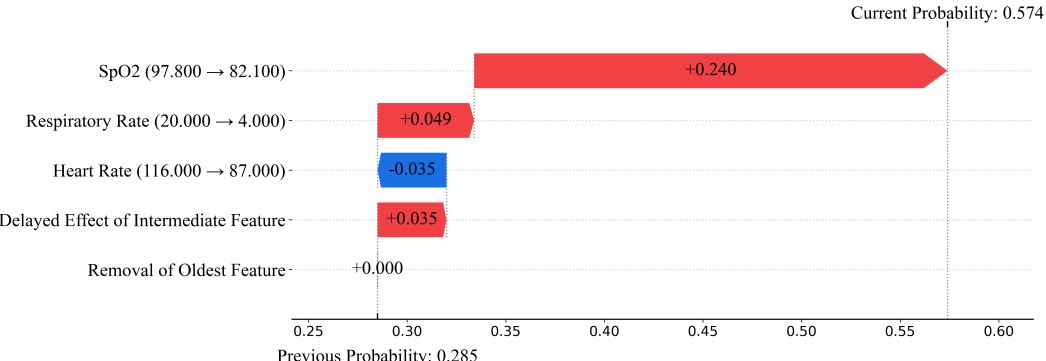

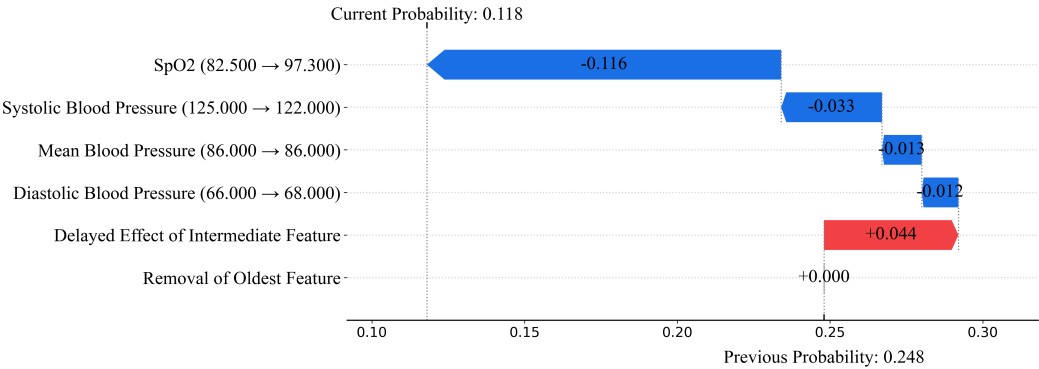

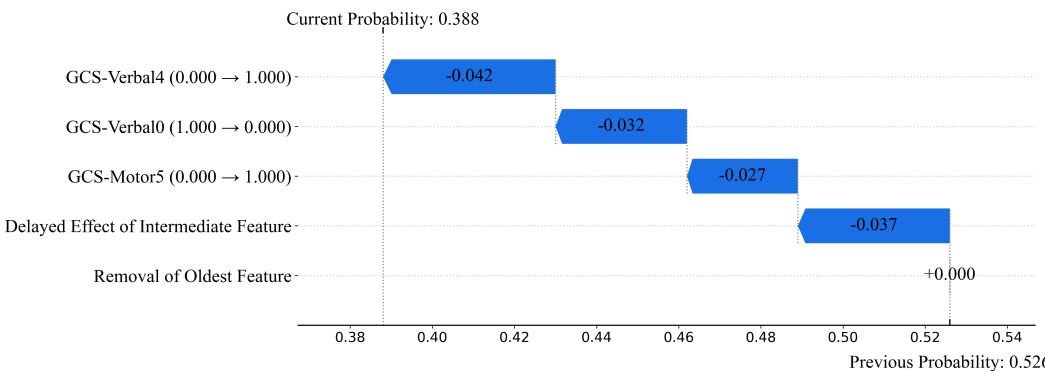

**Figure 11:** Qualitative case study for coherence analysis on decompensation risk. Each subfigure indicates a bar plot for a MIMIC-III (Johnson et al., 2016) sample processed with a LSTM (Hochreiter & Schmidhuber, 1997) backbone. Bars for individual features represent the attribution contributed by newly observed inputs at the current time step, while the 'Delayed Effect of Intermediate Features' and 'Removal of Oldest Features' summarize the aggregated attributions of their respective feature groups. These panels provide an intuitive understanding of how each variable contributes to the observed change in decompensation risk score and illustrate the clinical plausibility of SWING's explanations.

