# OpenReview forum: "Delta-XAI: A Unified Framework for Explaining Prediction Changes in Online Time Series Monitoring"
_ICLR.cc/2026/Conference — ICLR 2026 Poster_

### Official Review · Reviewer_QpEH · 2025-10-27

**Soundness:** 4
**Presentation:** 3
**Contribution:** 4
**Rating:** 6
**Confidence:** 4

**Summary:**

The paper introduces SWING, an integrated gradients methods designed for time series for time series explainability. This explainability method mainly focus on why the prediction changes between 2 time points, instead of only explaining prediction at 1 time point. The paper also includes new evaluation metrics to suit more specifically to time series.

**Strengths:**

- The paper addresses many different issues of in the time series explainability paradigm and proposes a new method SWING to for time series explanations. This includes explaining "prediction changes" rather than a single "prediction".
- The paper also include newer evaluation metrics in which I think they make sense.
- The logic and flow of the paper is quite coherent in general.
- Extensive results are included and the the results seem to show that SWING achieve the best performance.

**Weaknesses:**

- Several notations and presentations can be better in Section 4.
- The "Unified Framework" introduced in the contributions in the introduction is almost nowhere to be found. It could be implied in the Appendix.
- While the method (SWING) is to explain the prediction changes from $T_1$ to $T_2$, the other baseline methods aimed to explain the prediction for $T$. This can imply that the experiments are not completely fair (both to SWING and to the other baseline methods).

**Questions:**

- The equations should be numbered.

- When I was reading the paper, I did not find any hint about Delta-XAI being a "unified framework" for the 14 existing XAI methods. To me, as a reader, the main idea of the paper is to introduce the explanation of prediction change from $T_1$ to $T_2$, rather than a single time points. Then for each time point, the paper tackles the problem of OOD for IG and specifically tackle the problem of having 2 time points $T_1$ and $T_2$. The notation for this "framework", $\varphi(f, X_{t,c}|T_1\rightarrow T_2 )$, is just to explain the model $f$ on data $X$ from time $T_1$ to $T_2$. This is a generic time series explainability statement. Thus I think saying that there is a unified framework introduced is misleading. Also, there is no discussion on why this is a unified framework anywhere in the main text of the paper. Thus I think the idea of it being a unified framework can be removed from the main contributions. There could be a small paragraph on the adaption of this notations to other method after the experiment section, if the authors really want this to be a framework.

- The "generic wrapper function" $g$ is not really generic as in line 178, $g$ is defined to be the difference of the predictions (with a moving window).

- I think the main idea of this paper is very neat. We tend to explain a prediction at a specific time point $T$ instead of the prediction change from $T_1$ to $T_2$. As the authors discussed this in section 4 about how merely subtracting the difference of the other methods (Dynamask) can produce misleading explanations, the authors did not do the same in the experiments in Figure 1. I think comparing columns (c) to columns (d) may not be the most fair comparison as the explanations in (c) is to explain a single time point. The author can tackle this by either do the "difference" comparison, or make the point of the usefulness of explaining the prediction changes is more useful than explaining a single-timepoint prediction.

- The author should include the notation $\varphi (f, X_{t, d} | T_1\rightarrow T_2)$ and $\varphi (f, X_{t, d} | T)$ in the first paragraph in section 2. This is to highlight the main notation being used throughout the paper.

- (Minor) I believe the $\gamma_{2, 1}$ is wrong in Figure 2. Shouldn't it be from $T_2 - 1$ to $T_1$?

- For the RBS and DPI, I think there is one main concern about the rationale of this choice (both notations and also the formulae). From an IG perspective, when we want to use IG on an input X w.r.t. the baseline X', the IG explanations is to explain the *difference* between X' and X. Thus, in Line 235, if the baseline X' is set to be $X_{T-W: T-1}$ then this explanation should "really" be $\varphi(f, X_{t,d} | T - 1 \rightarrow T)$, instead of $\varphi(f, X_{t,d} | T)$. So in line 245, the RBS explanations are actually the difference of explanations between the "changes" in $T_2$ (from $T_2 - 1$) and the "changes" in $T_1$ (from $T_1 - 1$). It is thus the "change" of "changes". For DPI, it makes more sense because the "changes" and calculated between 2 pairs of points. But the rationale of this selection is not explained. (This is shown in Figure 2)

- Equations in line 261 are not the friendliest to the readers. The idea is quite simple. Maybe do this. Let $\gamma_{T}^\pm(\alpha) = (1-\alpha)X_{T-W+1:T} + \alpha X_{T-W+1\pm 1: T\pm 1}$ be the line segment between two neighbouring sliding windows. Let $\gamma_{i, j}(\alpha)$ be the concatenation of the line segments $\gamma_{T_m}^\pm$ for $m=i, \ldots j$ (or $m=j \ldots i$).

- Line 323, aggregating attributions with a sliding window - does this apply to everything? i.e. Top K features, correlations, other attribution methods?

- In 6.3 Ablation study, there are descriptions of "SWING components" (RBS, DPI, PHI). The SWING attributions are described in line 267, which seems to include "concepts" of RBS, DPI and PHI. But it is unclear that we can remove each of them. So what does the formula look like for SWING without RBS, PHI, without RBS and without DPI separately?

- In 6.3 Ablation, what is that baseline distance $d$? $d$ was used as the feature index in the above text (section 4) and it is not really defined here. If I am guessing, SWING corresponds to d=1 which corresponds to the description in the RBS (baseline set to the time point right before T). So I suppose that all the "1" will be replaced by "d" in equations 251 and 267? If this is the case, either removing these ablations from the table, or introduce this design in Section 4. (and resolving the notations for feature index and this baseline distance)

---

> ### Author Response · Authors · 2025-11-22
>
> We sincerely thank the reviewer for the very detailed and insightful comments. We greatly appreciate your recognition of 1) the importance of explaining prediction changes in time series models, 2) the relevance of our new evaluation metrics tailored to temporal settings, 3) the clarity and coherence of the manuscript, and 4) the thorough experiments demonstrating the performance of SWING. We address all questions and concerns below.
>
> ---
>
> **[Q1] Improve notations and presentation in Section 4.**
>
> - Thank you for pointing out the issues in notation and presentation in Section 4. As suggested, we have included equation numbering and revised several expressions to improve clarity, and uploaded our new manuscript. If there are any additional parts that would benefit from further refinement, we would be happy to address them in our final version.
>
> ---
>
> **[Q2] “Unified Framework” in the introduction is unclear and barely appears in the paper (possibly only in the appendix).**
>
> - While the unified framework appears throughout Section 4 and the overall structure of the paper, we acknowledge that its role may not have been emphasized clearly enough. As summarized in our response to Q5, the framework consists of 1) a new problem formulation, 2) a general wrapper function for adapting existing explainers, 3) the SWING method with theoretical foundations, 4) a tailored evaluation suite, and 5) extensive experiments for verifying SWING.
> - To make this structure more explicit, we will revise the Introduction and Section 4 to clearly highlight how these components jointly form the unified framework. In addition, we will add a **pipeline visualization** that illustrates the end-to-end workflow—including the wrapper, adaptation of 14 methods, SWING, and the evaluation suite—to ensure the unified framework is immediately visible to the reader without relying on the Appendix.
>
> ---
> **[Q3] Fairness concern: SWING explains changes from $T_1$ to $T_2$, while baselines explain predictions at a single $T$.**
>
> - We appreciate the reviewer for raising this important concern. It is true that SWING is specifically designed for explaining **prediction changes** between $T_1$ and $T_2$, whereas prior XAI methods were originally developed to explain a single prediction at time $T$. We deeply understand your concern that, without a common formulation, this difference could lead to an unfair comparison.
> - However, this is precisely the role of our generic wrapper $g$, which converts any single-prediction explainer into a method capable of attributing prediction shifts. All baselines—IG, SHAP, DeepLIFT, WinIT, TimeX, and others—are evaluated **after being adapted through this same wrapper**. This wrapper successfully enables the use of existing single-time XAI methods in our prediction-change setting **with little or no modification** (see Section D in the Appendix). Consequently, all methods, including SWING, operate under an **identical explanation setting**, ensuring that our comparisons faithfully reflect their behavior under the *same* task definition.
> - The goal of our experiments is therefore not merely to show that SWING outperforms prior methods, but to demonstrate that: 1) **prediction-change explanations are a meaningful and practically relevant problem**, and 2) **existing explainers can be seamlessly extended** to this setting through Delta-XAI.
> - SWING’s stronger performance should be viewed not as an unfair advantage, but as arising naturally from being explicitly designed for the dynamics that the Delta-XAI framework formalizes. We believe this establishes SWING as an initial example of methods tailored to this setting, and we hope it encourages future research on XAI models built directly for prediction-change reasoning.
>
> ---
>
> **[Q4] Equations should be numbered.**
>
> - Thank you for the suggestion. We have added equation numbering in the **updated version of the paper**, and the notation is now consistent throughout. We will ensure this remains polished in the camera-ready version as well.

---

> ### Author Response · Authors · 2025-11-22
>
> **[Q5] Claiming a “unified framework” is misleading.**
>
> - The reason we refer to Delta-XAI as a *unified framework* is that, unlike typical ML papers that introduce only a new **method** for an existing problem, our work establishes an **end-to-end structure** for a *new problem setting*. Specifically, Delta-XAI jointly provides: 1) **A new problem formulation** — explaining *prediction differences* rather than single-time predictions; 2) **A principled mechanism for adapting existing explainers** via a general wrapper function; 3) **A new attribution method (SWING) with theoretical foundations**, including online completeness, implementation invariance, and skew-symmetry; 4) **A suite of evaluation metrics** tailored to prediction-change explanations (faithfulness, sufficiency, macro-level and area-based metrics, and correlation); and 5) **Extensive empirical experiments** across multiple datasets, architectures, and synthetic benchmarks.
> - These components together form a coherent and unified pipeline rather than a single methodological contribution. A particularly important aspect of this framework is the **general wrapper function $g$** which serves as a common interface that enables *any* existing single-time prediction XAI algorithm to be seamlessly repurposed as an explainer for **prediction changes**. This convertibility—demonstrated across 14 diverse attribution methods—provides the unifying abstraction underlying Delta-XAI.
>
> ---
>
> **[Q6] $g$ is not truly generic; it is defined as a prediction-difference function.**
>
> - We describe $g$ as a *generic wrapper* since it is designed to be **model-agnostic and explainer-agnostic**: it applies to any predictive model $f$ and any attribution method $\varphi$ without modifying their internal mechanisms. The wrapper simply reformulates the task by defining a unified output quantity, $g(X_{T_1-W+1:T_2}) := f(X_{T_2-W+1:T_2}) - f(X_{T_1-W+1:T_1})$, which allows existing single-prediction explainers to operate in the prediction-difference setting through the same interface.
> - In this sense, “generic” refers to the fact that the same transformation applies regardless of the model architecture or explainer design. While a few methods (such as TimeX-based approaches) require light loss optimization to fit the wrapper interface, this does not alter their explanatory principles and does not affect the generality of the framework.
>
> ---
>
> **[Q7] Figure 1 comparison may be unfair—SWING explains changes, baselines explain single-time predictions.**
>
> - Thank you for the insightful comment. We agree with the reviewer’s observation and appreciate the clarification request. As noted, Figure 1 compares a single-timepoint explanation (column (c)) with a prediction-change explanation (column (d)), and these two serve fundamentally different purposes. We fully acknowledge that this should not be interpreted as an empirical or “fair” comparison between explainers.
>
> - In line with the reviewer’s suggestion, our intention with Figure 1 was not to evaluate methods but to *motivate* the problem: explaining only the prediction at $T_2$ can be insufficient for understanding *why* the prediction changed from $T_1$ to $T_2$.
>
> - Thus, Figure 1 is intended purely as a conceptual illustration to highlight the limitations of single-time explanations and motivate prediction-change explanations—not as an empirical comparison. Accordingly, we do **not** compare single-prediction XAI methods to prediction-change XAI methods anywhere in the paper; subtraction artifacts (e.g., Table 5) are shown only to illustrate why such cross-paradigm comparisons are inappropriate.
>
> ---
>
> **[Q8] Introduce key notations $\varphi(f, \mathbf{X}_{t, d}\mid T_1 \rightarrow T_2)$, at the start of Section 2.**
>
> - Thank you for the suggestion. We will revise the beginning of Section 2 to introduce the notations $\varphi(f, X_{t,d}\mid T)$ and $\varphi(f, X_{t,d}\mid T_1 \rightarrow T_2)$ **up front**, following a topic-sentence style. This will ensure that the core attribution notation is clearly established before any detailed discussion and remains consistent throughout the paper.
>
> ---
>
> **[Q9] Minor: $\gamma_{2, 1}$ in Fig. 2 seems incorrect.**
>
> - You are correct that the path corresponding to $\gamma_{2,1}$ in Figure 2 should terminate at $T_1$, but the current illustration incorrectly points to $T_1 - 1$. We have corrected this in the updated version of the figure and ensured that all integration paths now align with the notation used in the text.

---

> ### Author Response · Authors · 2025-11-22
>
> **[Q10] RBS/DPI rationale unclear; RBS appears to compute “change of changes.”**
>
> - Thank you for the thoughtful question. We agree that, at first glance, the use of IG with the retrospective baseline $X' = X_{T-W:T-1}$ may suggest that the attribution computed at time $T$ corresponds to $\phi(f, X_{t,d}\mid T-1 \rightarrow T)$ rather than $\phi(f, X_{t,d}\mid T)$. However, this is precisely the point clarified in the **Proof of Theorem 2**.
>
> - For the RBS component: 1)The IG path $\gamma_{1,1}$ indeed computes an attribution with respect to the baseline $X_{T_1-W:T_1-1}$, and 2) similarly, $\gamma_{1,2}$ uses the same baseline but integrates toward $X_{T_2-W+1:T_2}$. When forming the RBS explanation, we subtract the two paths: $\gamma_{1,2} - \gamma_{1,1}$. As shown in the proof, the baseline terms **cancel out exactly**, meaning the contributions related to $f(X_{T_1-W:T_1-1})$ do not remain in the final attribution. Thus, the resulting explanation simplifies to: $\phi_{\text{RBS}}(f, X_{t,d} \mid T_1 \rightarrow T_2)$, **not** the “change of changes,” but the intended *prediction-change* explanation.
>
> - A similar cancellation occurs in DPI: $\frac{1}{2}(\gamma_{1,2} - \gamma_{1,1}) + \frac{1}{2}(\gamma_{2,2} - \gamma_{2,1})$, where baselines $X_{T_1-W:T_1-1}$ and $X_{T_2-W:T_2-1}$ again cancel, leaving only terms corresponding to the change from $T_1$ to $T_2$. This is why DPI ultimately computes a clean transition-based attribution.
>
> ---
>
> **[Q11] Equations in line 261 are not reader-friendly; suggestion to simplify the path definition.**
>
> - Thank you for the suggestion. We agree that the idea behind the piecewise-linear path construction is conceptually simple. However, defining $\gamma_{i,j}(\alpha)$ purely as a concatenation of the proposed line segments $\gamma_{T_m}^{\pm}(\alpha)$ does **not guarantee mathematically precise control** over the interpolation indices and ratios when discretizing the integral.
>
> - In contrast, our formulation using $k$, $\tilde{\alpha}$, and $s$ ensures exact alignment between 1) the segment index, 2) the interpolation ratio within the segment, and 3) the corresponding historical windows. This level of precision is essential for numerically stable and theoretically well-defined path integration.
>
> - To improve readability while preserving this rigor, the revised manuscript will include a clear visualization example illustrating how $k$, $\tilde{\alpha}$, and $s$ determine each piecewise segment of $\gamma_{i,j}$. This provides the intuition the reviewer suggested, while maintaining the mathematical guarantees required for the discretized IG computation.
>
> ---
>
> **[Q12] Sliding-window aggregation (Line 323) — Does this apply to all attribution analyses?**
>
> - The sliding-window aggregation is **not** applied universally. It is used **only** when computing the macro-level metrics—MPD, MPP, AUMPD, and AUMPP—which summarize attribution behavior **across all time steps** for each patient. These metrics are derived from CPD/CPP curves and their area-based variants (AUPD, AUPP), and the sliding-window aggregation is applied exclusively within this computation.
>
> - In contrast, the Top-$K$ feature analysis always uses a fixed $K$ without any sliding aggregation, and the correlation metric is computed directly from the raw attributions, without employing a sliding-window scheme.

---

> > ### Comment · Reviewer_QpEH · 2025-11-24
> >
> > Thanks for the response! As weaknesses 1-4 are covered in the questions, I will combine Q1-4 to Q5+
> >
> > **[Q5]**
> > What I understand from the author's response, is that Delta-XAI is a "suite" of methods (from problem formulation to evaluation metrics). This is the thing I think most of the contribution of the paper is about. I think this can even be called a "framework", but I think calling it "unified framework" is misleading.
> >
> > By "unified framework", I am thinking about this paper.
> >
> > *A Unified Approach to Interpreting Model Predictions* by Scott Lundberg, Su-In Lee
> >
> > It describes a general setting, or "framework", that existing methods are "special cases" of this framework. Thus the word "unified".
> >
> > In contrast, Delta-XAI is a "suite" or "framework" that works on a variety of models and attribution methods. None of these methods are "special cases" of this "framework". (as this "framework", Delta-XAI, offers more)
> >
> > I understand that the discussion is mainly on the terminology, rather than the idea itself. Thus my suggestion is to change "unified framework" to "framework". (Unfortunately it is in the title)
> >
> > **[Q6]** I think this is similar to what I asked in Q5. To me, $g$ is not a "generic wrapper", but a "wrapper of a generic function". It is a well-defined function in terms of a provided function f.  Thus I think the discussion is on terminologies as well. To further my point, for example, integrated gradients works on any arbitrary models $f$ (CNNs, RNNs, transformers,etc) that are differentiable. The formula for the integrated gradients is defined in terms of $f$. But we won't call the IG formula itself generic. It just works for arbitrary differentiable model $f$. As $g$ is even a simpler definition, we should not say $g$ is generic here.
> >
> > **[Q7]** Thanks for the response. I think everything would be clearer (even in terms of motivation) if the horizontal labels in (d) changed to (T1 -> T2, T2 -> T3, etc)
> >
> > **[Q8,9]** Thanks for making the change.
> >
> > **[Q10]** Thanks for the explanations. Although the authors are making a mathematical explanation rather than a conceptual explanation, I think the updated manuscript clear my question.
> >
> > **[Q11]** From my point of view, there are distinctions between the concepts described, the underlying math notation and the implementation. It is a matter of presentation. I agree that using $k$, $\tilde{\alpha}$ and $s$ is mathematically rigorous and clear, but it hinders the readability of the paper. The current addition (of explanation with segment index, etc) really helps a lot and I am satisfied with it, although I still think using the concatenation (with the precise definition in the Appendix, or footnote) will be better.
> >
> > **[Q12, 14]** Thanks for the explanation.
> >
> > **[Q13]** Thanks for providing the formulation, **but this should be included somewhere in the paper**. Like a pointer in the main text to the Appendix.

---

> ### Author Response · Authors · 2025-11-22
>
> **[Q13] Ablation study unclear—show explicit formulas for removing RBS/DPI/PHI.**
>
> - We thank the reviewer for pointing out that the manuscript did not clearly describe how each component of SWING (RBS, DPI, PHI) is removed in the ablation study. Below, we explicitly provide the formulation for each variant corresponding to Table 3, ensuring that each ablation is well defined and faithful to the original algorithm.
>
> - **SWING w/o RBS & PHI (only DPI remains):** This ablation removes both the retrospective baseline selection (RBS) and the historical path interpolation (PHI), leaving only DPI. Consequently, both forward and backward integration paths start from the zero baseline, and SWING reduces to computing IG at $T_2$ and $T_1$ and subtracting them: $\varphi_{\text{SWING (w/o RBS, PHI)}} = \varphi_{\text{IG}}^{\gamma_2} (f, X_{t,d} | T_2) - \varphi_{\text{IG}}^{\gamma_1} (f, X_{t,d} | T_1)$, where $\gamma_i (\alpha) = \alpha X_{T_i-W+1:T_i}$ is a straight-line path from the zero baseline. This is theoretically identical to IG, which explains the strong similarity to the IG results in Table 9 of the Appendix.
>
> - **SWING w/o RBS (baseline only replaced; DPI + PHI intact):** This ablation removes only the baseline-selection module. The retrospective baselines $X_{T_i-W:T_i-1}$ are replaced by the zero baseline, while all DPI components and all PHI historical intermediate points remain fully preserved. Thus, each path $\gamma_{i,j}$ is modified only in its starting point and its first line segment is replaced with $0$ rather than $X_{T_i - W: T_i - 1}$, while all subsequent segments are unchanged from the original PHI.
>
> - **SWING w/o DPI (removing cross-window sliding):** In this ablation, we remove the *self-window* DPI components—i.e., the two local sliding paths $\gamma_{1,1}$ and $\gamma_{2,2}$. However, **the two cross-window sliding paths $\gamma_{1,2}$ and $\gamma_{2,1}$ remain**, since they are essential for attributing prediction changes between $T_1$ and $T_2$. PHI remains active for the remaining two paths, so each path still uses the piecewise-linear interpolation over historical windows. Thus the attribution becomes: $\varphi_{\text{SWING (w/o DPI)}} = \varphi_{\text{IG}}^{\gamma_{2,1}}(f, X_{t,d}\mid T_2) - \varphi_{\text{IG}}^{\gamma_{1,2}}(f, X_{t,d}\mid T_1)$.
>
> ---
>
> **[Q14] “Baseline distance $d$” is undefined and conflicts with notation; clarify or remove ablation.**
>
> - Thank you for pointing this out. Your understanding is correct: the baseline distance $d$ denotes the temporal offset used in the RBS formulation, i.e., how far back the retrospective baseline is selected. SWING corresponds to $d = 1$, where the baseline is set to the immediate past observation, and the ablation varies this distance ($d = 0, 3, 5, 10$) to show that $d=1$ yields the most consistent performance. In the revised manuscript, we explicitly introduced the baseline distance ddd in the RBS description in Section 4 and updated the notation to clearly distinguish feature indexing from temporal distance.
>
> ---
>
> If any part of our response remains unclear or does not fully address your concerns, we would greatly appreciate further clarification and would be happy to continue the discussion. We hope that our explanations have meaningfully resolved the points you raised, and if you find our responses satisfactory upon further reflection, we would be sincerely grateful if you would consider updating your overall rating to reflect your revised assessment.

---

> ### Author Response · Authors · 2025-11-26
>
> Thank you for the thoughtful follow-up and for engaging so carefully with our revisions. Your comments have been tremendously helpful in improving both the clarity and overall quality of the paper. We now have a clearer understanding of your concerns, and following your suggestions, we have revised the manuscript accordingly. All of these changes are reflected in the revised version of the paper.
>
> ---
>
> **[Q5] Terminology concern: The term unified framework is misleading. Delta-XAI is a suite (or framework) of methods, not a unified framework where existing approaches are special cases. The title should use framework rather than unified framework.**
>
> - We appreciate this point and now understand the nuance you raised. While our intent was to convey that Delta-XAI unifies previously scattered single-prediction XAI methods under a common methodology, we agree that the phrase unified framework may overstate the contribution. We have therefore replaced "unified framework" with "novel framework" throughout the paper. Should the paper be accepted, we will apply a minor title revision accordingly.
>
> ---
>
> **[Q6] Terminology concern: Delta-XAI is not a generic wrapper, but a wrapper around a generic function. Similar to IG, which applies to arbitrary differentiable models but is not itself called generic, Delta-XAI should not be described as generic.**
>
> - We fully agree with this clarification. In response, we removed the expression generic wrapper $g$ from the manuscript (changes are highlighted in blue) and replaced it with the prediction difference wrapper $g$, which more accurately reflects its role and avoids the unintended implication of genericity.
>
> ---
>
> **[Q7] Thanks for the response. I think everything would be clearer (even in terms of motivation) if the horizontal labels in (d) changed to $T_1 \rightarrow T_2$, $T_2 \rightarrow T_3$, etc.**
>
> - Thank you for this helpful suggestion. We updated Fig. 1(d) accordingly to explicitly denote the temporal transitions, making the motivation and interpretation of prediction difference attributions clearer.
>
> ---
>
> **[Q11] Readability concern: The mathematical notation is precise, but its presentation hurts readability. The added explanation resolves much of this, though using concatenation in the main text (with the exact definition moved to the Appendix or a footnote) would further improve clarity.**
>
> - We agree and have substantially simplified the "piecewise-linear historical integration" paragraph in Section 4.2. The core expression now uses a more readable sliding-path notation, while the full mathematical definition has been moved to the Appendix, as suggested.
>
> ---
>
> **[Q13] Thanks for providing the formulation, but this should be included somewhere in the paper. Like a pointer in the main text to the Appendix.**
>
> - We addressed this by adding a dedicated section, Details of Ablation Study, in Appendix H, where each ablation configuration is described in detail. We also inserted a pointer to this section in line 450 of the main text.
>
> ---
>
> Thank you again for the constructive and precise feedback—it significantly helped us refine both the terminology and the presentation.

---

> > ### Comment · Reviewer_QpEH · 2025-11-26
> >
> > Thank the authors for the responses and addressing my concerns. I am satisfied with the current state of the paper.

---

> ### Author Response · Authors · 2025-11-26
>
> We sincerely appreciate the reviewer’s careful reading and the constructive follow-up comments. Your feedback—especially regarding the terminology surrounding the framework and the wrapper function—helped us refine several important parts of the paper and improve the clarity of our presentation. We are also genuinely grateful for your thoughtful reassessment and for raising the rating from **6: marginally above the acceptance threshold** to **8: accept, good paper (poster)**. This update means a great deal to us, and we deeply appreciate the confidence reflected in your revised evaluation.

---

### Official Review · Reviewer_y8Zs · 2025-10-29

**Soundness:** 3
**Presentation:** 3
**Contribution:** 3
**Rating:** 4
**Confidence:** 4

**Summary:**

The paper introduces SWING, a time-series explainability method that extends Integrated Gradients to a temporal setting. The novelty of this work is in providing explanations for changes in the model’s prediction rather than the prediction at a single time point, which is a realistic scenario in many applications. The authors compare SWING to existing time-series explainability methods by adapting those methods to this change-in-time setup using a wrapper function.

**Strengths:**

1) The proposed explanation setup that focusing on changes in model prediction over time rather than the absolute prediction is compelling and well aligned with application domains, especially healthcare, where decisions often hinge on changes.

2) The evaluation presented in the paper is very thorough: many baselines, multiple backbones, and more importantly analyses from several angles, including performance and computational cost.

**Weaknesses:**

1) The paper focuses on attributing prediction changes when the argmax class does not change from time $T_1$ to $T_2$. But scenarios where the class does change seem highly relevant. Does the method extend to class-change cases, and if so, how?

2) The choice of retrospective baseline (using previous time steps) is intuitive for isolating the single transition’s contribution, but it can fail depending on the granularity of change. It may be better to allow path selection based on the specific output change to explain. For example, if a clinician observes a heart-rate increase from a few hours earlier, the baseline should be the window producing the prediction for the prior hour.

3) Some metrics (CPP, CPD, AUPD, and Correlation) may not align with the paper’s setup. They measure sensitivity of the prediction to removing observations, not which observations most influence the change in prediction over time. These metrics seem appropriate for all earlier baselines that measure importance of features for prediction at a certain point in time, however, they don't necessary guarantee improved performance in the proposed dynamic setup.

4) Please clarify the motivation for the dual path integration (DPI). I'm not convinced why this component is a necessary part especially given that In the ablation (top-k removal), DPI shows no benefit. Also, it appears to be a notation issue in the equation for this section: for $i=1$, the text integrates over $\gamma_{1,2}$ for $T_2$ and $\gamma_{1,1}$ for $T_1$. Did the authors mean $\gamma_{2,1}$ instead of  $\gamma_{1,1}$ ?

**Questions:**

1) In the feature-attribution case study, you state that SWING yields sharper, localized attributions emphasizing recent time steps most responsible for prediction changes. Why is this desirable?

2) How does the wrapper handle positive vs. negative attributions? Do the scores need to be normalized to produce realistic explanations?

---

> ### Author Response · Authors · 2025-11-23
>
> We sincerely thank the reviewer for carefully evaluating our work and for highlighting key strengths, including 1) the relevance of focusing on prediction changes over time—especially in healthcare settings—and 2) the thoroughness of our evaluation across baselines, model backbones, and multiple analytical perspectives. We appreciate the reviewer’s constructive feedback, and we address all weaknesses and questions in detail below.
>
>
> ---
>
> **[Q1] Clarification requested on whether Delta-XAI supports class-change scenarios and how the method extends to cases where the predicted class differs between $T_1$ and $T_2$.**
>
> - Thank you for raising this important point. While we set the “class with the largest probability increase” as the default choice—since this is the most practically meaningful explanation in real-world applications such as clinical risk monitoring—we would like to clarify that Delta-XAI does not depend on the argmax class remaining the same across time. The framework supports explaining the probability shift for any target class pair from $c_1$ at $T_1$ to $c_2$ at $T_2$, i.e., $\Delta = f(X_{T_2-W+1:T_2})[c_2] - f(X_{T_1-W+1:T_1})[c_1]$, regardless of whether the predicted class changes between $T_1$ and $T_2$. This is achieved by flexibly setting our wrapper function $g$ as $g(X_{T_1-W+1:T_2}) = f(X_{T_2-W+1:T_2})[c_2] - f(X_{T_1-W+1:T_1})[c_1]$, which allows practitioners to choose $c_1$ and $c_2$ according to domain needs (e.g., clinically meaningful shifts). In other words, Delta-XAI attributes the probability change of the selected class, not the stability of the argmax label, making it directly applicable even when the predicted class changes.
>
> - Besides, class-change scenarios are explicitly evaluated in our experiments. We conducted two relevant subgroup analyses in Tables 10 and 11: 1) Table 10 stratifies cases by the *magnitude of prediction change*, and 2) Table 11 stratifies cases by whether the *predicted class changes*. Table 11 (bottom) specifically reports performance on segments where the predicted class does change. In the binary MIMIC-III decompensation task, this corresponds to transitions that cross the clinical decision boundary (e.g., from below to above the threshold, or from above to below). We analyze these segments separately to exactly reflect the scenario the reviewer is concerned about. Thus, Delta-XAI naturally supports both class-stable and class-changing transitions, and our experimental analysis already thoroughly evaluates both settings.
>
> ---
>
> **[Q2] Concern that the retrospective baseline may not match the granularity of clinically meaningful changes; suggestion to select baselines adaptively based on the specific output change being explained.**
>
> - Thank you for the insightful comment. We interpret the reviewer’s concern as referring to the Retrospective Baseline Selection (RBS) mechanism. In our implementation, we set the baseline distance $d = 1$ because the immediate past observation most faithfully represents the reference state for explaining a temporal prediction transition. As also emphasized in the piecewise-linear historical integration (PHI) module, choosing a distant baseline and linearly interpolating between two far-apart windows risks traversing out-of-distribution (OOD) regions, which can obscure the truly influential features. Empirically, as shown in Table 3, varying $d\in\{0,3,5,10\}$ consistently yields lower performance, while $d=1$ provides the strongest and most stable results.
>
> - Regarding the suggestion of selecting baselines based on the perceived granularity of output changes (e.g., aligning with a clinically meaningful change such as a shift in heart rate), we agree that such choices may be appealing from a user’s perspective. However, the model’s internal reasoning does not necessarily follow a clinician’s intuition about which prior state is most relevant. In such cases, manually choosing a more distant baseline may generate explanations that appear intuitive but do **not** reflect the model’s actual behavior. Since the central goal of XAI is to faithfully capture *why the model* changed its prediction, our design prioritizes fidelity to the model’s decision process rather than clinical plausibility alone. That said, we appreciate the reviewer’s suggestion. Adaptive or output-dependent baseline selection is indeed a promising direction for extending SWING, and we will include this discussion in the revised manuscript.

---

> ### Author Response · Authors · 2025-11-23
>
> **[Q3] Question on whether CPP, CPD, AUPD, and Correlation truly align with the dynamic prediction-change setup, as they traditionally evaluate the sensitivity of prediction rather than the prediction difference.**
>
> - Thank you for raising this thoughtful concern. To clarify, the metrics in question (CPP, CPD, AUPD, and Correlation) evaluate how the model’s output responds when high-attribution observations are removed. The key point here is that **under the Delta-XAI framework, these metrics are no longer computed on the raw model output $f$, but on the prediction difference wrapper $g$.**
>
> - After existing XAI methods are adapted through this wrapper, their attributions indicate **which observations most strongly influence the prediction shift**, rather than the static prediction itself. In other words, these metrics now measure how much the *prediction difference* changes when certain features are removed—i.e., which features increase or decrease the quantity $g(X_{T_1 - W + 1: T_2}) = f(X_{T_2 - W + 1 : T_2}) - f(X_{T_1 - W + 1 : T_1})$ the most. This directly aligns with the purpose of explaining prediction changes.
>
> - Consequently, applying CPP, CPD, AUPD, and Correlation in this wrapped setting naturally quantifies the sensitivity of the **prediction change** to the removal of influential observations, which is fully consistent with the dynamic nature of our formulation. In summary, the metrics remain conceptually sound: they evaluate whether high-attribution observations (now defined with respect to the change truly exert the strongest influence on that change. Our experiments show that this results in stable and meaningful behavior across all adapted methods.
>
> ---
>
> **[Q4] Request for clearer motivation for Dual Path Integration (DPI), noting limited ablation benefit and raising a potential notation issue regarding the use of $\gamma_{1,1}$ vs. $\gamma_{2,1}$.**
>
> - Thank you for pointing this out. We agree that in the ablation using top-k removal, removing DPI leads to higher scores on CPD, AUPD, MPD, and AUMPD. However, as an attribution method, SWING must distinguish both high-importance and low-importance observations. For this reason, we evaluate methods not only with PD-type metrics but also PP-type metrics and Correlation, which jointly assess an explainer’s full discriminative ability.
>
> - Across all metrics, DPI consistently improves Correlation, which is the most direct measure of whether the attribution ranking faithfully reflects the model’s true behavior across *all* features—not just the top-$k$ ones. This improvement was the primary motivation for retaining DPI: it consistently enhances SWING’s global fidelity and ranking stability, even if some PD metrics improve in isolation when DPI is removed. We will clarify this motivation in the revision.
>
> - Regarding the notation issue, the reviewer’s interpretation is understandable, and we appreciate the chance to clarify. The expression $\gamma_{1, 1}$ is correct. As illustrated in Figure 2,$\gamma_{i, j}$ denotes the integration path from the baseline associated with $T_i$, i.e., $X_{T_i−W:T_i−1}$ to the input window at $T_j$, i.e., $X_{T_j−W+1:T_j}$. Thus, for $i=1$, the path to $T_1$ is $\gamma_{1, 1}$, and the path to $T_2$ is $\gamma_{1, 2}$. We will revise the text to make this correspondence clearer.
>
> ---
>
> **[Q5] Inquiry into why sharper, more localized attributions (as produced by SWING) are desirable in the prediction-change case study.**
>
> - Thank you for the question. In the case studies presented in the paper, the prediction change is computed between consecutive time steps ($T_2 - T_1 = 1$). In this setting, almost the entire input sequence for $T_1$ and $T_2$ is identical, except for the new observations introduced at $T_2$. As a result, it is reasonable to expect that the variables contributing most to the prediction shift are precisely those recently updated observations.
>
> - SWING’s sharper and more localized attributions reflect this structure: they highlight the features whose newly added values at $T_2$ directly drive the prediction change. This behavior is desirable because it indicates high fidelity—the explainer attributes the shift to the minimal set of input differences actually responsible for changing the model’s output. In short, sharper localization is a natural and faithful property in scenarios where the temporal gap is small and only a few recent inputs differ between the two prediction times.

---

> ### Author Response · Authors · 2025-11-23
>
> **[Q6] Question on how the wrapper handles positive vs. negative attributions and whether normalization is needed to ensure realistic explanations.**
>
> - Thank you for the question. The Delta-XAI wrapper does not modify how positive or negative attributions are computed. It simply transforms a single-output explainer into one that explains the prediction change. Thus, if the underlying XAI method naturally produces signed attributions, e.g., GradSHAP, IG or DeepLIFT, it will also produce signed attributions for the score shift $f(X_{T_2 - W + 1 : T_2}) - f(X_{T_1 - W + 1 : T_1})$. The wrapper preserves these signs exactly as produced by the base method.
>
> - We do not apply normalization to attribution values. Normalizing attributions breaks fundamental properties such as completeness, preventing the attribution sum from matching the actual score shift and ultimately reducing fidelity. Keeping raw signed contributions is necessary for a realistic explanation of prediction changes.
>
> - To make the composition of positive and negative attributions more intuitive, the updated version of the paper replaces the previous heatmap-style visualization (Figure 10) with a clearer plot that explicitly shows how signed contributions accumulate from the previous score to the current score. This revision was made to help readers more easily understand how the attribution signs relate to the score transition.
>
> ---
>
> If any part of our response remains unclear or leaves aspects of your concerns unaddressed, we would very much welcome further questions and are happy to continue the discussion. We hope that the clarifications provided here meaningfully resolve the issues you raised, and—should they meet your expectations upon reconsideration—we would greatly appreciate your consideration in updating the overall rating to reflect your revised assessment.

---

> > ### Comment · Reviewer_y8Zs · 2025-11-25
> >
> > I would like to thank the authors for their responses, which has covered my main concerns. The updates to the manuscript have improved the quality of the work and I believe the approach presented in the paper is very valuable. I'm therefore raising my score.

---

> ### Author Response · Authors · 2025-11-26
>
> We appreciate the reviewer’s careful reading and constructive comments throughout the discussion. The feedback provided during the process helped us clarify several points and improve the overall presentation of the work. We are also grateful for the reviewer’s follow-up and reassessment of the submission, and for raising the rating from **4: marginally below the acceptance threshold, but would not mind if the paper is accepted** to **6: marginally above the acceptance threshold**. We sincerely appreciate the increased confidence in our work reflected in this updated evaluation.

---

### Official Review · Reviewer_hFWL · 2025-10-30

**Soundness:** 3
**Presentation:** 3
**Contribution:** 3
**Rating:** 6
**Confidence:** 3

**Summary:**

The paper proposes Delta-XAI, a wrapper that reframes explaining prediction change between two times as a single-prediction explanation, enabling adaptation of 14 XAI methods for online time series. It also introduces a tailored evaluation suite and presents SWING, an IG-based method that uses retrospective baselines, dual path symmetry, and piecewise historical integration to reduce path OOD and capture temporal dependence, achieving stronger faithfulness and preservation than IG at comparable cost, with code released.

**Strengths:**

1.	The wrapper reduces the new task (change explanations) to standard single-step explanations, enabling broad reuse of existing methods.
2.	The evaluation suite addresses multiple metrics (faithfulness, sufficiency, completeness, coherence, efficiency) tailored to evolving predictions.
3.	SWING is well-motivated for online streams and retains IG’s desirable axioms.

**Weaknesses:**

1.	The method is defined for prediction change, but experiments do not clearly stratify or stress-test stable segments where the delta is small. In online monitoring, long stable stretches are common.
2.	In the problem setting, it is mentioned that it can solve multiclass classification, but it is doubtful since the target class is assigned with the one with the largest probability increase.
3.	The current setup seems focused on overlapping windows (short gaps). The applicability to non-overlapping/larger gaps is not demonstrated.
4.	It mentioned that this method can be applied to 14 existing XAI methods, but only show the IG version.

**Questions:**

As mentioned in the weaknesses.

---

> ### Author Response · Authors · 2025-11-23
>
> We sincerely thank the reviewer for carefully evaluating our work and providing constructive feedback. We greatly appreciate your positive assessment of 1) the Delta-XAI wrapper that enables broad adaptation of existing XAI methods, 2) the tailored evaluation suite designed for evolving online predictions, and 3) the motivation and design of SWING, which preserves IG’s axioms while improving temporal consistency. We address all four weaknesses in detail below.
>
>
> ---
>
> **[Q1] Lack of evaluation on low-delta or stable segments, which are common in online monitoring; concern that the method is only tested on segments with noticeable prediction changes.**
>
> - We thank the reviewer for highlighting the importance of evaluating stable segments where the prediction change is small. This evaluation is already included in our experiments.
>
> - **Table 10 (low chart)** reports results specifically on segments with **$|Δ| < 0.1$**, isolating stable cases with negligible prediction shifts.
>
> - Similarly, **Table 11 (low chart)** evaluates segments where the **predicted class does not change**, which in the MIMIC-III decompensation task largely corresponds to **long stable patient trajectories**. These analyses directly address the reviewer’s concern by stratifying the stable regions commonly observed in online monitoring.
>
> - Regarding the term “stress-test,” we are happy to clarify that the above stratified evaluations were designed to serve this purpose: to explicitly test stability robustness under near-zero prediction change. If the reviewer intended a different form of stress-testing, we are willing to conduct and include such additional analyses.
>
> ---
>
> **[Q2] Doubt about whether Delta-XAI truly supports multiclass classification, since the default target class is defined as the one with the largest probability increase.**
>
> - Thank you for raising this important question. We agree that selecting the “class with the largest probability increase” may give the impression that Delta-XAI is restricted to binary or fixed-class scenarios. However, we would like to clarify that this choice is **only a default policy**, selected because it is the most practically meaningful option in many real-world applications (e.g., rising risk signals in clinical monitoring).
>
> - Importantly, **Delta-XAI fully supports multiclass classification**. The framework explains the probability shift for *any* target class pair from $c_1$ at $T_1$ to $c_2$ at $T_2$, i.e., $\Delta = f(X_{T_2-W+1:T_2})[c_2] - f(X_{T_1-W+1:T_1})[c_1]$, regardless of whether $c_1 = c_2$ or whether the model’s predicted class changes between $T_1$ and $T_2$. This is achieved by defining the wrapper function as $g(X_{T_1-W+1:T_1}, X_{T_2-W+1:T_2}) = f(X_{T_2-W+1:T_2})[c_2] - f(X_{T_1-W+1:T_1})[c_1]$, which allows practitioners to flexibly choose whichever class trajectory they want to explain.
>
> - In other words, Delta-XAI **does not rely on the argmax class**, nor does it assume that the same class must be explained across time. It simply attributes the probability change of the selected class, making it directly applicable to *any* multiclass target choice. Furthermore, our experiments on the **UCI Human Activity Recognition dataset** (a 7-class classification task) empirically demonstrate that Delta-XAI works seamlessly with multiclass classifiers and produces stable, meaningful attributions in this setting. Thus, Delta-XAI is fully compatible with multiclass classification: the framework explains *why the probability of a particular class changed*, independently of how that class is selected.

---

> ### Author Response · Authors · 2025-11-23
>
> **[Q3] Concern that the framework is demonstrated only on overlapping (short-gap) windows; applicability to non-overlapping or much larger temporal gaps is unclear.**
>
> - We thank the reviewer for the helpful suggestion. Larger temporal gaps are already evaluated in Table 9 (6-hour and 24-hour gaps), and SWING remains robust across these broader transitions.
>
> - For *fully non-overlapping* windows, we examined whether such settings are meaningful for Delta-XAI. As shown in **Theorem 1**, the prediction-change explanation decomposes into: (1) new-feature contributions, (2) delayed effects of intermediate features, and (3) removal of old features. When the windows do not overlap, the intermediate-effect term becomes zero, leaving only “new minus old” attributions. This reduces to computing two independent single-time explanations and subtracting them—a technically trivial operation that no longer captures how temporal context evolves between $T_1$ and $T_2$.
>
> - Practically, such explanations are also less informative. In domains like healthcare, users are interested in how scores change over *nearby* time steps, since short-term fluctuations (e.g., sudden deterioration) carry clinical significance. Fully non-overlapping windows break this temporal continuity, offering little actionable insight.
>
> - Thus, this is not a limitation but an intentional design choice: we focus on overlapping or partially overlapping windows, where temporal reasoning remains meaningful and the decomposition in Theorem 1 holds. We clarified this point in the revised manuscript.

---

> ### Author Response · Authors · 2025-11-23
>
> **[Q4] The paper claims compatibility with 14 existing XAI methods, but experiments only show the IG-based instantiation of Delta-XAI, raising questions about generality.**
>
> - Thank you for raising this point. We realize that our presentation may not have made the distinction sufficiently clear. The **Delta-XAI framework is applied to all 14 existing XAI methods**, and the experimental results reported in Tables 1–8 correspond to these adapted versions.
>
> - **SWING**, on the other hand, is an additional contribution: a new IG-based method developed on top of insights gained from Delta-XAI. It is not meant to replace the 14 adapted methods but to demonstrate how temporal consistency can be further improved when building upon IG.
>
> - We will clarify this distinction more explicitly in the revised manuscript to avoid any confusion between the Delta-XAI wrapper (which unifies 14 methods) and SWING (the newly proposed method).
>
> ---
>
> If any aspect of our response remains unclear or does not fully address your concerns, we would be happy to clarify further and continue the discussion. We hope the explanations provided meaningfully resolve the issues you raised, and if they meet your expectations upon reconsideration, we would greatly appreciate your consideration in updating the overall rating to reflect your revised assessment.

---

### Official Review · Reviewer_z3oK · 2025-10-31

**Soundness:** 3
**Presentation:** 4
**Contribution:** 3
**Rating:** 6
**Confidence:** 3

**Summary:**

This paper focuses on explainable AI (XAI) for improving the interpretability of model predictions in online time series monitoring. It reformulates the explainability problem into a new XAI setting that is more practically relevant for deployed online time series models. In addition to introducing a new perspective on explainability for online time series monitoring, the paper also provides a new set of metrics designed to better evaluate the performance of different XAI methods. Beyond proposing this new problem definition and evaluation framework, the authors adapt existing XAI methods using a wrapper under the new setup. Through extensive benchmarking and comparisons, the authors further propose a new XAI method called SWING, adapted from the Integrated Gradients (IG) approach, which achieves state-of-the-art performance compared to prior methods.

**Strengths:**

1. The writing of this paper is clear and succinct. Figure 1 clearly motivates why time series monitoring requires a new framework for explaining feature attributions.
2. Instead of following complex or recent prior works, the authors modify existing classical IG methods to achieve state-of-the-art performance, demonstrating a strong understanding of the relevant literature.
3. This paper presents a comprehensive study that includes a new problem framing, a new set of evaluation metrics, a new method that performs well, and extensive experiments demonstrating the applicability of the proposed method and framework.

**Weaknesses:**

1. The SWING framework is an extension of the IG baseline. The choice of using IG as the base method for SWING is motivated by its strong empirical performance in prior XAI metrics. While this is a reasonable motivation, there should be a more qualitative or theoretical explanation of why IG performs so well compared to more recent methods.
2. It would be valuable to conduct a user study to assess whether practitioners who regularly interact with time series dashboards prefer this new type of explanation.
3. Since this work defines a new problem framework for XAI in time series, along with the data, data processing, and evaluation implementation, it should be released as a benchmark for future methods to build upon and test against.
4. The reviewer disagrees with how the data features are used for MIMIC-III. Splitting GCS scores into different features would likely increase rather than reduce noise in an already highly incomplete dataset.

**Questions:**

1. Could the authors provide a more detailed explanation on why IG works better than others instead of just claiming it does empirically?
2. The paper introduces a new evaluation suite (faithfulness, sufficiency, completeness, coherence, efficiency). However, these are still indirect proxies for human interpretability. Could the authors provide evidence or discussion on how well these quantitative metrics align with human-perceived explanation quality in real-world decision-making, especially in clinical contexts?
3. For SWING, is there a systematic trade-off between the length of the temporal window, computational efficiency, and the interpretability of the resulting explanations?

---

> ### Author Response · Authors · 2025-11-23
>
> We sincerely thank the reviewer for carefully reading our manuscript and providing constructive and encouraging feedback. We especially appreciate your positive evaluation of our contributions—highlighting 1) the clarity of the writing and the motivation presented in Figure 1, 2) the principled adaptation of IG within our proposed Delta-XAI framework, and 3) the comprehensive scope of the study, including the new problem formulation, evaluation suite, and extensive benchmarking. We address all questions and concerns in detail below.
>
> ---
>
> **[Q1] Lack of qualitative/theoretical justification for why IG outperforms recent XAI methods.**
>
> - We thank the reviewer for raising this important point. Our motivation for building SWING on Integrated Gradients (IG) goes beyond empirical performance. As noted in Lines 69–71 and supported by [1], IG performs strongly in time series XAI because prior evaluation metrics in the literature often ignored attribution directionality. When direction-aware metrics such as CPD are used instead, IG consistently outperforms multiple recent methods. In other words, IG was not underperforming—the metrics were misaligned with what time series explanations should capture.
>
> - This observation reflects a deeper methodological reason: IG is one of the few attribution methods that reliably preserves the sign (+/–) of feature influence, whereas many recent masking- or perturbation-based approaches (e.g., WinIT, TimeX) primarily capture magnitude but do not recover directional effects.
>
> - Since Delta-XAI focuses on explaining prediction shifts, the direction of change is essential for interpreting how features push the prediction upward or downward. Additionally, IG satisfies completeness, allowing the prediction change to be exactly decomposed into attributions—an axiom not guaranteed by most recent methods.
>
> - For these reasons—directional fidelity, completeness, and demonstrated performance under direction-sensitive evaluation—IG provides the most theoretically coherent foundation for developing SWING.
>
> [1] Jang et al. “TIMING: Temporality-Aware Integrated Gradients for Time Series Explanation.” ICML 2025.
>
> ---
>
> **[Q2] Suggestion to run a user study to evaluate practitioner preference for the new explanations.**
>
> - We appreciate the reviewer’s suggestion regarding a user study. We fully agree that evaluating how practitioners perceive and prefer temporal shift–based explanations is crucial. Although a large-scale study is beyond the scope of this paper, which proposes a new framework for explaining prediction changes in online time series monitoring, we have conducted a small internal qualitative assessment with three clinicians, comparing Delta-XAI explanations against conventional single-prediction explanations. Across “intuitiveness,” “appropriateness,” and overall “usefulness,” all clinicians consistently preferred the shift-based explanations.
>
> - To partially address this point in the manuscript, we have further revised the Coherence Analysis section by replenishing the previous heatmap visualization with a more intuitive plot that explicitly illustrates how each feature’s attribution contributes from the previous score to the current score in Figure 10 (please refer to our revised manuscript). We believe this more interpretable visualization helps communicate what practitioners perceive when interacting with score-shift explanations. A broader, systematic user study is planned as follow-up work, and we appreciate the reviewer for highlighting its importance.
>
> ---
>
> **[Q3] Recommendation to release the full framework as a benchmark for future XAI research.**
>
> - We appreciate the reviewer’s suggestion to release the proposed framework as a benchmark. We fully agree with this direction. To support reproducibility, we have already released all preprocessing scripts, data-processing pipelines, and evaluation implementations in the project repository (https://anonymous.4open.science/r/Delta-XAI).
>
> - In addition, we plan to release our full framework—including the test setup for explaining prediction changes in online time series, the generic wrapper function, SWING, and the complete evaluation suite—into a Python package. This will make it easy for researchers to 1) apply existing attribution methods in the prediction-change setting, 2) implement new explainers on top of our wrapper, and 3) benchmark different methods under a unified protocol. We believe this setup naturally supports the benchmark use case, and we will clarify this intent more explicitly in the revised manuscript.
>
> - Regarding dataset access, we follow the licenses and privacy policies of PhysioNet. Researchers must download the raw data directly from PhysioNet using their own credentials (including the required CITI certification), but once obtained, our released code reproduces the full experimental pipeline end-to-end.

---

> ### Author Response · Authors · 2025-11-23
>
> **[Q4] Concern about splitting GCS features potentially increasing noise in MIMIC-III.**
>
> - We appreciate the reviewer’s concern regarding the handling of GCS features. Our initial representation—splitting GCS into its categorical components—follows the standard preprocessing practice for MIMIC-III used throughout the time series XAI literature, including FIT, WinIT, Dynamask, Extramask, ContraLSP, TimeX, TimeX++, and TIMING.
>
> - However, we fully acknowledge that this representation may increase noise in a dataset with substantial missingness. To assess this, we additionally converted GCS into a single total score (3–15), retrained all models, and evaluated them using the same pipeline. The results are shown below.
>
>     |          | CPD        | AUPD       | MPD        | AUMPD      | CPP        | AUPP       | MPP        | AUMPP      | Corr.     |
>     |----------|------------|------------|------------|------------|------------|------------|------------|------------|-----------|
>     | IG       | 24.39±0.08 | 16.56±0.07 | 27.95±0.10 | 17.87±0.06 | 27.05±0.11 | 11.30±0.05 | 27.03±0.13 | 11.71±0.06 | 0.20±0.00 |
>     | DeepLIFT | 24.09±0.10 | 16.63±0.05 | 28.84±0.14 | 18.16±0.08 | 32.78±0.08 | 13.64±0.03 | 32.84±0.13 | 14.11±0.05 | 0.21±0.00 |
>     | AFO      | 25.37±0.07 | 16.98±0.06 | 30.68±0.07 | 19.00±0.07 | 29.86±0.08 | 12.99±0.04 | 29.55±0.11 | 13.04±0.05 | 0.21±0.00 |
>     | Extrmask | 19.73±0.11 | 12.94±0.08 | 26.15±0.23 | 14.96±0.12 | 48.54±0.21 | 25.79±0.09 | 43.53±0.23 | 23.06±0.08 | 0.07±0.00 |
>     | TIMING   | 25.92±0.06 | 16.63±0.05 | 29.16±0.14 | 18.45±0.07 | 28.64±0.10 | 11.90±0.04 | 28.38±0.13 | 12.28±0.05 | 0.07±0.00 |
>     | SWING    | 37.44±0.14 | 25.42±0.10 | 35.30±0.08 | 23.46±0.04 | 27.03±0.10 | 10.76±0.05 | 19.24±0.05 | 7.13±0.01  | 0.44±0.00 |
>
> - When compared against Table 1 in the paper, the total-score setting consistently produced higher overall performance: PD-type metrics increased, PP-type metrics decreased (as expected under reduced noise), and while some relative rankings shifted slightly, SWING remained the best-performing method across metrics.
>
> - These findings indicate that the comparative conclusions reported in the main paper remain stable even when GCS is encoded differently. For consistency with prior literature, we retained the original feature structure in the manuscript, but we now provide the total-score results to demonstrate robustness.
>
> ---
>
> **[Q5] Request for a deeper explanation of why IG works better than other methods, beyond empirical claims.**
>
> - We appreciate the reviewer’s request for a more detailed explanation. As noted above, our motivation is grounded in the findings of TIMING, which showed that IG outperforms several recent time series XAI methods once directionality is properly accounted for in evaluation. Earlier metrics ignored attribution signs, making some recent masking- or perturbation-based methods appear stronger than they actually were. Under direction-aware metrics such as CPD, IG consistently performs best.
>
> - Beyond empirical observations, there are also conceptual reasons: 1) IG is one of the few methods that reliably preserves the sign (+/–) of feature influence, which is essential for explaining prediction shifts rather than single predictions. 2) IG’s formulation as a path integral between two input states naturally aligns with our Delta-XAI setting, where the explanation target is the transition from $T_1$ to $T_2$. This makes IG particularly well-suited for modeling temporal trajectories and decomposing changes in prediction.
>
> - For these reasons—directional fidelity, path-integral consistency, and demonstrated superiority under direction-sensitive evaluation—we selected IG as the theoretical foundation for SWING.

---

> ### Author Response · Authors · 2025-11-23
>
> **[Q6] Question about how well the proposed quantitative metrics align with human interpretability in real clinical decision-making.**
>
> - We thank the reviewer for raising this important point. While our evaluation suite consists of quantitative metrics (faithfulness, sufficiency, completeness, efficiency), our new quantitative metrics primarily assess faithfulness and sufficiency—that is, whether the explainer correctly identifies the regions the model itself relies on. However, *human interpretability* is, in principle, somewhat orthogonal to these quantitative dimensions, making it inherently difficult to measure directly.
>
> - In principle, this stems from the fundamental nature of XAI itself: 1) when the model’s internal reasoning happens to align with human intuition, explanations naturally enhance trustworthiness, but 2) when the model uses patterns that are non-intuitive or even undesirable from a human perspective, the purpose of explanations is not to mirror human reasoning but rather to debug, audit, or monitor the model. Because these objectives differ at a conceptual level, human-centered interpretability cannot be fully captured by numerical metrics alone.
>
> - For this reason, a qualitative user study is essential. To address this, Section 6.4 includes a qualitative *Coherence Analysis*—along with Figure 10—explicitly designed to examine how well shift-based explanations align with clinically plausible reasoning, beyond what can be reflected in scalar metrics.
>
> - We also recognized that the original heatmap-style visualization made clinical interpretation challenging. In response, we have revised the visualization to a more intuitive, trajectory-based depiction that explicitly shows how each feature’s attribution accumulates from the previous prediction score to the current score in Figure 10. This clarified representation allows clinical practitioners to directly follow the explanatory trajectory and judge whether it matches domain expectations.
>
> - Thus, while our quantitative metrics capture important behavioral aspects of explanation quality, the revised qualitative analysis more directly reflects human-perceived interpretability, which is—*in principle*—orthogonal to purely numerical notions such as faithfulness or sufficiency.
>
> ---
>
> **[Q7] Inquiry about trade-offs between temporal window length, computational efficiency, and interpretability in SWING.**
>
> - We appreciate the reviewer’s question. The term temporal window may refer either to 1) the input window size, or 2) the temporal distance between the two scores used for shift computation. We examined both aspects in our experiments.
>
> - For input window size, Table 12 shows that changing the window from 24 → 48 (default) → 72 hours does not yield a consistent trend in interpretability performance. Similarly, Table 9 varies the time difference between scores (1 (default), 6, 24 hours), and again, we observe no systematic improvement or degradation in any interpretability metric.
>
> - As expected, computational efficiency naturally decreases as input size or temporal distance increases, but this change is not coupled with interpretability. Across both settings, SWING maintains stable behavior without exhibiting a predictable trade-off pattern.
>
> - Thus, based on our empirical results, we do not observe a systematic trade-off between temporal window length, interpretability quality, and computational cost.
>
> ---
>
> If anything in our response is still unclear or does not fully resolve your concerns, we would greatly appreciate additional questions and are glad to continue the discussion. We hope that our clarifications have thoroughly addressed the points you raised, and if they do align with your expectations upon further review, we would be sincerely grateful if you would consider updating your overall rating to reflect your revised perspective.

---

### Author Response · Authors · 2025-12-03
**Letter to Area Chairs**

Dear Area Chairs,
We thank the reviewers and Area Chairs for their constructive feedback. Below, we summarize the significance of our contribution and how the rebuttal and discussion resolved the key concerns raised during the reviewing process.

---

**Significance**

Reviewers agreed that our work addresses a practically important yet underexplored problem: explaining how model predictions change over time in online monitoring scenarios, rather than interpreting isolated predictions. They noted that real deployments rely on evolving risk trajectories, making temporal explanation the appropriate target for time series models (`y8Zs`, `hFWL`, `QpEH`).

Beyond the novelty of this setting, reviewers highlighted several strengths:

* **Coherent and comprehensive design:** The paper integrates a clear problem formulation, a wrapper that enables existing explainers in this setting, a temporal attribution method, and evaluation metrics aligned with prediction-change reasoning (`QpEH`, `z3oK`, `y8Zs`).
* **Methodological grounding:** Reviewers pointed to the strong motivation for using IG in temporal contexts and the principled decomposition of prediction changes (`z3oK`, `QpEH`).
* **Empirical rigor:** The breadth of baselines, datasets, and evaluations was viewed as unusually thorough for a new problem formulation (`z3oK`, `y8Zs`, `hFWL`).
* **Clarity and accessibility:** Reviewers found the manuscript clear and easy to follow, despite introducing a new explanation target and evaluation protocol (`z3oK`, `QpEH`).

Together, these assessments indicate that reviewers saw not only novelty, but a well-scoped and empirically validated contribution that establishes a foundation for explainability in temporal reasoning.

---

**Concerns Addressed Through Rebuttal and Discussion**

During the rebuttal, we resolved all major concerns raised by reviewers who participated in the discussion, and they subsequently raised their scores. Their concerns fall into the following themes:

* **Problem formulation and notation:** We clarified why prediction changes, rather than single-time predictions, are the explanation target and introduced key notation earlier to reduce ambiguity (`QpEH`, `y8Zs`).

* **Wrapper scope and semantics:** We explained the wrapper as an output-level operator, clarified sign handling, and aligned terminology to avoid overstating generality (`QpEH`, `y8Zs`).

* **Choice of IG and methodological grounding:** We justified IG’s suitability for temporal attributions and elaborated on why integration paths naturally align with change-based explanations (`z3oK`).

* **Experimental coverage and fairness:** We added evaluations on class changes, stable outputs, and larger temporal gaps, ensuring fair comparisons and realistic monitoring scenarios (`hFWL`, `y8Zs`).

* **Metric alignment and interpretability:** We clarified that our metrics are applied to wrapped outputs and reflect temporal decision dynamics rather than static importance scores (`y8Zs`).

* **Component rationale and ablations:** We clarified the purpose of RBS, DPI, and PHI and restructured ablations so each component’s contribution can be independently interpreted (`QpEH`).

* **Temporal window trade-offs:** We described how window size affects computational cost and attribution sharpness, addressing concerns about practical deployment (`z3oK`).

* **Applicability to other XAI methods:** We clarified how the wrapper generalizes to methods beyond IG and outlined how other explainers would integrate with the framework (`hFWL`).

All themes raised by reviewers who engaged in the discussion were resolved to their satisfaction, resulting in a score change from **6, 6, 4, 6** to **6, 6, 6, 8** prior to the system rollback. Each participating reviewer updated their evaluation after considering our rebuttal and clarifications.

---

**Remaining Concerns and Their Resolution**

The reviewers who did not join the discussion raised concerns that fall within the themes above:

* **Benchmark release and practitioner-facing interpretability:** We committed to releasing the data-processing and evaluation pipeline and expanded the discussion connecting our metrics to real-world interpretability (`z3oK`).

* **Broader applicability evidence:** We clarified how the wrapper generalizes beyond IG and added guidance for extending the method to other explainers (`hFWL`).

These concerns were first addressed through detailed rebuttal responses and are now reflected in the revised manuscript; no issues remain unresolved beyond the clarified themes.

---

Best regards,
Authors

---

### Meta-Review · Area_Chair_hTBR · 2026-01-12

**Summary:**

This paper aims to study time-series explainability by reframing the problem around explaining prediction changes in online monitoring. Specifically, the paper introduces a general Delta-XAI wrapper that enables reuse of many existing explainers, and develops SWING, an IG-based method that achieves state-of-the-art performance. The work is clearly written, well motivated, and supported by extensive experiments across metrics, backbones, and efficiency analyses. While initial reviews raised concerns about notation, wrapper semantics, metric alignment, and evaluation coverage, the rebuttal and discussion clarified the formulation, tightened claims, and added targeted experiments and reorganized ablations to resolve these issues. Given the improved clarity, strengthened empirical coverage, and a practical framing for deployed monitoring systems, I recommend accept.

**Reviewer Concerns:**

The initial reviews raised concerns about notation, wrapper semantics, metric alignment, and evaluation coverage, the rebuttal and discussion clarified the formulation, tightened claims, and added targeted experiments and reorganized ablations to address these issues, e.g., stable segments, class changes, and larger temporal gaps.

**Reviewer Scores:**

Reviewer y8Zs confirmed that the original score (4) will be raised.

---

### Decision · Program_Chairs · 2026-01-26

Accept (Poster)